# Anomalies, a mod 2 index, and dynamics of 2d adjoint QCD

Aleksey Cherman[1*], Theodore Jacobson[1†], Yuya Tanizaki[2,3‡] and Mithat Ünsal[3°]

**1** School of Physics and Astronomy, University of Minnesota, Minneapolis, MN 55455, USA
**2** Yukawa Institute for Theoretical Physics, Kyoto University, Kyoto, 606-8502, Japan
**3** Department of Physics, North Carolina State University, Raleigh, NC 27607, USA

* acherman@umn.edu, † jaco2585@umn.edu,
‡ yuya.tanizaki@yukawa.kyoto-u.ac.jp, ° unsal.mithat@gmail.com

## Abstract

We show that 2d adjoint QCD, an $SU(N)$ gauge theory with one massless adjoint Majorana fermion, has a variety of mixed 't Hooft anomalies. The anomalies are derived using a recent mod 2 index theorem and its generalization that incorporates 't Hooft flux. Anomaly matching and dynamical considerations are used to determine the ground-state structure of the theory. The anomalies, which are present for most values of $N$, are matched by spontaneous chiral symmetry breaking. We find that massless 2d adjoint QCD confines for $N > 2$, except for test charges of $N$-ality $N/2$, which are deconfined. In other words, $\mathbb{Z}_N$ center symmetry is unbroken for odd $N$ and spontaneously broken to $\mathbb{Z}_{N/2}$ for even $N$. All of these results are confirmed by explicit calculations on small $\mathbb{R} \times S^1$. We also show that this non-supersymmetric theory exhibits exact Bose-Fermi degeneracies for all states, including the vacua, when $N$ is even. Furthermore, for most values of $N$, 2d massive adjoint QCD describes a non-trivial symmetry-protected topological (SPT) phase of matter, including certain cases where the number of interacting Majorana fermions is a multiple of 8. As a result, it fits into the classification of $(1+1)$d SPT phases of interacting Majorana fermions in an interesting way.

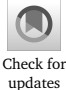 Check for updates

# 1   Introduction and Summary

The theory of strong nuclear interactions is described by quantum chromodynamics (QCD), a four-dimensional non-Abelian gauge theory with two characteristic low-energy phenomena: quark confinement and spontaneous chiral symmetry breaking. It remains a tough problem to understand these phenomena in a reliable theoretical manner. In order to tackle questions of this nature, it is quite helpful to think about related phenomena in a simpler setting. Therefore, in this paper we study confinement and chiral symmetry breaking in a two-dimensional non-Abelian gauge theory.

     The specific theory we examine is one-flavor adjoint QCD — an $SU(N)$ gauge theory coupled to one adjoint-representation Majorana fermion — in two spacetime dimensions. The Euclidean action of this theory is[1]

---

[1]In the literature on 2d adjoint QCD, the theory is defined by the first line of the action in Eq. (1), with $c_1$ and $c_2$ set to 0. The resulting theory is super-renormalizable, but it has renormalizable deformations with dimension-2 operators, and when writing down effective actions it is customary to include all renormalizable terms consistent with symmetries. The second line in Eq. (1) includes perturbatively marginal terms that are allowed by symmetries (including the chiral symmetry), which are discussed in detail in Section 2.

$$S = \int \mathrm{d}^2x \left[ \frac{1}{2g^2}\mathrm{tr}[G_{\mu\nu}G^{\mu\nu}] + \mathrm{tr}[\psi^T \mathrm{i}\gamma^\mu D_\mu] \right]$$
$$+ \int \mathrm{d}^2x \left[ \frac{c_1}{N}\mathrm{tr}(\psi_+\psi_+\psi_-\psi_-) + \frac{c_2}{N^2}\mathrm{tr}(\psi_+\psi_-)\mathrm{tr}(\psi_+\psi_-) \right], \tag{1}$$

where $G_{\mu\nu}$ is the $SU(N)$ field strength, $g$ is the gauge coupling, $\gamma = \mathrm{i}\gamma_1\gamma_2$, and $\psi$ is a massless Majorana fermion in the adjoint representation of $SU(N)$ with chiral components $\psi_\pm$. Generically, two-dimensional quantum field theories are more tractable than four-dimensional ones, providing a nice setting to better understand non-Abelian gauge dynamics. At the same time, and more interestingly, 2d adjoint QCD is not *too* simple, and still displays interesting strongly-coupled dynamics. Unlike the Schwinger model [1] and the 't Hooft model at large $N$ [2], it is not exactly solvable. Also, one of the earliest results about 2d adjoint QCD [3–5] is the existence of infinitely many Regge-like trajectories in the large $N$ limit, which is similar to the behavior of large $N$ QCD in four dimensions, but contrasts sharply with the far simpler spectrum of the 't Hooft and Schwinger models. The interesting dynamics of 2d adjoint QCD come from the fact that Majorana fermions in the adjoint representation have $O(N^2)$ propagating microscopic degrees of freedom, making the properties of 2d adjoint QCD more similar to those of four-dimensional confining gauge theories.

Our interest in 2d adjoint QCD also stems from the fact that this model is a setting where recently developed ideas about generalized global symmetries [6–8], along with a modern approach to semiclassical expansions for confining gauge theories initiated in Refs. [9,10], can be used to understand important but previously opaque aspects of strong-interaction dynamics. Many of our results follow from the existence of mixed 't Hooft anomalies, which are derived using a recent mod 2 index theorem, see Section 4.1 for its statement. Our analysis shows that massless 2d adjoint QCD is a strongly interacting confining gauge theory with intriguing and unusual spectral properties. In addition, our work suggests that 2d adjoint QCD fits into an interesting refinement of the classification of symmetry-protected topological phases for interacting Majorana fermions in one spatial dimension [11–18].

We now summarize our results in more detail, splitting the discussion into two categories. First, massless adjoint $SU(N)$ QCD in two spacetime dimensions gives a setting in which one can explore confinement, chiral symmetry breaking, and the nature of the excitation spectrum of a highly non-trivial non-Abelian gauge theory. We find that test charges in representations of $N$-ality $q$ are confined unless $q = N/2$, and give evidence that chiral symmetry is spontaneously broken for most values of $N$. More precisely, we find that discrete chiral symmetry $(\mathbb{Z}_2)_\chi$ is spontaneously broken for $N = 4n$, $4n + 2$, and $4n + 3$ as

$$(\mathbb{Z}_2)_\chi \to 1, \tag{2}$$

and that center symmetry $\mathbb{Z}_N^{[1]}$ is partially broken, for $N = 2n$, as

$$\mathbb{Z}_N^{[1]} \to \mathbb{Z}_{N/2}^{[1]}. \tag{3}$$

Moreover, it turns out that the bosonic and fermionic gauge-invariant excitations of the theory have degenerate energies for all even $N$. The reason that finding relations between bosonic and fermionic excitations is remarkable is that two-dimensional massless adjoint QCD with the action (1) is manifestly not supersymmetric at the Lagrangian level, because its microscopic matter content is not sufficient to fill out a minimal non-chiral 2d supersymmetry multiplet. So the states of the massless theory do not transform under the minimal non-chiral 2d supersymmetry algebra $\mathcal{N} = (1, 1)$. And yet, we will see that the physical spectrum of the theory

enjoys *exact* Bose-Fermi pairing for all even $N$. There is no exact Bose-Fermi degeneracy for finite odd values of $N$, but smoothness of the large $N$ limit requires Bose-Fermi degeneracies to emerge at odd $N$ in the large $N$ limit. At large $N$, the bosonic and fermionic spectrum is expected to exhibit Hagedorn growth, and our results imply that the associated spectral densities match *exactly* at large $N$. This gives a tractable parallel to similar surprising phenomena recently observed in 4d adjoint QCD [19–22].

Second, we find an interesting relation between our results on 2d adjoint QCD and some ideas from condensed matter physics. If we turn on a mass $m$ for the Majorana fermion with $m < 0$, it turns out that for most values of $N$ the vacuum of the theory is a symmetry-protected topological (SPT) phase. For even $N$, the non-trivial phase is protected by fermion parity $(-1)^F$ and center symmetry $\mathbb{Z}_N^{[1]}$. SPT phases of matter are trivially gapped, but their boundary theories must be described by interesting quantum field theories with 't Hooft anomalies for global symmetries [23, 24]. In the case of 2d adjoint QCD with even $N$, this SPT phase boundary corresponds to the domain wall associated with spontaneous $(\mathbb{Z}_2)_\chi$ chiral symmetry breaking in the $m \to 0$ limit. These domains walls host $(0 + 1)$d deconfined Majorana fermions, and anomaly inflow further shows that one can think of these edge modes as quarks of $N$-ality $N/2$. For $N = 2$, this is essentially the same phenomenon found in the boundary of the $S = 1$ Haldane phase [25–28]. These observations are discussed in detail in Sec. 5.

Moreover, we find that 2d adjoint QCD is also in an SPT phase for certain odd values of $N$, specifically $N = 4n + 3, n \in \mathbb{Z}_{\geq 0}$. To see why this is interesting, we recall that Fidkowski and Kitaev [14, 15], see also [16], showed that interactions change the classification of the SPT phases associated with Majorana fermions in one spatial dimension [11] from $\mathbb{Z}$ to $\mathbb{Z}_8$. They did this by showing that there exist $(\mathbb{Z}_2)_F$ and $(\mathbb{Z}_2)_\chi$ symmetry-preserving interactions that can drive systems with 8 Majorana fermions (or any integer multiple of 8) to a trivial gapped phase. Microscopically, adjoint QCD has $N^2 - 1$ adjoint Majorana fermions. For even $N$, $N^2 - 1$ is not divisible by 8, so our result that 2d adjoint QCD with even $N$ sits in a non-trivial SPT phase is consistent with Ref. [14, 15] in a straightforward way, although the $\mathbb{Z}_N^{[1]}$ center symmetry is an important extra ingredient. When $N$ is odd, on the other hand, $N^2 - 1$ is divisible by 8, so one might be tempted to guess that adjoint QCD should not be in a non-trivial SPT phase for any odd $N$. However, as mentioned above, when $N = 4n + 3$ 2d adjoint QCD realizes a non-trivial SPT phase which is protected by a combination of the fermion number $(\mathbb{Z}_2)_F$ and charge conjugation $(\mathbb{Z}_2)_C$ symmetries of the gauge theory. The $(\mathbb{Z}_2)_C$ symmetry flips the sign of $N(N-1)/2$ out of the $N^2 - 1$ fermions in adjoint QCD. Insisting on interactions consistent with these symmetries, one can construct SPT phases of interacting $8, 48, 120, 224, \ldots$ Majorana fermions. As a result, 2d adjoint QCD with $N = 4n + 3$ fits into the classification of interacting Majorana fermions in one spatial dimension in an interesting way.

Lastly, we note that 2d adjoint QCD theory was studied extensively, mostly in the 1990s, see e.g. Refs. [3–5, 29–59]. Among these works we must highlight the remarkable paper of Lenz, Shifman, and Thies [31], whose analysis uncovered some aspects of the 't Hooft anomalies of adjoint QCD for $N = 2$ long before discrete anomalies were widely understood. However, our results differ from e.g. the central claim of Ref. [35] (which was used in many other references above) that 2d $SU(N)$ massless adjoint QCD is not confining for any test charge representation. The differences, which we discuss in Sec. 9, stem from our use of powerful tools that were not available in the 1990s: a certain mod 2 index theorem and various discrete 't Hooft anomalies, a refined understanding of center symmetry and constraints on its realization, and an improved understanding of semiclassical expansions for gauge theory.

## 2 Symmetries of 2d adjoint QCD

To keep things simple, in this section (and indeed in most of the paper), we focus on spacetime manifolds of the form $\mathbb{R}^2$ and its flat compactifications to a cylinder and a torus. Let us write the action of 2d adjoint QCD more carefully than we did in the introduction. It takes the form

$$S = S_{\text{kinetic}} + S_{\text{mass}} + S_{\psi^4}. \tag{4}$$

The part of the action $S_{\psi^4}$ which contains terms quartic in the quarks $\psi$ will be written after a discussion of the global symmetries we wish to impose. The kinetic term $S_{\text{kinetic}}$ is

$$S_{\text{kinetic}} = \int \mathrm{d}^2 x \left\{ \frac{1}{2g^2} \mathrm{tr}[G_{\mu\nu}G^{\mu\nu}] + \mathrm{tr}[\psi^T \mathrm{i}\gamma^\mu D_\mu \psi] \right\}, \tag{5}$$

where $G_{\mu\nu}$ is the field strength of the $SU(N)$ gauge field $a_\mu = a_\mu^a T^a$, $T^a$ are the generators of $\mathfrak{su}(N)$ in the fundamental representation normalized to $\mathrm{tr}\, T^a T^b = \frac{1}{2}\delta^{ab}$, and $\psi$ is the Majorana fermion in adjoint representation, $D_\mu \psi = \partial_\mu \psi + \mathrm{i}[a_\mu, \psi]$. The transpose, $^T$, in $\psi^T$ is understood to act only on the spinor indices. We write the Euclidean gamma matrices $\gamma^\mu$ as $\gamma^1 = \sigma_1$ and $\gamma^2 = \sigma_3$, and take $\gamma = \mathrm{i}\gamma^1\gamma^2 = \sigma_2$. The mass term is

$$S_{\text{mass}} = \int \mathrm{d}^2 x \, \mathrm{tr}[m\,\psi^T \gamma \psi]. \tag{6}$$

The other candidate mass term $\mathrm{tr}[\psi^T \psi]$ vanishes identically due to anti-commutativity of $\psi$.

Now we can discuss the internal symmetries of the massless theory. The internal symmetry $G$ of (5) with $m = 0$ is[2]

$$G = \begin{cases} \mathbb{Z}_N^{[1]} \times (\mathbb{Z}_2)_F \times (\mathbb{Z}_2)_\chi & (N = 2), \\ \left[\mathbb{Z}_N^{[1]} \rtimes (\mathbb{Z}_2)_C\right] \times (\mathbb{Z}_2)_F \times (\mathbb{Z}_2)_\chi & (N > 2). \end{cases} \tag{7}$$

The actions of $\mathbb{Z}_N^{[1]}$ and $(\mathbb{Z}_2)_C$ involve the gauge fields. The one-form center symmetry $\mathbb{Z}_N^{[1]}$ only acts on Wilson loops, and will be discussed in detail in the next section. The charge conjugation symmetry $(\mathbb{Z}_2)_C$ acts as

$$U_{\mathsf{C}} : a_{ij}^\mu \mapsto -a_{ji}^\mu, \quad \psi_{ij} \mapsto \psi_{ji}, \tag{8}$$

where $i, j = 1, \dots, N$ are color indices. Charge conjugation is a global symmetry only when $N > 2$; when $N = 2$ the transformation in Eq. (8) is an $SU(2)$ gauge transformation. The $(\mathbb{Z}_2)_F$ and $(\mathbb{Z}_2)_\chi$ symmetries act only on the fermions.[3] Fermion parity $(\mathbb{Z}_2)_F$ is the vector-like symmetry which acts as

$$U_{\mathsf{F}} : \psi \mapsto -\psi, \tag{9}$$

---

[2] The fact that center symmetry does not commute with $(\mathbb{Z}_2)_C$ was recently emphasized in Ref. [60].

[3] We should check that we have found all of the internal symmetries. Let us first discuss the subgroup that acts linearly on $\psi$ but does not change the gauge field $a_\mu$. Any such symmetry $U$ should be a symmetry of $N^2-1$ free Majorana fermions, so $U \in O(N^2-1)_{\mathrm{L}} \times O(N^2-1)_{\mathrm{R}}$. But because we are looking for symmetries that do not affect $a_\mu$, to be a symmetry of Eq. (5), $U$ must commute with gauge transformations. Since the adjoint representation of $SU(N)$ is irreducible on the left- and right-moving Majorana fermions, Schur's lemma implies that $U$ must be proportional to the identity on each of the left- and right-moving Majorana fermions. All such transformations can be written as combinations of $U_{\mathsf{F}}$ and $U_\chi$.

Now, let us examine the most general symmetry that can also act on $a_\mu$ in a nontrivial way. This corresponds to calculating the normalizer $\mathcal{N}$ of $SU(N)$ in $O(N^2-1)_{\mathrm{L}} \times O(N^2-1)_{\mathrm{R}}$. The discussion in the paragraph above gives the centralizer $\mathcal{C} = (\mathbb{Z}_2)_F \times (\mathbb{Z}_2)_\chi \subset \mathcal{N}$, so what we are really looking for is the quotient group $\mathcal{N}/\mathcal{C}$. But in general, the quotient group $\mathcal{N}/\mathcal{C}$ is a subgroup of the outer automorphism group, which is nothing but charge conjugation $(\mathbb{Z}_2)_C$ for $SU(N)$ with $N \geq 3$, while for $N = 2$ it is trivial. This completes the discussion on the fact that we have found all the symmetry of the Lagrangian.

We thank Zohar Komargodski for emphasizing the importance of making this argument precise and pointing out that it is useful to consider the normalizer in doing so.

while the discrete chiral symmetry $(\mathbb{Z}_2)_\chi$ acts as

$$U_\chi : \psi \mapsto \gamma \psi \,. \tag{10}$$

For future use, note that $\gamma$ can be used to define left-moving and right-moving components of $\psi$ by

$$\psi_\pm = \tfrac{1}{2}(1 \pm \gamma)\psi \,. \tag{11}$$

The system also has reflection symmetries $x_1 \mapsto -x_1$, $x_2 \mapsto -x_2$. The associated transformations on the fermions are[4]

$$R_1 : \psi(x_1, x_2) \mapsto \gamma^2 \psi(-x_1, x_2) \quad \text{and} \quad R_2 : \psi(x_1, x_2) \mapsto \gamma^1 \psi(x_1, -x_2) \,. \tag{12}$$

Turning on $m \neq 0$ breaks $(\mathbb{Z}_2)_\chi$ and the spacetime reflection symmetries, but does not break the other internal symmetries. Since there is no massive deformation at the quadratic level consistent with $(\mathbb{Z}_2)_\chi$, there may be quantum anomalies that involve $(\mathbb{Z}_2)_\chi$. These anomalies are discussed in Sec. 4.

So far, we have analyzed the symmetries of the fermion and gauge field kinetic terms in Eq. (5). In 2d, a theory with an action which includes only the kinetic terms is super-renormalizable because the gauge coupling $g$ has mass dimension 1. But it is important to note that the arguments in this paper can be applied to a modified version of the theory with certain four-fermion interaction terms. These four-fermion terms are perturbatively renormalizable in 2d and do not break any of the symmetries discussed above. Since the Lagrangian (5) without the four-Fermi terms is super-renormalizable, we do not have to introduce them as UV counterterms at the super-renormalizable point. But the point where the four-Fermi terms are set to 0 is not protected by any symmetries, so not writing them is a fine-tuning in the sense of Wilsonian effective field theory.

Let us discuss these missing terms in the action explicitly. All fermion bilinear operators that were not already written in Eq. (5) break spacetime reflection or chiral symmetries,[5] so they can be consistently set to zero when $m = 0$. However, when $N > 2$, there are two independent terms quartic in $\psi$ which are consistent with the $(\mathbb{Z}_2)_F, (\mathbb{Z}_2)_\chi, (\mathbb{Z}_2)_C$ and $\mathbb{Z}_N^{[1]}$ symmetries:

$$S_{\psi^4} = \int \mathrm{d}^2 x \left[ \frac{c_1}{N} \mathrm{tr}(\psi_+ \psi_+ \psi_- \psi_-) + \frac{c_2}{N^2} \mathrm{tr}(\psi_+ \psi_-)\mathrm{tr}(\psi_+ \psi_-) \right] \,. \tag{14}$$

When $N = 2$, these terms are related by a trace identity, so one should only write one of them. The couplings $c_i$ run logarithmically under renormalization-group (RG) flow. To get a smooth large $N$ limit, one should hold $\lambda = g^2 N$ fixed along with $c_1, c_2$. Then, all terms in the action are $O(N^2)$. At large $N$, the running of $c_2$ does not affect the running of $c_1$. At

---

[4]These reflections satisfy $R_i^2 = 1$ and commute with the Dirac operator $\slashed{D}$. They can be used to define a Pin$^+$ structure on non-orientable manifolds. But if we wanted to discuss the theory with $m \neq 0$ on nonorientable manifolds, things are more subtle, because the mass term cannot be invariant under Euclidean spacetime reflections R with $R^2 = 1$. So 2d adjoint QCD with $m \neq 0$ cannot be defined on non-orientable manifolds with a Pin$^+$ structure. Instead, one can use the Pin$^-$ structure, because one can define another spacetime reflection R′ with $R'^2 = -1$ [61]. Then the mass term is invariant under R′. Exploring 2d adjoint QCD on non-orientable manifolds would probe whether there are some 't Hooft anomalies involving spacetime reflection symmetries. We leave exploring this to other works.

[5]Apart from the mass term, one can consider two classically marginal fermion bilinear Pauli-like terms

$$d_1 \mathrm{tr}(\psi^T \psi \, \varepsilon_{\mu\nu} G_{\mu\nu}) + d_2 \mathrm{tr}(\psi^T \gamma \psi \, \varepsilon_{\mu\nu} G_{\mu\nu}) \,. \tag{13}$$

These terms can be consistently set to zero in the action when $m = 0$, because the first term breaks the spacetime reflection symmetries, while the second term breaks $(\mathbb{Z}_2)_\chi$. But if $m \neq 0$, they are not forbidden by symmetries.

any $N$, by an appropriate choice of signs for the coefficients $c_i$, these couplings can be made asymptotically free in the UV. Consequently, without fine tuning, they can be relevant at long distances, and induce corresponding dynamical strong scales, similarly to the Gross-Neveu model [62]. These non-perturbatively generated mass scales control the long-distance physics along with the Yang-Mills mass scale $\lambda$. The fact that 2d adjoint QCD allows these marginally-relevant interaction terms was not previously appreciated.[6]

Finally, we note that there is a four-Fermi term which preserves the $(\mathbb{Z}_2)_F, (\mathbb{Z}_2)_\chi$ and $\mathbb{Z}_N^{[1]}$ symmetries, but breaks the $(\mathbb{Z}_2)_C$ symmetry:

$$S_{\text{C-breaking}} = \int \mathrm{d}^2 x \, \frac{c_3}{N} \, \mathrm{tr}(\psi_+ \psi_- \psi_+ \psi_-). \tag{15}$$

This term can be made marginally relevant by a choice of sign of $c_3$. In the majority of this paper, we assume the existence of the $(\mathbb{Z}_2)_C$ symmetry. This corresponds to setting $c_3 = 0$.

## 3 Center symmetry in two spacetime dimensions

Adjoint QCD has a global $\mathbb{Z}_N$ center symmetry. In the language of Ref. [6], center symmetry is a one-form symmetry, so in what follows we will denote it by $\mathbb{Z}_N^{[1]}$. This just means that it acts on some line operators - specifically, Wilson loop operators - but does not act on local operators. For example, given a Wilson loop of the fundamental representation along a closed loop $C$

$$W(C) = \mathrm{tr}\left( \mathcal{P} \exp\left[ \mathrm{i} \int_C a \right] \right), \tag{16}$$

where $a = a_\mu \mathrm{d}x^\mu$ is the $SU(N)$ gauge field written locally as a one-form, center symmetry acts by

$$W(C) \mapsto \mathrm{e}^{2\pi \mathrm{i}/N} W(C). \tag{17}$$

In the language of Ref. [6], ordinary (zero-form) symmetries are generated by codimension-1 topological defects,[7] while one-form symmetries are generated by codimension-2 topological defects [6]. In two dimensions, codimension-2 "manifolds" are just points. Let us denote the symmetry-generating defects associated to center symmetry by

$$U_S(p), \tag{18}$$

where $p$ is a point in spacetime. Then $U_S(p)$ satisfies

$$\langle W(C) U_S(p) \cdots \rangle = \exp\left( \frac{2\pi \mathrm{i}}{N} \mathrm{Link}(C, p) \right) \langle W(C) \cdots \rangle, \tag{19}$$

where $\mathrm{Link}(C, p)$ is the linking number between the loop $C$ and the point $p$. For many practical purposes, it is useful to note that the insertion of the defect operator $U_S$ is equivalent to performing the path integral with non-trivial 't Hooft magnetic flux [64].

---

[6] Similar terms, like $(\bar{\psi}\gamma_\mu \psi)(\bar{\psi}\gamma^\mu \psi)$, can also be written in the Schwinger model [63].

[7] For readers to whom this language is unfamiliar, we recall that this is simply a formal but useful way of saying that symmetries are generated by charges. For standard continuous symmetries, one can compute a charge $Q$ by integrating the time component of a current density $j_0$ over a region in space $M_{\text{space}}$. In that context, the "codimension-1 topological defect" generating the symmetry is the exponential of the charge $\mathrm{e}^{\mathrm{i}Q}$, and the codimension-1 manifold in question is just $M_{\text{space}}$. The defect is topological in the sense that varying $M_{\text{space}}$ does not change the properties of the defect, unless of course the new region includes new charged operators. This language may seem overly formal when discussing ordinary symmetries, but it is useful in working with discrete symmetries, and especially with higher-form symmetries.

The realization of center symmetry is a diagnostic for quark confinement: an unbroken center symmetry is associated to an area law for the expectation values of large Wilson fundamental-representation loops.[8] The realization of center symmetry can also be related to the expectation values of topologically non-trivial Wilson loops where $C$ goes around a large non-contractible circle $S^1$ in spacetimes like $\mathbb{R} \times S^1$. If we define the Polyakov loop $P(x) = \mathcal{P} \exp\left[i \int_{S^1} a_0(x_0, x) dx_0\right]$, then an unbroken center symmetry corresponds to

$$\lim_{x \to \infty} \langle \text{tr}(P(x))^q \text{tr}(P^\dagger(0))^q \rangle = 0 \text{ for all } q \neq 0 \bmod N . \tag{20}$$

It is a standard fact that these probes of center symmetry are related. If the $S^1$ direction $x_0$ has circumference $\beta$, one can relate the behavior of the correlation function

$$\langle \text{tr}(P(x))^q \text{tr}(P^\dagger(0))^q \rangle = e^{-F_q(x)} \tag{21}$$

to the expectation value of a rectangular $N$-ality $q$ Wilson loop extending along the $x_1$ by a distance $x$ and along the $x_0$ direction by a distance $\beta$. With unbroken center symmetry, the function $F_q(x)$ scales for large $x$ and $\beta$ as $F(x) \sim x\beta\sigma_q$, where $\sigma_q$ is the $q$-string tension, verifying the desired link between the two order parameters.

Gauging center symmetry means summing over all possible choices of 't Hooft flux boundary conditions in the path integral, or, equivalently, summing over all possible insertions of $U_S(p)$.[9] This reduces the gauge group to $SU(N)/\mathbb{Z}_N$. This was not widely appreciated in the 1990s, and much of the literature on adjoint QCD from that period asserted that the gauge group is necessarily $SU(N)/\mathbb{Z}_N$. From the modern perspective, this is not mandatory. One can define adjoint QCD as an $SU(N)$ gauge theories without violating unitarity and locality. Therefore one can choose whether center symmetry is a gauge symmetry, in which case the gauge group is $SU(N)/\mathbb{Z}_N$, or a global symmetry, in which case the gauge group is $SU(N)$. Importantly, Wilson loops are not genuine gauge-invariant line operators in $SU(N)/\mathbb{Z}_N$ theories [65].[10]

Reference [6] observed that discrete one-form symmetries are always unbroken in 2d. The argument for this result has some assumptions that were not spelled out in Ref. [6], and we will discuss them below. However, to the extent that the argument of Gaiotto et. al. can be applied to 2d adjoint QCD, it would imply that center symmetry cannot be spontaneously broken, so that 2d adjoint QCD is in a confining phase on $\mathbb{R}^2$, with an area law for large fundamental-representation Wilson loops. We will see that this conclusion is almost correct. The only exception is that when $N$ is even, representations with $N$-ality $N/2$ are not confined.

Let us review the argument for the absence of spontaneous symmetry breaking of discrete one-form symmetries in two spacetime dimensions. The key observation is that the behavior of the order parameter $\langle W[S^1] \rangle$ is determined by an effective field theory (EFT) formulated in the $d-1$ spacetime dimensions transverse to the Wilson loop. From the point of view of this lower-dimensional EFT, $W[S^1]$ is just a local operator, so the one-form center symmetry acting

---

[8]To make this precise, let $V$ be the spatial volume and let $\ell$ be the radius of the Wilson loop $W(C)$. Then, the large Wilson loop order parameter can be defined as $\lim_{\ell \to \infty} \lim_{V \to \infty} \langle W(C) \rangle$, and one can discuss area law by taking limits in this specific order.

[9]In a lattice description, gauging the center symmetry corresponds to multiplying plaquettes by $\mathbb{Z}_N$ phases, and summing over all possible ways of doing so. In a continuum description [7], center symmetry can be gauged by introducing a $U(1)$ two-form gauge field $B$, which obeys the constraint $\frac{N}{2\pi}\int_{M_2} B \in \mathbb{Z}$. To gauge the symmetry, the $SU(N)$ gauge field $a$ should be embedded into a $U(N)$ gauge field $\tilde{a}$, and then one should demand that the (minimally-coupled) Lagrangian be invariant under the one-form gauge transformations, $B \mapsto B+d\lambda$ and $\tilde{a} \mapsto \tilde{a}+\lambda$.

[10]Two-dimensional $SU(N)/\mathbb{Z}_N$ gauge theory has an additional parameter, called a discrete $\theta$ angle, see Refs. [7, 65] for related 4d discussions. In 2d, the number of insertions of $U_S$ can be viewed as a topological invariant modulo $N$, so we can attach the $\mathbb{Z}_N$ phase to the path integral in gauging $\mathbb{Z}_N^{[1]}$: $\mathcal{Z}_{SU(N)/\mathbb{Z}_N, p} / \mathcal{Z}_{SU(N)} = \sum_k e^{2\pi i p k/N} \langle U_S(x)^k \rangle_{SU(N)}$. This label $p \in \mathbb{Z}_N$ corresponds to the discrete theta angle discussed in Ref. [29].

on $W[S^1]$ is just a conventional zero-form symmetry in the EFT. But it is a standard fact that zero-form symmetries cannot break spontaneously in one spacetime dimension (up to some caveats we explain next): there is no spontaneous symmetry breaking in quantum mechanics. As long as these remarks apply, the lowest spacetime dimension in which a discrete one-form center symmetry can break spontaneously is $d = 3$. In particular, center symmetry cannot be spontaneously broken in a two-dimensional quantum field theory like 2d adjoint QCD.

The reason that symmetries generally cannot break in quantum mechanics is the existence of quantum tunneling. Generically, the would-be ground states charged under discrete global symmetries can be connected by tunneling events with non-zero amplitudes. These tunneling processes, which are associated to finite-action instantons in the path integral formalism, restore global symmetries. The only way to evade symmetry restoration by instantons is to ensure that all the tunneling amplitudes sum to exactly zero.

This is hard to do. We have been able to think of only four options:

(a) If the action is not purely real, tunneling amplitudes might have non-trivial phase factors, and then one might hope that they cancel exactly. The only known way to achieve this without fine tuning is to have an appropriate mixed 't Hooft anomaly which forces this conclusion. For an example of such a system in quantum mechanics, see Ref. [8].

(b) If a theory has dimensionless parameters, one can imagine that perhaps they can be fine tuned to make string tensions vanish.

(c) If one considers a limit where the number of degrees of freedom per point diverges, such as a large $N$ limit, then the actions of instantons associated with tunneling between would-be ground states might scale with a positive power of $N$. Then a discrete 1-form symmetry can break at $N = \infty$ even with less than 3 non-compact directions. This phenomenon has been seen in studies of $SU(N)$ gauge theories compactified on $S^3 \times S^1$ [66–68].

(d) It could be that all tunneling events connecting candidate ground states carry exact fermion zero modes. This requires an appropriate index theorem, which in turn requires an appropriate mixed 't Hooft anomaly. Instantons that carry robust fermionic zero modes cannot connect bosonic states. Then tunneling is forbidden, the candidate ground states become separated into distinct selection sectors, and the discrete symmetry is spontaneously broken.[11]

Option (a) is irrelevant to 2d adjoint QCD, except possibly in the limited form we discuss in option (b) below. The basic issue is that two-dimensional $SU(N)$ adjoint QCD has no topological $\theta$ term, because

$$\theta \int \text{tr}(G) = 0 \tag{22}$$

due to the color trace. So there is no 'topological' way to attach phases to tunneling amplitudes.

Option (b) naively requires a theory to have $\sim N/2$ dimensionless parameters which affect the string tension. To see this, note that an $SU(N)$ gauge theory generally has $\lfloor N/2 \rfloor$ independent string tensions if charge conjugation symmetry is not spontaneously broken, because charge conjugation relates string tensions for charges of $N$-ality $q$ and $N-q$. But as we explain in Sec. 2, massless adjoint QCD with charge conjugation symmetry has only two dimensionless parameters when $N > 2$. While we know of no reason for it to happen, one might imagine that by tuning these two independent parameters one can make two independent string tensions vanish. It is possible that the process by which the tunneling amplitudes would vanish in this

---

[11]This discussion is formulated for a bosonic discrete symmetry, so that all candidate ground states are bosonic.

scenario would involve the fermion effective action, which is not always manifestly positive in the presence of Hubbard-Stratonovich auxiliary fields associated to the four-Fermi terms, see Appendix A. But once $N \geq 7$, there are more independent string tensions than independent parameters. Any such $\mathbb{Z}_N^{[1]}$-breaking-by-fine-tuning mechanism (which seems far-fetched in the first place) should stop working once $N \geq 7$.

Option (c) implies that the large $N$ limit and the large $N_f$ limit (where $N_f$ is the number of adjoint fermions) require special consideration. We discuss the large $N$ limit in Section 8.1. In this paper we focus on $N_f = 1$, and leave a discussion of what happens at $N_f > 1$ to a separate paper.

Option (d) is the most interesting one for our purposes. For example, consider the charge $q$ Schwinger model — 2d electrodynamics coupled to charge $q$ fermions — which has some superficial similarities to 2d adjoint QCD. It has a $\mathbb{Z}_q^{[1]}$ one-form center symmetry, and also a $\mathbb{Z}_{2q}$ chiral symmetry. It is known that it is in a screening phase on $\mathbb{R}^2$ [69–71], in the sense that center symmetry is spontaneously broken to $\mathbb{Z}_1$.[12] The reason this classic result is consistent with our discussion is that there is a mixed 't Hooft anomaly involving $\mathbb{Z}_q^{[1]}$ and $\mathbb{Z}_{2q}$. This mixed anomaly gives rise to an index theorem, forcing the instantons which could have naively restored the $\mathbb{Z}_q^{[1]}$ symmetry to carry exact fermion zero modes. Therefore tunneling is forbidden, and center symmetry is spontaneously broken in the charge-$q$ Schwinger model without conflicting with our refinement of the theorem of Ref. [6].

This means that to understand whether 2d adjoint QCD confines, it is essential to understand the 't Hooft anomalies for its global symmetries, and then work out what they imply for tunneling amplitudes connecting candidate center-breaking vacua. This will be our goal in the following sections.

# 4 't Hooft anomalies and the mod 2 index

In this section we explore mixed 't Hooft anomalies between $(\mathbb{Z}_2)_\chi$ and the other discrete symmetries of 2d adjoint QCD. We will see that for certain choices of background gauge fields for $(\mathbb{Z}_2)_F, (\mathbb{Z}_2)_C$ and $\mathbb{Z}_N^{[1]}$, the Euclidean partition function of 2d adjoint QCD transforms as

$$(\mathbb{Z}_2)_\chi : \mathcal{Z} \mapsto e^{\pi i \zeta} \mathcal{Z} = -\mathcal{Z}, \qquad \text{heuristically} \tag{23}$$

under $(\mathbb{Z}_2)_\chi$ transformations. More precisely, the 't Hooft anomaly is tied to the presence of fermionic zero modes, and the fermion path-integral measure gets this phase in appropriate background gauge fields for $(\mathbb{Z}_2)_F, (\mathbb{Z}_2)_C$ and $\mathbb{Z}_N^{[1]}$, as we shall see in Sec. 4.2. When present, these fermionic zero modes make the partition function vanish. This is what makes the expression in Eq. (23) heuristic. Quantities like the non-normalized "expectation value" $\langle \psi^T \gamma \psi \rangle$ become non-zero on compact spacetimes, and ratios of partition functions with positive and negative fermion masses differ by a sign, see Sec. 5.1. These features can be interpreted as evidence for the existence of mixed 't Hooft anomalies.

The quantity $\zeta$ is a mod 2 index for the Dirac operator for adjoint Majorana fermions. We define $\zeta$ to be the number of zero modes of the Dirac operator $\not{D}$ with positive chirality, and review the proof that $\zeta$ is a topological invariant mod 2. The argument for the topological invariance of $\zeta$ is implicit in Ref. [78], is mentioned in passing in Ref. [61], and discussed in detail in Ref. [79]. This index was recently discussed in a variety of contexts in Refs. [17, 80–83]. Indeed, as we have already mentioned, given the connection between 't Hooft anomalies

---

[12]Screening behavior in the charge-1 Schwinger model was understood much earlier, see e.g. [1, 72–74], where the test charge was taken to be any real number. Also, we should note that supersymmetric versions of 2d charge-$q$ Abelian gauge theories were discussed in Refs. [75–77] in a string theory context.

and SPT phases, one can interpret much of the literature on fermionic SPT phases in one spatial dimension as discussions of 't Hooft anomalies of 2d Majorana fermions.

### 4.1 The mod 2 index theorem and symmetries of the Dirac spectrum

Let us review the proof of the mod 2 index theorem for the Dirac operator $\slashed{D}$ associated to a Majorana fermion in two spacetime dimensions [61,78,79]. In this section, we will work on an arbitrary orientable closed spacetime manifold $M_2$. In two dimensions, orientable manifolds are also spin manifolds. The restriction to orientable manifolds allows us to use the fact that chirality as computed from

$$\gamma \equiv i\gamma^1\gamma^2 \tag{24}$$

is globally well-defined. In the conventions explained in Appendix A, $\gamma = \sigma_2$.

First, note that $(\slashed{D})^\dagger = -\slashed{D}$, meaning that the Dirac spectrum is purely imaginary. The eigenvalue equation is

$$\slashed{D}u = i\lambda u, \tag{25}$$

with $\lambda \in \mathbb{R}$ and $u : M_2 \to \mathbb{C}^2 \otimes \mathfrak{su}(N)$, where $M_2$ is an orientable spacetime manifold. Since $\slashed{D}$ is a real anti-symmetric matrix, we can complex conjugate Eq. (25) to get

$$\slashed{D}u^* = -i\lambda u^*. \tag{26}$$

Here, $u^*$ should be understood as $(u^a)^* T^a$, meaning that we do not complex conjugate the basis of $\mathfrak{su}(N)$. This shows that, for $\lambda \neq 0$, we have a pair of eigenstates $u$ and $u^*$ with the eigenvalues $\pm i\lambda$.

Using $\{\gamma, \slashed{D}\} = 0$ one can check that

$$\slashed{D}(\gamma u^*) = i\lambda(\gamma u^*), \tag{27}$$

so that $u$ and $\gamma u^*$ have the same eigenvalue. Moreover, the eigenvectors $u$ and $\gamma u^*$ are linearly independent. If they were not, we could obtain $u = \gamma u^*$ by an appropriate phase rotation. But then $u = \gamma(\gamma u^*)^* = -u$, due to the identities $\gamma^2 = 1$ and $\gamma^* = -\gamma$. That would mean that $u = 0$, which would be a contradiction. We therefore find degenerate pairs of eigenstates associated to the eigenvalues $\pm i\lambda$:

$$\begin{array}{c||c|c} +i\lambda & u & \gamma u^* \\ \hline -i\lambda & u^* & \gamma u \end{array} \tag{28}$$

Out of these four states, we can construct two positive-chirality states and two negative-chirality, while they are not eigenstates of $\slashed{D}$:

$$u_\pm = u \pm \gamma u, \ u_\pm^* = u^* \pm \gamma u^*. \tag{29}$$

This double degeneracy of eigenstates on orientable manifolds is analogous to Kramers doubling, and ensures that the path-integral measure for the non-zero Dirac spectrum is invariant under discrete chiral transformations [61]. Thanks to this feature, the Pfaffian for the nonzero spectrum can be defined to be a positive semi-definite function of gauge fields, $\text{Pf}'(i\slashed{D}-m\gamma) \geq 0$, for any gauge fields $a$ and real mass $m \in \mathbb{R}$, including $m = 0$. This positivity result can also be shown using the idea of Majorana positivity [84–86], as discussed in Appendix A.

Next, let us consider the zero modes, which satisfy $\slashed{D}u_0 = 0$. Since $\slashed{D}$ vanishes, we can say trivially that $[\gamma, \slashed{D}] = 0$ on that subspace, and $\gamma$ and $\slashed{D}$ are simultaneously diagonalizable. Therefore, let us say that we have a zero-mode with positive chirality,

$$\slashed{D}u_{0,+} = 0, \ \gamma u_{0,+} = u_{0,+}. \tag{30}$$

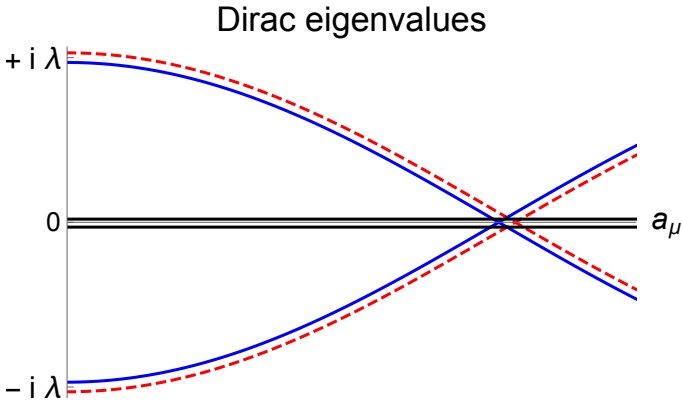

Figure 1: Sketch of the flow of eigenvalues of the massless Dirac operator as a function of the bosonic field $a_\mu$ on an orientable manifold. The non-zero eigenvalues come in quartets, with two modes with eigenvalue $+i\lambda$ and another two with eigenvalue $-i\lambda$. We also sketch a pair of Dirac zero modes with opposite chirality. Since quartets of non-zero modes must go through zero together, the number of e.g. right-handed zero modes mod 2 is a topological invariant, which we call $\zeta$. For clarity of presentation, all degenerate eigenvalues are shown slightly offset from each other.

Taking the complex conjugate of these expressions, we find that

$$\slashed{D} u_{0,+}^* = 0, \ \gamma u_{0,+}^* = -u_{0,+}^*, \tag{31}$$

so we find a zero-mode with negative chirality $u_{0,-} = u_{0,+}^*$. Therefore, on orientable manifolds, we always have the same number of zero modes for positive and negative chiralities. This implies that the usual index, in the sense of Atiyah and Singer, vanishes:

$$\text{index}(\slashed{D}) = \dim \ker(\slashed{D}) - \dim \text{coker}(\slashed{D}) = 0. \tag{32}$$

Of course, this is consistent with the standard Atiyah-Singer index theorem because $\frac{1}{2\pi} \int \text{tr}(G) = 0$ when the gauge group is $SU(N)$.

Still, it is possible to define a non-trivial mod 2 index on 2d orientable manifolds. Consider an arbitrary continuous change of gauge fields, the metric, and so on. Then, in general, the number of zero modes will change, because non-zero modes can become zero modes, and vice versa. The total number of zero modes mod 2 is always 0. However, the number of zero-modes with positive chirality cannot change mod 2, because the non-zero spectrum always consists of two positive and two negative chiralities, as we have seen above. We sketch the flow of the eigenvalues of the Dirac operator as a function of bosonic fields $a_\mu$ in Fig. 1. This discussion completes our review of the mod 2 index theorem relevant to our discussion. The theorem can be summarized as follows [61,78,79]:

**Theorem 1** *The number $\zeta$ of zero modes of the Dirac operator with positive chirality mod 2,*

$$\zeta = \dim(\ker \slashed{D} \cap \{\gamma = +1\}) \ \text{mod} \ 2 \tag{33}$$

*is a topological invariant on two-dimensional closed orientable manifolds.*

We will refer to $\zeta$ as the mod 2 index of the Dirac operator.

## 4.2 The mod 2 index and mixed 't Hooft anomalies with discrete chiral symmetry

An 't Hooft anomaly is an obstruction to gauging a global symmetry. If one can resolve the obstruction by coupling the theory to a topological action in one dimension higher, then the

obstruction is invariant under RG flow. This important theorem, called 't Hooft anomaly matching [87–89], was first established for continuous chiral symmetries in even dimensions, and the recent development of topological phases shows that anomaly matching arguments can be generalized to a much broader class of symmetries (see, e.g., Refs. [8, 69–71, 90–117]).

In order to compute the 't Hooft anomaly of 2d massless adj QCD, the mod 2 index $\zeta$ defined in the previous subsection is of great importance. To see this explicitly, let us write a mode decomposition of the fermion field $\psi(x)$ as

$$
\begin{aligned}
\psi(x) \;=\; & \sum_{i_0=1}^{\zeta} \left( \widetilde{\psi}_{i_0,+} u_{0(i_0),+}(x) + \widetilde{\psi}_{i_0,-} u_{0(i_0),-}(x) \right) \\
& + \sum_{\lambda \neq 0} \left( \widetilde{\psi}_{\lambda,1} u_\lambda(x) + \widetilde{\psi}_{-\lambda,1} u_\lambda^*(x) + \widetilde{\psi}_{\lambda,2} \gamma u_\lambda^*(x) + \widetilde{\psi}_{-\lambda,2} \gamma u_\lambda(x) \right),
\end{aligned}
\tag{34}
$$

where $\widetilde{\psi}$'s are Grassmannian variables, and $u$'s are the eigenfunctions of the Dirac operators. We can formally define the gauge-invariant fermion measure as

$$
\mathcal{D}\psi = \prod_{i_0=1}^{\zeta} \mathrm{d}\widetilde{\psi}_{i_0,+} \mathrm{d}\widetilde{\psi}_{i_0,-} \prod_{\lambda \neq 0} \mathrm{d}\widetilde{\psi}_{\lambda,1} \mathrm{d}\widetilde{\psi}_{-\lambda,1} \mathrm{d}\widetilde{\psi}_{\lambda,2} \mathrm{d}\widetilde{\psi}_{-\lambda,2}.
\tag{35}
$$

Then a schematic expression for the path integral is given by

$$
\mathcal{Z} = \int \mathcal{D}a \, \mathrm{e}^{-\frac{1}{2g^2} \mathrm{tr}|G|^2} \int \mathcal{D}\psi \, \mathrm{e}^{-\int \left( \mathrm{tr}[\psi^T (i\slashed{D} - m\gamma)\psi] + O(\psi^4) \right)}.
\tag{36}
$$

When we put $m = 0$, the Lagrangian is invariant under the discrete chiral symmetry $(\mathbb{Z}_2)_\chi$. Let us discuss whether the fermion path-integral measure is invariant under $(\mathbb{Z}_2)_\chi$, following Refs. [118, 119].

Let us first show that the non-zero spectrum, $\lambda \neq 0$, is manifestly invariant under the chiral transformation. Indeed, the discrete chiral transformation acts on the Grassmannian coefficient $\widetilde{\psi}$ as

$$
\widetilde{\psi}_{\lambda,1} \leftrightarrow \widetilde{\psi}_{-\lambda,2}, \; \widetilde{\psi}_{\lambda,2} \leftrightarrow \widetilde{\psi}_{-\lambda,1}.
\tag{37}
$$

We therefore find that, under the discrete chiral transformation,

$$
\mathrm{d}\widetilde{\psi}_{\lambda,1} \mathrm{d}\widetilde{\psi}_{-\lambda,1} \mathrm{d}\widetilde{\psi}_{\lambda,2} \mathrm{d}\widetilde{\psi}_{-\lambda,2} \mapsto \mathrm{d}\widetilde{\psi}_{-\lambda,2} \mathrm{d}\widetilde{\psi}_{\lambda,2} \mathrm{d}\widetilde{\psi}_{-\lambda,1} \mathrm{d}\widetilde{\psi}_{\lambda,1} = \mathrm{d}\widetilde{\psi}_{\lambda,1} \mathrm{d}\widetilde{\psi}_{-\lambda,1} \mathrm{d}\widetilde{\psi}_{\lambda,2} \mathrm{d}\widetilde{\psi}_{-\lambda,2},
\tag{38}
$$

so that the non-zero Dirac spectrum does not produce an anomaly. This is due to the Kramers-doubling feature of the non-zero Dirac spectrum. We assume in this formal discussion that we can take a UV regularization that does not violate this feature.

Next, we discuss the Dirac zero modes, $\widetilde{\psi}_{i_0,\pm}$. The discrete chiral transformation acts on them as

$$
\widetilde{\psi}_{i_0,+} \mapsto \widetilde{\psi}_{i_0,+}, \; \widetilde{\psi}_{i_0,-} \mapsto -\widetilde{\psi}_{i_0,-}.
\tag{39}
$$

Therefore, for each Dirac zero mode, the fermion measure gets a $(-1)$ sign,

$$
\mathrm{d}\widetilde{\psi}_{i_0,+} \mathrm{d}\widetilde{\psi}_{i_0,-} \mapsto (-1)\mathrm{d}\widetilde{\psi}_{i_0,+} \mathrm{d}\widetilde{\psi}_{i_0,-},
\tag{40}
$$

so that $\mathcal{D}\psi \mapsto (-1)^\zeta \mathcal{D}\psi$. Since $\zeta$ does not depend on continuous changes of $a_\mu$, we find the 't Hooft anomaly of Eq. (23). In the following subsections, we compute $\zeta$ under various twisted boundary conditions to find the explicit form of the 't Hooft anomalies of 2d adjoint QCD.

## 4.3 $(\mathbb{Z}_2)_F \times (\mathbb{Z}_2)_\chi$ and $\mathbb{Z}_N^{[1]} \times (\mathbb{Z}_2)_\chi$ anomalies

We now switch our focus to the computation of $\zeta$ on the two-torus $T^2$. Our goal is to understand the Hilbert space of the theory, and one can give a Hilbert space interpretation of the path integral on $T^2$ or $\mathbb{R} \times S^1$, but not on manifolds like $S^2$. Nevertheless, for completeness we present an evaluation of the mod 2 index on $S^2$ in Appendix B.

We would like to know how the value of $\zeta$ depends on background gauge fields for $(\mathbb{Z}_2)_F \times \mathbb{Z}_N^{[1]}$. A choice of background gauge fields for $(\mathbb{Z}_2)_F$ is just a choice of spin structure $s$, which on $T^2$ amounts to specifying whether the fermion is periodic (P) or anti-periodic (AP) around each cycle. The background gauge field associated to $\mathbb{Z}_N^{[1]}$ is a two-form field $B$, whose integrals on $M_2$ obey

$$\frac{N}{2\pi} \int_{M_2} B = k \in \mathbb{Z}, \tag{41}$$

and $k$ should be identified with $k + N$. The dependence of the mod 2 index $\zeta$ on $s$ and $B$ can be summarized as

**Theorem 2** *For $SU(N)$ adjoint QCD on the spacetime manifold is $T^2$, the mod 2 index is*

$$\zeta = \begin{cases} \zeta_{\text{free}}(s) + \dfrac{N}{2\pi} \displaystyle\int B, & (\text{even } N) \\ 0, & (\text{odd } N) \end{cases} \mod 2, \tag{42}$$

*where $\zeta_{\text{free}}(s)$ is the mod 2 index of a single free Majorana fermion with the spin structure $s$.*

If we set $\int B = 0$, Eq. (42) implies that there is a mixed 't Hooft anomaly between $(\mathbb{Z}_2)_F$ and $(\mathbb{Z}_2)_\chi$ when $N$ is even. In particular, if we choose a background gauge field for $(\mathbb{Z}_2)_F$ corresponding to periodic boundary conditions on both cycles of the torus, there is a robust fermion zero mode, and $\zeta = 1 \mod 2$.

- This mod 2 index theorem guarantees that the Euclidean partition function of 2d adjoint QCD on the torus with periodic boundary conditions on both cycles, $\tilde{\mathcal{Z}}$, has a robust fermion zero mode when $N$ is even. This implies that $\tilde{\mathcal{Z}}$ vanishes when $N$ is even. The Hilbert space interpretation of this fact is the presence of exact degeneracies between the energies of bosonic and fermionic states, which is discussed in detail in Sec. 8.

When $N$ is even and $\frac{N}{2\pi} \int B = 2p, p \in \mathbb{Z}$, the value of $\zeta$ does not depend on $B$. As a result, background gauge fields in the $\mathbb{Z}_{N/2}^{[1]}$ subgroup of $\mathbb{Z}_N^{[1]}$ do not contribute to the index.

- Physically, this means that the $\mathbb{Z}_{N/2}^{[1]}$ subgroup of center symmetry does not suffer from any 't Hooft anomalies.

This will have important consequences for confinement later in this paper.

To compute the value given in Eq. (42) for the mod 2 index, let the coordinates on the two-torus $T^2$ be $(x_1, x_2) \in \mathbb{R}^2$ with the identification $(x_1, x_2) \sim (x_1 + 1, x_2) \sim (x_1, x_2 + 1)$. Then the fields need to be identified up to gauge transformations. Let us introduce the transition functions $\Omega_1(x_2)$ for $(x_1 + 1, x_2) \sim (x_1, x_2)$, and $\Omega_2(x_1)$ for $(x_1, x_2 + 1) \sim (x_1, x_2)$. For gauge fields, the connection formula is given by

$$\begin{aligned} a_\mu(x_1 + 1, x_2) &= -i\Omega_1^\dagger(x_2) D_\mu \Omega_1(x_2) \\ &= \Omega_1^\dagger(x_2) a_\mu(x_1, x_2) \Omega_1(x_2) - i\Omega_1^\dagger(x_2) \partial_\mu \Omega_1(x_2), \\ a_\mu(x_1, x_2 + 1) &= -i\Omega_2^\dagger(x_1) D_\mu \Omega_2(x_1) \\ &= \Omega_2^\dagger(x_1) a_\mu(x_1, x_2) \Omega_2(x_1) - i\Omega_2^\dagger(x_1) \partial_\mu \Omega_2(x_1). \end{aligned} \tag{43}$$

For the Majorana fermions,

$$\psi(x_1+1,x_2) = (-1)^{n_1}\Omega_1^\dagger(x_2)\psi(x_1,x_2)\Omega_1(x_2),$$
$$\psi(x_1,x_2+1) = (-1)^{n_2}\Omega_2^\dagger(x_1)\psi(x_1,x_2)\Omega_2(x_1), \tag{44}$$

where $n_i$ are integers determining the spin structure $s$ on $T^2$. As already mentioned, varying $n_i$ corresponds to turning on background gauge fields for the $(\mathbb{Z}_2)_F$ symmetry, and in a Hilbert space picture it corresponds to choices of grading by the fermion parity $(-1)^F$ within the partition function.

We can relate $\psi(x_1+1,x_2+1)$ to $\psi(x_1,x_2)$ in two ways, and demanding that they be consistent leads to the condition

$$\Omega_1(x_2+1)\Omega_2(x_1) = \Omega_2(x_1+1)\Omega_1(x_2)\exp\left(-\frac{2\pi i}{N}k\right) \tag{45}$$

for some $k = 0, 1, \ldots, N-1$. The integer $k$ corresponds to the 't Hooft magnetic flux [64]. When considering an $SU(N)$ gauge theory, only field configurations with $k = 0$ are allowed. This corresponds to making fundamental representation Wilson loops well-defined gauge-invariant operators. If one were to consider an $SU(N)/\mathbb{Z}_N$ gauge theory, one would need to sum over all possible values of $k$ in the path integral, and then the fundamental Wilson loop is no longer a genuine line operator [6, 7, 65], because it is no longer gauge-invariant. Here we wish to consider $SU(N)$ gauge theory, but we want to understand the effect of turning on a non-trivial background gauge field for the $\mathbb{Z}_N^{[1]}$ one-form symmetry. This corresponds to computing the index $\zeta$ with some nonzero choice of $k$.

When $k = 0$, we can choose the trivial transition functions $\Omega_1 = \Omega_2 = \mathbf{1}_N$. Since $\zeta$ is a topological invariant mod 2, we can take $a_\mu = 0$ for the computation of $\zeta$. Therefore the index must be the same as that of $(N^2-1)$ free Majorana fermions:

$$\zeta = (N^2-1)\zeta_{\text{free}}(s) = \begin{cases} \zeta_{\text{free}}(s), & (\text{even } N) \\ 0, & (\text{odd } N) \end{cases} \mod 2. \tag{46}$$

Here, $\zeta_{\text{free}}(s)$ is the mod 2 index of a single Majorana fermion on the torus with spin structure $s$ and $\not{D} = \not{\partial}$. This matches Eq. (42) by setting $\int B = \frac{2\pi k}{N} = 0$. To be more explicit, let us enumerate the values of $\zeta_{\text{free}}$ on $T^2$:

(a) If $n_1 = n_2 = 0$, corresponding to periodic boundary condition in both directions, then the zero modes are constant spinors, and $\zeta_{\text{free}} = 1$.

(b) If either $n_1$ or $n_2$ are 1, then constant spinors are no longer zero modes, and indeed, there are no zero modes at all. Then $\zeta_{\text{free}} = 0$.

Importantly, case (b) implies that $\zeta = 0$ for certain spin structures, and that the discrete chiral symmetry in massless 2d adjoint QCD continues to be well-defined after quantization.

Next, we compute the index with nonzero 't Hooft flux $k > 0$. Then Eq. (45) is satisfied by the coordinate-independent transition functions

$$\Omega_1 = C, \ \Omega_2 = S^k, \tag{47}$$

where $C$ and $S$ are clock and shift matrices,

$$C = e^{(N+1)\pi i/N}\begin{pmatrix} 1 & 0 & \cdots & 0 \\ 0 & e^{2\pi i/N} & \cdots & 0 \\ \vdots & \vdots & \ddots & \vdots \\ 0 & 0 & \cdots & e^{2\pi i(N-1)/N} \end{pmatrix}, \quad S = e^{(N+1)\pi i/N}\begin{pmatrix} 0 & 1 & 0 & \cdots & 0 \\ 0 & 0 & 1 & \cdots & 0 \\ \vdots & \vdots & \vdots & \ddots & 0 \\ 0 & 0 & 0 & \cdots & 1 \\ 1 & 0 & 0 & \cdots & 0 \end{pmatrix}. \tag{48}$$

Since the index is a topological invariant, we can take the simplest setup for its computation: having satisfied the required commutation relation with coordinate-independent transition functions, we can again set $a_\mu = 0$. The Dirac operator again becomes $\slashed{D} = \slashed{\partial}$, and any Dirac zero modes must be constant spinors.

Since $C^N = S^N = e^{(N+1)\pi i} \mathbf{1}_N$, it follows that

$$\psi(x_1 + N, x_2) = (-1)^{n_1 N} \psi(x_1, x_2), \quad \psi(x_1, x_2 + N) = (-1)^{n_2 N} \psi(x_1, x_2). \tag{49}$$

The only way the Dirac operator can have zero modes is when $(-1)^{n_1 N} = (-1)^{n_2 N} = 1$. When $N$ is even, this is satisfied for any choice of spin structures, but when $N$ is odd, we must take $(-1)^{n_1} = (-1)^{n_2} = 1$ to have zero modes. If we decompose a candidate zero mode as $\psi = T\eta$, where $T \in \mathfrak{su}(N)$ and $\eta$ is a constant Majorana spinor with positive chirality, then $T$ must obey $CTC^{-1} = (-1)^{n_1} T$ and $S^k T S^{-k} = (-1)^{n_2} T$. So the number of such zero modes — namely, $\zeta$ — is given by

$$\zeta(s, k) = \dim\{T \in \mathfrak{su}(N) \,|\, CTC^{-1} = (-1)^{n_1} T, \; S^k T S^{-k} = (-1)^{n_2} T\} \bmod 2. \tag{50}$$

Making color indices explicit, $T = T_{ij}$ must satisfy

$$(CTC^{-1})_{ij} = e^{\frac{2\pi i}{N}(i-j)} T_{ij} = (-1)^{n_1} T_{ij}, \tag{51}$$

$$(S^k T S^{-k})_{ij} = T_{i+k, j+k} = (-1)^{n_2} T_{ij}. \tag{52}$$

We now work out $\zeta(s, k)$ separately for odd and even $N$.

When $N$ is odd, zero modes can only exist with periodic boundary conditions on both cycles. When $n_1 = 0, n_2 = 0$, one can prove that the number of linearly independent $\mathfrak{su}(N)$ matrices satisfying Eq. (51) and Eq. (52) is given by $\gcd(k, N) - 1$. When $N$ is odd $\gcd(k, N)$ is odd, so $\zeta = \gcd(k, N) - 1 \bmod 2 = 0$. Hence, there are no robust zero modes for odd $N$.

As an example, consider the specific case $N = 9$ and $k = 3$. Eq. (51) implies that $T$ is diagonal,

$$T = \mathrm{diag}(a_1, a_2, a_3, a_4, a_5, a_6, a_7, a_8, a_9), \tag{53}$$

with $a_i$ real so that $T$ is Hermitian. Eq. (52) equates $a_1 = a_4 = a_7$, $a_2 = a_5 = a_8$, and $a_3 = a_6 = a_9$. Demanding that $T$ be nonzero and traceless implies that it is specified by two real numbers. This matches the formula $\gcd(3, 9) - 1 = 2$, so there are two zero modes with positive chirality. As expected based on the general formula, $\zeta = 2 \bmod 2 = 0$.

When $N$ is even, it is possible to have zero modes for more spin structures and the index $\zeta$ can be non-zero. We first enumerate the zero modes when $k = 1$.

(a) When $n_1 = 0, n_2 = 0$, one can verify that $T_{ij} \propto \delta_{ij}$, and thus $\mathrm{tr}(T) \neq 0$ unless $T = 0$. This means that there are no zero modes, so $\zeta = 0$.

(b) When $n_1 = 0$, $n_2 = 1$, we obtain $T \propto \mathrm{diag}(+1, -1, \ldots, +1, -1)$, so $\zeta = 1$.

(c) When $n_1 = 1$, $n_2 = 0$, we have $T_{ij} \propto \delta_{i, j+N/2} + \delta_{j, i+N/2}$, so $\zeta = 1$.

(d) Finally, if $n_1 = n_2 = 1$, we find $T_{ij} \propto (-1)^i (\delta_{i, j+N/2} + \delta_{j, i+N/2})$ when $N/2$ is even and $T_{ij} \propto (-1)^i (i\delta_{i, j+N/2} - i\delta_{j, i+N/2})$ when $N/2$ is odd, and again $\zeta = 1$.

The calculation above generalizes to arbitrary $k$. To see how this works, we consider the specific case $N = 6$ and $k = 2$.

(a) When $n_1 = 0, n_2 = 0$, Eq. (51) implies that $T$ is diagonal, $T = \mathrm{diag}(a_1, a_2, a_3, a_4, a_5, a_6)$, where $a_i$ are real so that $T$ is Hermitian. Eq. (52) equates $a_{i+2} = a_i$, so we find $T = \mathrm{diag}(a_1, a_2, a_1, a_2, a_1, a_2)$. Finally, enforcing the traceless condition gives $a_2 = -a_1$. We therefore find one zero mode, $T \propto \mathrm{diag}(1, -1, 1, -1, 1, -1)$.

(b) When $n_1 = 0, n_2 = 1$, Eq. (51) and Hermiticity again imply $T = \text{diag}(a_1, a_2, a_3, a_4, a_5, a_6)$ with $a_i$ real. Eq. (52) now gives $a_{i+2} = -a_i$. Here the indices are understood mod 6, so we find $a_1 = -a_3 = a_5 = -a_1$ and $a_2 = -a_4 = a_6 = -a_2$. As a result $a_i = 0$ and there are no zero modes.

(c) Now we let $n_1 = 1, n_2 = 0$. Compared to the previous case we have simply relabeled the two cycles of the torus, and we should find a physically identical result. Enforcing Eq. (51) and Hermiticity, we find that $T$ has the form

$$
T = \begin{pmatrix} & & & a_1 & 0 & 0 \\ & & & 0 & a_2 & 0 \\ & & & 0 & 0 & a_3 \\ a_1^* & 0 & 0 & & & \\ 0 & a_2^* & 0 & & & \\ 0 & 0 & a_3^* & & & \end{pmatrix},
\tag{54}
$$

with $a_i \in \mathbb{C}$. Eq. (52) implies that $a_1 = a_3 = a_2^*$ and $a_2 = a_1^* = a_3^*$. Thus, the zero modes are characterized by the single complex number $a_1$. We can define two linearly independent zero modes by letting $a_1$ be purely real or purely imaginary, so there are two linearly-independent zero modes, and generic zero mode can be written as

$$
T = a \begin{pmatrix} & & & 1 & 0 & 0 \\ & & & 0 & 1 & 0 \\ & & & 0 & 0 & 1 \\ 1 & 0 & 0 & & & \\ 0 & 1 & 0 & & & \\ 0 & 0 & 1 & & & \end{pmatrix} + b \begin{pmatrix} & & & i & 0 & 0 \\ & & & 0 & -i & 0 \\ & & & 0 & 0 & i \\ -i & 0 & 0 & & & \\ 0 & i & 0 & & & \\ 0 & 0 & -i & & & \end{pmatrix},
\tag{55}
$$

where $a, b$ are real. Of course, P/AP and AP/P boundary conditions must be physically equivalent. But this implies that two zero modes (corresponding to AP/P boundary conditions) must be equivalent to none (corresponding to P/AP boundary conditions). So from this explicit computation we see that the number of zero modes of a given chirality is only well-defined mod 2, even without knowing the mod 2 index theorem.

(d) When $n_1 = 1, n_2 = 1$, Eq. (51) and Hermiticity again imply that $T$ has the form of Eq. (54). Eq. (52) gives $a_1 = -a_3 = a_2^* = -a_1$ and $a_2 = -a_1^* = a_3^* = -a_2$. As a result $a_i = 0$ and there are no zero modes.

The above calculations demonstrate a pattern which holds for arbitrary values of $k$ and $N$. To summarize, we find $\zeta = 0$ for odd $N$, while for even $N$ the result can be expressed as

$$
\zeta(s, k) = \zeta_{\text{free}}(s) + k \mod 2.
\tag{56}
$$

This confirms Eq. (42) upon identifying $\frac{N}{2\pi} \int B = k$.

## 4.4 $(\mathbb{Z}_2)_F \times (\mathbb{Z}_2)_C \times (\mathbb{Z}_2)_\chi$ anomaly

In this section we derive a mixed 't Hooft anomaly for $(\mathbb{Z}_2)_F \times (\mathbb{Z}_2)_C \times (\mathbb{Z}_2)_\chi$ by computing the mod 2 index $\zeta$ with background gauge fields for $(\mathbb{Z}_2)_F \times (\mathbb{Z}_2)_C$.

To compute the index, let us again set the $SU(N)$ gauge field $a = 0$, so that the zero modes of $i\slashed{D} = \slashed{\partial}$ become constant in spacetime. We work on $T^2$ with the following boundary conditions

$$
\begin{aligned}
\psi(x_1 + 1, x_2) &= \psi(x_1, x_2), \\
\psi(x_1, x_2 + 1) &= -U_C \psi(x_1, x_2).
\end{aligned}
\tag{57}
$$

Turning on $U_C$ in the boundary conditions corresponds to turning on a background gauge field for charge conjugation. If the $x_2$ direction is interpreted as Euclidean time, then the associated path integral computes a partition function with a grading by charge conjugation. Satisfying these boundary conditions with constant configurations requires that the color components of $\psi$ obey

$$\psi_{ij} = -\psi_{ji}. \tag{58}$$

This condition is satisfied by the $N(N-1)/2$ purely imaginary generators of $\mathfrak{su}(N)$. As a result, the mod 2 index takes the values

$$\zeta = \frac{N(N-1)}{2} \bmod 2 = \begin{cases} 0, & N = 4n, \\ 0, & N = 4n+1, \\ 1, & N = 4n+2, \\ 1, & N = 4n+3. \end{cases} \tag{59}$$

Before connecting this result to 't Hooft anomalies, let us consider the alternative choice of boundary conditions

$$\begin{aligned} \psi(x_1+1, x_2) &= \psi(x_1, x_2), \\ \psi(x_1, x_2+1) &= U_C \psi(x_1, x_2). \end{aligned} \tag{60}$$

Compared to the previous set-up, this amounts to changing the background gauge field for the $(\mathbb{Z}_2)_F$ symmetry. The associated path integral can be viewed as calculating a partition function graded by $(-1)^F$ along with charge conjugation. Then one can show that

$$\zeta = \frac{(N+2)(N-1)}{2} \bmod 2 = \begin{cases} 1, & N = 4n, \\ 0, & N = 4n+1, \\ 0, & N = 4n+2, \\ 1, & N = 4n+3. \end{cases} \tag{61}$$

For even $N$, comparing Eq. (59) and Eq. (61) shows that we can define a non-anomalous charge conjugation symmetry by working with $U_C$ for $N = 4n$ and $U_{C'} = U_F U_C$ for $N = 4n+2$. So there is no distinct 't Hooft anomaly involving charge conjugation when $N$ is even, and the only anomalies for the internal global symmetries were the ones we already discussed in the preceding subsection.

However, when $N$ is odd, such re-definitions of the charge-conjugation generator do not do anything. This means that when $N = 4n+3$, there is a mixed 't Hooft anomaly for $(\mathbb{Z}_2)_F \times (\mathbb{Z}_2)_C \times (\mathbb{Z}_2)_\chi$.

We have not found any 't Hooft anomalies for $N = 4n+1$. In particular, we have checked that there are no new anomalies involving all four discrete internal symmetries $\mathbb{Z}_N^{[1]}, (\mathbb{Z}_2)_C, (\mathbb{Z}_2)_F, (\mathbb{Z}_2)_\chi$.

## 5 Anomaly matching and low-energy behavior

In this section, we discuss the 't Hooft anomaly matching conditions for the anomalies we have uncovered. The idea of 't Hooft anomaly matching is by now textbook material: it says that an 't Hooft anomaly computed with UV degrees of freedom must be reproduced with the low-energy effective degrees of freedom. Anomaly matching implies that systems with 't Hooft anomalies cannot be trivially gapped, with a unique gapped ground state on any closed spatial manifold.

Since the low-energy description for a system with 't Hooft anomalies cannot be trivially gapped, the low-energy theory must involve some combination of

(a) intrinsic topological order,

(b) gapless excitations,

(c) spontaneous symmetry breaking.

We will now argue that in 2d adjoint QCD the 't Hooft anomalies must be matched by option (c), spontaneous symmetry breaking.

To rule out option (a), we note that in two dimensions, it is known that intrinsic topological order does not appear [16].

Option (b) is harder to exclude. Ref. [4] proposed a nice argument for ruling out a conformal fixed point, which we review below. The argument implicitly assumes that the four-fermi terms in Eq. (14) are fine-tuned to be irrelevant. Provided that this is done, the mass of any massive excitations must be set by $g\sqrt{N} = \lambda^{1/2}$, where $\lambda$ is the 't Hooft coupling, and one can take $\lambda \to \infty$ to focus on the massless states. The gauge kinetic term can be ignored when $\lambda \to \infty$, and then the gauge field $a_\mu$ becomes a pure Lagrange multiplier. Using non-Abelian bosonization [120], one obtains a Wess-Zumino-Witten model on the coset space $O(N^2-1)/\text{Ad}(SU(N))$. The central charge $c$ of the coset model is given by the formula [121]

$$c = c_{\text{UV}} - c_{\text{gauge}}, \tag{62}$$

where $c_{\text{UV}} = (N^2 - 1)/2$ is the central charge of the free Majorana fermion before gauging $SU(N)$, $c_{\text{gauge}} = k(N^2 - 1)/(k + N)$ describes how many degrees of freedom are removed by gauging $SU(N)$, and $k$ is the level of the current algebra. For the adjoint representation, $k = N$. This gives

$$c = \frac{N^2 - 1}{2} - \frac{N(N^2 - 1)}{N + N} = 0. \tag{63}$$

So the would-be CFT contains no gapless degrees of freedom. This conclusion is consistent with numerical results from discretized light-cone quantization of the theory [5,45]. For the theory with four-fermion interaction terms, however, this argument is not sufficient. Nevertheless, in this paper we will assume that gapless excitations do not exist. We believe that if a gapless phase is possible at all in 2d adjoint QCD, it could only appear as a result of fine-tuning $c_1$ and $c_2$, and would not be generic.

Given this discussion, any 't Hooft anomalies must be matched by spontaneous symmetry breaking, which is option (c) above. In our discussion below, we will assume the minimal scenario that the anomalies are matched by spontaneous symmetry breaking for all values of $N \geq 2$. Our explicit semi-classical analysis in Sec. 7 gives evidence that this minimal scenario is indeed realized, but it would be very interesting to study 2d adjoint QCD directly by e.g. numerical Monte Carlo lattice calculations.

## 5.1 Even $N$

For even $N$, Eq. (42) states that the theory has a mixed anomaly between $(\mathbb{Z}_2)_\chi$ and $(\mathbb{Z}_2)_F$, and also between $(\mathbb{Z}_2)_\chi$ and $\mathbb{Z}_N^{[1]}$. Fermion parity $(\mathbb{Z}_2)_F$ cannot be spontaneously broken if we assume a Lorentz-invariant vacuum, which is natural for 2d adjoint QCD. Assuming that the 't Hooft anomaly of $(\mathbb{Z}_2)_F \times (\mathbb{Z}_2)_\chi$ is matched through symmetry breaking then implies that

- discrete chiral symmetry $(\mathbb{Z}_2)_\chi$ is spontaneously broken, $(\mathbb{Z}_2)_\chi \to 1$, for even $N$.

This means that there are two degenerate vacua related by the discrete chiral symmetry. They are distinguished by the value of the order parameter

$$\langle \text{tr}[\psi^T i\gamma\psi] \rangle \sim \pm\Lambda, \tag{64}$$

where $\Lambda$ is a mass scale built from combinations of $\lambda$ and the non-perturbative strong scales associated to the $c_1$ and $c_2$ couplings when they are asymptotically free.

However, this is not the whole story. Discrete anomaly matching requires gapless excitations localized on the domain wall between these two vacua, see e.g. [8,99,122–124]. In order to explain this, let us turn on a small mass term $m \ll \lambda$, and compare the partition function with positive mass $m = +M$, $\mathcal{Z}_{m=+M}$, with the partition function with negative mass $m = -M$, $\mathcal{Z}_{m=-M}$. Since a chiral transformation flips the sign of the mass, these two partition functions are related by $(\mathbb{Z}_2)_\chi$. As we discussed in Sec. 4.2, under a chiral transformation the fermion measure $\mathcal{D}\psi$ picks up a phase $(-1)^\zeta$, where $\zeta$ is the mod 2 index. Hence,

$$\frac{\mathcal{Z}_{m=-M}}{\mathcal{Z}_{m=+M}} = (-1)^\zeta. \tag{65}$$

Let us define the system with positive $m$ to be a trivially gapped phase. Then, this result and Eq. (42) imply the following:

- For even $N$, 2d adjoint QCD with $m = -M$ is a nontrivial SPT phase (a "topological superconductor"), protected both by the fermion parity $(-1)^F$ and by the one-form center symmetry $\mathbb{Z}_N^{[1]}$.

Suppose that $m$ is positive for $x < 0$, and becomes negative for $x > 0$. Since the bulk is gapped, we expect the partition function to (approximately) factorize as

$$\mathcal{Z}[A] = \mathcal{Z}_{x<0,m=M}[A]\mathcal{Z}_{x=0}[A]\mathcal{Z}_{x>0,m=-M}[A], \tag{66}$$

where $A$ formally represents the background gauge fields for $[\mathbb{Z}_N^{[1]} \rtimes (\mathbb{Z}_2)_C] \times (\mathbb{Z}_2)_F$. Taking the normalization $\mathcal{Z}_{m=M} = 1$, Eq. (65) implies that

$$\mathcal{Z}[A] = \mathcal{Z}_{x=0}[A](-1)^{\zeta[A]|_{x>0}}. \tag{67}$$

The quantity $\zeta$ was a mod 2 topological invariant on closed oriented manifolds, but now it is no longer a topological invariant because of the boundary. The anomaly inflow mechanism [125] implies that there should be a gapless mode localized at $x = 0$. Eq. (42) for the mod 2 index tells us that this gapless excitation must actually be a $(0 + 1)$d Majorana fermion (that is, a real one-component Grassmann field) in a representation of $N$-ality $N/2$ under the $\mathbb{Z}_N^{[1]}$ center symmetry.

Let us demonstrate this explicitly by turning on a background gauge field for center symmetry. Using the two-form field $B$, we can write $(-1)^{\zeta|_{x>0}} = \exp(i\frac{N}{2}\int_{x>0} B)$ using Eq. (42). Under a 1-form gauge transformation $B \mapsto B+d\lambda$, the bulk partition function for $x > 0$ changes by $\exp(i\frac{N}{2}\int_{x>0} d\lambda) = \exp(i\frac{N}{2}\int_{x=0} \lambda)$ using Stokes' theorem. This gauge variation cannot be eliminated by any 1d local counterterm, so it must be compensated by a boundary excitation. The coefficient $N/2$ in the exponent means that the boundary excitation should have charge $N/2$. The fact that the boundary excitation must be a fermion can be understood using the Jackiw-Rebbi mechanism [126].

If we take the limit $M \to 0$, we can interpret the boundary of the SPT phase at $x = 0$ as a domain wall connecting the two chiral symmetry-breaking vacua. If we consider a long spatial region, it might have a large number $\mathcal{N}$ of such chiral symmetry-breaking domains. The fact that the domain walls host Majorana fermion modes implies that the number of approximate ground states $\mathcal{N}_{\text{g.s.}}$ in such a region scales exponentially in $\mathcal{N}$, $\mathcal{N}_{\text{g.s.}} \sim 2^{\mathcal{N}-1}$.

We now note that to make this story consistent, the string tension for charges of $N$-ality $N/2$ must vanish![13] If it did not, the ground-state energy density of a topologically nontrivial region would be larger than that of the topologically trivial region, due to the energetic

---

[13]Let us make some comments on this point from a more formal perspective. Usually, when a theory has a mixed

cost of separating the charges localized on the walls of the topologically non-trivial region. This would contradict the fact that chiral symmetry-breaking vacua must be degenerate. As a consequence, 't Hooft anomaly matching by discrete chiral symmetry breaking simultaneously requires

- spontaneous breaking of $\mathbb{Z}_N^{[1]}$ to $\mathbb{Z}_{N/2}^{[1]}$ for even $N$.

This amounts to the statement that test charges with $N$-ality $N/2$ are screened. It also has implications for the relations between the non-vanishing string tensions. Given a test charge with $N$-ality $q_1$ and another one with $N$-ality $q_2$, we can make a test charge of $N$-ality $q_1 + q_2$ mod $N$ by bringing the associated charges close together. The string tensions for test charges of $N$-ality $q$ and $N - q$ are related by charge conjugation, so they must be the same if charge conjugation is unbroken. The vanishing of the string tension for test charges of $N$-ality $N/2$ implies a string tension degeneracy:[14]

$$\sigma_{\frac{N}{2}} = 0 \implies \sigma_q = \sigma_{N-q} = \sigma_{q+\frac{N}{2}} = \sigma_{-q+\frac{N}{2}}. \tag{68}$$

Such sets can have either four or two distinct elements. More specifically:

- When $N = 4n + 2$, the string tensions fall into $n$ degenerate groups of 4 tensions.

- When $N = 4n$, the string tensions fall into $n - 1$ degenerate groups of 4 non-vanishing tensions, and one further group of 2 degenerate non-vanishing tensions.

We show a sketch of the dependence of the string tension on the $N$-ality that follows from this discussion in Fig. 2. It would be interesting to verify this behavior through lattice calculations.

We should emphasize that 't Hooft anomaly matching by discrete chiral symmetry breaking only requires screening for test charges with $N$-ality $N/2$. The general arguments of Sec. 3 concerning the impossibility of the spontaneous breaking of discrete one-form symmetries in 2d apply to the subgroup $\mathbb{Z}_{N/2}^{[1]}$, which does not participate in the 't Hooft anomaly. This is an important difference with scenarios proposed in earlier literature [34, 35].

## 5.2 Odd $N$

When $N$ is odd and satisfies $N = 4n + 3$, there is a $(\mathbb{Z}_2)_F \times (\mathbb{Z}_2)_C \times (\mathbb{Z}_2)_\chi$ 't Hooft anomaly. Since the 't Hooft anomaly does not involve center symmetry, and we are considering a two-dimensional theory, center symmetry cannot be spontaneously broken for the reasons explained in Sec. 3. Consequently, the fundamental string tension must be non-vanishing at any odd $N$, and we expect that the 't Hooft anomaly must be matched by spontaneous symmetry

---

anomaly between two symmetries $G_1$ and $G_2$, it is sufficient to break one of the symmetries down to the subgroup which does not produce the mixed anomaly. Therefore, our claim about partial deconfinement may seem to require unnecessary dynamics. But this is not the case. In $(1 + 1)$-dimensional spacetimes, putting a test particle at some point $x$ divides space into two disconnected regions. The vacua in those regions must satisfy different boundary conditions at $x$ due to the presence of the test particle. When those boundary conditions can be related by the broken 0-form symmetry, such a test particle is deconfined. This means that the mixed anomaly between 0-form and 1-form symmetries in $(1 + 1)$-dimensions is very special, as the 0-form symmetry generator is charged under the 1-form symmetry. The argument can be generalized to the mixed anomaly between 0-form and $(d - 1)$-form symmetries in $d$-dimensional spacetime manifolds (see Ref. [127] for more details).

When spacetime is a torus $T^2$, there is another subtlety about the deconfinement of non-contractible loops (Polyakov-type loops) because of the mixed anomaly between the chiral and fermion parity symmetries. This shall be discussed in Sec. 7.

[14] This unconventional predicted behavior for the string tension can be tested in numerical lattice simulations. It is difficult to implement exactly massless fermions on the lattice. However, for sufficiently light fermions, our results imply that $\sigma_{\frac{N}{2}}$ should be parametrically smaller than other tensions, $\sigma_{\frac{N}{2}} \sim m\Lambda$, where $\Lambda$ is a strong scale built out of some combination of $\lambda$ and strong scales associated to $c_1, c_2$.

## String tension for even N

Figure 2: A sketch of the string tension in 2d $SU(N)$ adjoint QCD for even $N$ as a function of $N$-ality $k$. We draw a smooth curve, but of course for finite $N$ the spectrum of string tensions is discrete. The string tension must vanish at $k = N/2$ to satisfy 't Hooft anomaly matching. The sketch also shows the (generically) four-fold degeneracy of the string tension predicted by our discussion. Note that this double bump structure is required only for even $N$. It is not required for odd $N$.

breaking. If we assume that Lorentz invariance is unbroken, then $(\mathbb{Z}_2)_F$ cannot be spontaneously broken. So at least one of the $(\mathbb{Z}_2)_C$ and $(\mathbb{Z}_2)_\chi$ symmetries must be spontaneously broken. Semiclassical analysis on $\mathbb{R} \times S^1$, which we describe in the following section, is consistent with spontaneous breaking of $(\mathbb{Z}_2)_\chi$, and we will assume that this remains true on $\mathbb{R}^2$ in what follows.[15] This discussion can be summarized as

- 2d adjoint QCD with $N = 4n + 3$ confines test charges of all non-trivial $N$-alities, and spontaneously breaks chiral symmetry.

Arguments which are identical to those in Sec. 5.1 imply that domain walls separating chiral symmetry-breaking vacua must host gapless Majorana modes. In contrast to the story with even $N$, these domain-wall Majorana fermions are neutral under the $SU(N)$ gauge group. If we turn on a small fermion mass $m$ and let it vary in space, then

- 2d adjoint QCD with $N = 4n + 3$ and $m < 0$ is a nontrivial SPT phase, protected by the combination of fermion parity and charge conjugation symmetries.

Finally, let us consider the case $N = 4n + 1$, where there are no 't Hooft anomalies. In this case we may utilize the results of Fidkowski and Kitaev, who showed that systems of $8k, k \in \mathbb{N}$ Majorana fermions can be connected to a trivially gapped phase by symmetry-preserving interactions [15]. Adjoint QCD with $N = 4n + 1$ describes an interacting system of $8n(2n + 1)$ Majorana fermions. As a result, such a system must generically have a trivial gapped ground state. Of course, given that 2d adjoint QCD has some dimensionless parameters, it is possible that somewhere in its parameter space chiral symmetry is spontaneously broken, because it is a zero-form symmetry, and hence can in principle break in two spacetime dimensions. But such spontaneous chiral symmetry breaking should not be a generic property of the theory with $N = 4n + 1$. Center symmetry, meanwhile, is a one-form symmetry, and cannot break in two

---

[15]It may be tempting to try to use an argument due to Vafa and Witten [128] concerning the possibility of spontaneous parity breaking in 4d gauge theory to rule out $(\mathbb{Z}_2)_C$ breaking in our 2d gauge theory context. We can indeed prove the corresponding diamagnetic inequality. Unfortunately, Ref. [128] had some gaps [129–131] that have been repaired in the context of 4d gauge theory [132], but they have not yet been addressed in the present context.

spacetime dimensions without being forced to do so by an appropriate 't Hooft anomaly. So center symmetry cannot be be spontaneously broken on $\mathbb{R}^2$ when $N = 4n + 1$. This discussion can be summarized as

- 2d adjoint QCD with $N = 4n + 1$ confines test charges in all representations. There are no anomalies forcing chiral symmetry to break spontaneously. So whether or not chiral symmetry is spontaneously broken can depend on the region in parameter space that one considers.

# 6 Classification of fermionic SPT phases in one spatial dimension

As we mentioned earlier, Fidkowski and Kitaev pointed out that interactions change the $\mathbb{Z}$ classification of free-fermion SPT phases of matter in one spatial dimension into a $\mathbb{Z}_8$ classification [14, 15]. In the continuum language of this paper, they showed that systems of $8k, k \in \mathbb{N}$ Majorana fermions with interactions that preserve $(\mathbb{Z}_2)_F$ and $(\mathbb{Z}_2)_\chi$ symmetries can lie in a trivial gapped phase. For more on the classification of SPT phases realizable in fermionic systems in one spatial dimension, see e.g. [11–18]. In Refs. [17, 133, 134] it was suggested that the mathematical classification of SPT phases should be in terms of certain cobordism groups. This proposal was proved in Refs. [135, 136] in the context of relativistic topological field theories.

Our results imply that 2d adjoint QCD with $m < 0$ is an SPT state when $N = 4n, 4n+2, 4n+3$. In this section, we would like to discuss how this conclusion fits into the current knowledge of SPT phases. One can think of 2d $SU(N)$ adjoint QCD as a system of $N^2 - 1$ interacting Majorana fermions. When $N$ is even, the number of Majorana fermions is odd, so the $\mathbb{Z}_8$ classification by Fidkowski and Kitaev [14, 15] already tells us that it should realize a nontrivial SPT state. However, the quantity characterizing the relevant mixed 't Hooft anomaly, as given by $\zeta$ in Eq. (42), contains an extra contribution compared to a generic system of an odd number of Majorana fermions: the $\frac{N}{2} \int B$ term. This extra term is due to the fact that the system we consider has some extra symmetries compared to the generic case, and one of them, the one-form $\mathbb{Z}_N$ center symmetry, is involved in the 't Hooft anomaly at even $N$. This leads to a number of important consequences related to confinement/deconfinement, as discussed in Sec. 5.1.

One might find the $N = 4n + 3$ case more mysterious, because in this case the number of Majorana fermions is a multiple of 8; $N^2 - 1 = 8(n + 1)(2n + 1)$. A triple mixed 't Hooft anomaly involving $(\mathbb{Z}_2)_F \times (\mathbb{Z}_2)_\chi \times (\mathbb{Z}_2)_C$ at $m = 0$ indicates that 2d $SU(4n + 3)$ adjoint QCD with $m < 0$ is a nontrivial SPT state, although it contains multiples of 8 Majorana fermions. How is this result consistent with Refs. [14, 15]? The point is that we have more symmetries than Refs. [14, 15], and now it is the charge conjugation symmetry which plays the important role. Therefore, our results are not in conflict with Ref. [14, 15], which considered interactions that broke $(\mathbb{Z}_2)_C$.

Indeed, in 2d adjoint QCD we could turn on the four-Fermi operator in Eq. (15), repeated here for convenience

$$S_{\text{C-breaking}} = \int \mathrm{d}^2 x \, \frac{c_3}{N} \operatorname{tr}(\psi_+ \psi_- \psi_+ \psi_-). \tag{69}$$

This breaks $(\mathbb{Z}_2)_C$, and eliminates all 't Hooft anomalies for odd $N$. If the sign of $c_3$ is chosen appropriately, this operator will be marginally relevant, and the theory should have a trivial gapped phase for all odd $N$, including $N = 4n + 3$, consistent with the results of Fidkowski and Kitaev [15]. This can be seen explicitly in a semiclassical analysis on small $\mathbb{R} \times S^1$ in the next section.

From the perspective of Refs. [17, 133–136], the Fidkowski-Kitaev $\mathbb{Z}_8$ classification of SPT phases without any symmetries other than time reversal and fermion parity corresponds to a classification of SPT phases by the 2d pin$^-$ cobordism $\Omega_2^{\text{pin}^-}$. Here, we are looking at theories with more global symmetries. Following Refs. [17, 133–136], we can see that the SPT phases realized by 2d adjoint QCD with $m < 0$ should correspond to nontrivial elements of $\Omega_2^{\text{pin}^-}(\mathcal{B}^2\mathbb{Z}_N)$ for $N = 2n$, and $\Omega_2^{\text{pin}^-}(\mathcal{B}^1\mathbb{Z}_2)$ for $N = 4n+3$. Here $\mathcal{B}^k\mathbb{Z}_\ell$ represents an Eilenberg-MacLane space, $\ell$ is the order of the relevant cyclic symmetry group, which is either $\mathbb{Z}_2$ or $\mathbb{Z}_N$ for us, and $k$ represents the degree of background gauge fields associated with each symmetry. Zero-form symmetries are associated with one-form gauge fields, while one-form symmetries are associated with two-form gauge fields.

Our results can be generalized. In our gauge theory context with $N > 2$, $(\mathbb{Z}_2)_C$ is the only symmetry we can impose on the fermions other than $(\mathbb{Z}_2)_F$ and $(\mathbb{Z}_2)_\chi$, because the $\mathfrak{su}(N)$ algebra only has one outer automorphism, namely charge conjugation. But suppose instead we were to consider a system of 8 Majorana fermions $\psi_a, a = 1, 2, \ldots, 8$. (The construction we will now describe generalizes in a simple way to $8k$ Majorana fermions.) Suppose that the interactions preserve $(\mathbb{Z}_2)_F$, $(\mathbb{Z}_2)_\chi$, and also a symmetry $(\mathbb{Z}_2)_O$ which acts by flipping the sign of one of fermions, say $\psi_1 \to -\psi_1$. This transformation is related to one of the generators of the group $O(8)$ under which the fermions transform in the vector representation, but we do not assume that $O(8)$ is a symmetry, except for its $(\mathbb{Z}_2)_O$ part. The results of this paper imply that there is a mixed 't Hooft anomaly involving $(\mathbb{Z}_2)_F \times (\mathbb{Z}_2)_\chi \times (\mathbb{Z}_2)_O$. Consequently, systems of $8k$ interacting Majorana fermions can realize non-trivial SPT phases of matter if they enjoy symmetries like $(\mathbb{Z}_2)_O$.

# 7 Dynamics of adjoint QCD on $\mathbb{R} \times S^1$

We now explore how the general constraints discussed in the previous section are satisfied in a calculable region of the parameter space of 2d adjoint QCD. This calculable domain appears if the theory is formulated on $\mathbb{R} \times S^1$. Let us regard $\mathbb{R}$ as the Euclidean time direction, and take $L$ to be the circumference of $S^1$. Then if $\lambda L^2 \ll 1$ the theory becomes weakly coupled, and can be described by a 1d effective field theory — quantum mechanics. Our goal is to understand how the global symmetries are realized in this calculable regime.

From the point of view of the 1d effective field theory, the one-form symmetry $\mathbb{Z}_N^{[1]}$ acts like an ordinary $\mathbb{Z}_N$ symmetry: it acts on Polyakov loops by $\mathbb{Z}_N$ phase rotations. We will denote the operator generating center symmetry as $\widehat{U}_S$, which obeys $\widehat{U}_S^N = 1$. The other global symmetries, fermion parity $(\mathbb{Z}_2)_F$, charge conjugation $(\mathbb{Z}_2)_C$, and the discrete chiral symmetry $(\mathbb{Z}_2)_\chi$, are generated by operators $\widehat{U}_F$, $\widehat{U}_C$, and $\widehat{U}_\chi$, respectively, which obey $\widehat{U}_F^2 = \widehat{U}_C^2 = \widehat{U}_\chi^2 = 1$. The symmetry group of the 1d effective field theory is

$$G_{\text{QM}} = \begin{cases} \mathbb{Z}_N \times (\mathbb{Z}_2)_F \times (\mathbb{Z}_2)_\chi & N = 2 \\ [\mathbb{Z}_N \rtimes (\mathbb{Z}_2)_C] \times (\mathbb{Z}_2)_F \times (\mathbb{Z}_2)_\chi & N > 2. \end{cases} \tag{70}$$

The classical commutation relations are

$$\widehat{U}_S \widehat{U}_C = \widehat{U}_C \widehat{U}_S^{-1}, \quad \widehat{U}_S \widehat{U}_F = \widehat{U}_F \widehat{U}_S, \quad \widehat{U}_F \widehat{U}_C = \widehat{U}_C \widehat{U}_F, \tag{71}$$

for the vector-like symmetries, while the commutation relations involving chiral symmetry are

$$\text{(classical)} \quad \begin{cases} \widehat{U}_S \widehat{U}_\chi = \widehat{U}_\chi \widehat{U}_S, \\ \widehat{U}_F \widehat{U}_\chi = \widehat{U}_\chi \widehat{U}_F, \\ \widehat{U}_C \widehat{U}_\chi = \widehat{U}_\chi \widehat{U}_C. \end{cases} \tag{72}$$

The existence of the 't Hooft anomaly (42) means that the symmetry group acts projectively on the Hilbert space. Since the vector-like symmetries do not participate in the anomalies by themselves, the 't Hooft anomaly can only modify the part of the algebra in Eq. (72) which involves the discrete chiral symmetry.

The form of the modification of the operator commutation relations depends on $N$ as well as the choice of fermion boundary condition on $S^1$. To explain why boundary conditions enter the story, we recall that anomalies involving one-form symmetries always persist under compactification in the sense that an anomaly in a $d$-dimensional theory descends to an anomaly in a $d-1$ dimensional theory obtained by dimensional reduction on a circle [8]. But whether anomalies for zero-form symmetries persist under dimensional reduction in general depends on the choice of boundary conditions, see e.g. [103–105]. Here we will consider periodic and anti-periodic boundary conditions for the fermions. Let us summarize our results before explaining their derivation.

For anti-periodic (AP) boundary conditions, we find that Eq. (72) is modified to

$$
\text{(quantum, AP)} \quad
\begin{cases}
\widehat{U}_{\mathsf{S}} \widehat{U}_\chi = (-1)^{N-1} \widehat{U}_\chi \widehat{U}_{\mathsf{S}}, \\
\widehat{U}_{\mathsf{F}} \widehat{U}_\chi = \widehat{U}_\chi \widehat{U}_{\mathsf{F}}, \\
\widehat{U}_{\mathsf{C}} \widehat{U}_\chi = \widehat{U}_\chi \widehat{U}_{\mathsf{C}}.
\end{cases}
\tag{73}
$$

This means that the $(\mathbb{Z}_2)_\chi \times \mathbb{Z}_N^{[1]}$ anomaly for even $N$ survives dimensional reduction with these boundary conditions, but the $(\mathbb{Z}_2)_F \times (\mathbb{Z}_2)_\chi$ anomaly and $(\mathbb{Z}_2)_F \times (\mathbb{Z}_2)_\chi \times (\mathbb{Z}_2)_C$ anomalies do not. This means that the energy eigenstates must be doubly degenerate for even $N$, but for all odd $N$ there can be no robust spectral degeneracies. Some aspects of the projective nature of symmetry group for this $S^1$-compactified theory with anti-periodic boundary condition were studied for $N = 2$ and $N = 3$ in the limit $gL \ll 1$ by Lenz, Shifman, and Thies in Ref. [31]. Our work explains the 2-dimensional origin of their results from a discrete 't Hooft anomaly in the 2d theory on $\mathbb{R}^2$, and generalizes the story to general $N$.

For periodic (P) boundary conditions, we find that all of the 't Hooft anomalies survive the reduction to quantum mechanics, and Eq. (72) is modified to

$$
\text{(quantum, P)} \quad
\begin{cases}
\widehat{U}_{\mathsf{S}} \widehat{U}_\chi = (-1)^{N-1} \widehat{U}_\chi \widehat{U}_{\mathsf{S}}, \\
\widehat{U}_{\mathsf{F}} \widehat{U}_\chi = (-1)^{N-1} \widehat{U}_\chi \widehat{U}_{\mathsf{F}}, \\
\widehat{U}_{\mathsf{C}} \widehat{U}_\chi = (-1)^{\frac{(N-2)(N-1)}{2}} \widehat{U}_\chi \widehat{U}_{\mathsf{C}}.
\end{cases}
\tag{74}
$$

So if $N = 4n, 4n + 2, 4n + 3$, the eigenstates must be at least doubly-degenerate, and these degeneracies can be interpreted as a signal of the spontaneous breaking of the appropriate global symmetries. On the other hand, when $N = 4n + 1$ there can be no robust spectral degeneracies, and the global symmetries are not spontaneously broken.

The fact that all of the constraints due to the 't Hooft anomalies persist on $\mathbb{R} \times S^1$ with periodic boundary conditions means that compactification on small circles with these boundary conditions provides a particularly clean way to study the physics of 2d adjoint QCD in a controlled semi-classical setting.

## 7.1 Anti-periodic boundary conditions

In this section we take the fermions to have anti-periodic boundary conditions. The first step in our analysis is to determine the realization of center symmetry. This can be done by computing a Gross-Pisarski-Yaffe (GPY) effective potential for $\langle \mathrm{tr} P \rangle$ [137]. There is no tree level potential for the holonomy. Assuming that $L\sqrt{\lambda} \ll 1$, the physics at the scale of the circle is weakly coupled, and then a one-loop treatment of the effective potential is valid. At one loop, the

effective potential on $\mathbb{R} \times S^1$ is

$$V_{\text{eff,AP}}(P) = -\frac{1}{2LV} \ln \det(-D_{\text{adj}}^2) = \frac{1}{\pi L^2} \sum_{n=1}^{\infty} \frac{(-1)^n}{n^2} |\text{tr}(P^n)|^2 . \tag{75}$$

Note that in 2d gauge theory gluons do not propagate any physical degrees of freedom, and so do not contribute to $V_{\text{eff}}(P)$. For example, in a gauge where $P$ is diagonal, their contribution is completely canceled by Faddeev-Popov ghosts.

To get a feeling for the minima of this potential, consider the case $N = 2$, and pick a gauge where $P = \text{diag}(e^{i\alpha}, e^{-i\alpha})$. Then

$$V_{\text{eff,AP}}(P) = \frac{1}{2\pi L^2} \sum_{n=1}^{\infty} \frac{(-1)^n}{n^2} (1 + \cos 2n\alpha) = \frac{1}{2\pi L^2} \min_{k \in \mathbb{Z}} \left[ (\alpha + \pi k)^2 - \frac{\pi^2}{6} \right]. \tag{76}$$

There are two minima corresponding to $\alpha = 0, \pi$ within the unit cell $[0, \pi]$ (The unit cell is determined by the action of the Weyl group on $\alpha$). These center-breaking minima correspond to $P = \pm \mathbf{1}$.

For generic $N$, the potential has $N$ distinct minima $P = e^{2\pi i k/N} \mathbf{1}$. These minima are related by center symmetry. If the vacuum were to be frozen into one of these minima, center symmetry would be spontaneously broken. However, there are finite-action tunneling events connecting the center-breaking minima.[16] Let us study these tunneling events from two perspectives:

(1) Tunneling from the perspective of the 2d gauge theory.

(2) Tunneling from the perspective of the 1d EFT,

Let us start with the first approach. Call the states associated with the $N$ minima $|0\rangle, |1\rangle, \ldots, |N-1\rangle$. They are characterized by the phase of Polyakov loop as

$$\frac{1}{N} \text{tr}[P] |\ell\rangle = \omega^\ell |\ell\rangle, \tag{77}$$

where $\omega = e^{2\pi i/N}$. The matrix element associated with the tunneling process $\ell \to \ell + 1$ can be written as

$$\langle \ell + 1 | \exp(-\beta \widehat{H}_{\text{eff}}) | \ell \rangle \simeq \frac{1}{N} \text{tr} \left[ \exp(-\beta \widehat{H}_{\text{eff}}) \widehat{U}_{\text{S}} \right]. \tag{78}$$

The right-hand-side is nothing but the path integral with anti-periodic boundary condition for fermions and the minimal non-trivial 't Hooft flux $\int B = 2\pi/N$. This means that we must consider the implications of our mod 2 index theorem. The index theorem discussed in Sec. 4.3 says that $\zeta = 1$ mod 2 for even $N$ and $\zeta = 0$ mod 2 for odd $N$, so that

$$\zeta = N - 1 \text{ mod } 2. \tag{79}$$

We note that previous works [31, 34] have studied the fermionic zero modes associated with such instantons within an Abelian ansatz, and found $2(N-1)$ fermionic zero modes, half with positive chirality and half with negative chirality. But since the index theorem associated to 2d adjoint QCD is defined mod 2, most of these putative zero modes can be — and hence will be — lifted in generic gauge field backgrounds. We note that the explicit computation of $\zeta$ in Sec. 4.3 gives an example of such a background: it is given by the gauge configuration with

---

[16]The GPY potential for 2d adjoint QCD with anti-periodic boundary conditions was calculated in Ref. [4], but the role of non-perturbative tunneling events in the restoration of center symmetry in 2d gauge theory was not discussed there. Tunneling events between center-breaking vacua were later considered in Ref. [31] for $N = 2$ and $N = 3$.

minimal 't Hooft flux. The discussion in Sec. 4.3 shows that this background is associated with precisely one pair of zero modes for any even $N$, and no zero modes for any odd $N$.

The existence of $(N-1)$ zero modes in the Abelian ansatz implies that the $(\mathbb{Z}_2)_\chi$ charges of $|\ell\rangle$ and $|\ell+1\rangle$ are different by $(-1)^\zeta = (-1)^{N-1}$. We therefore find

$$\widehat{U}_S|\ell\rangle = |\ell+1\rangle, \quad \widehat{U}_\chi|\ell\rangle = (-1)^{(N-1)\ell}|\ell\rangle, \tag{80}$$

while $\widehat{U}_F = 1$ on these states. These rules are consistent with the commutation relations,

$$\widehat{U}_S\widehat{U}_\chi = \begin{cases} \widehat{U}_\chi\widehat{U}_S, & \text{odd } N, \\ (-1)\widehat{U}_\chi\widehat{U}_S, & \text{even } N, \end{cases} \tag{81}$$

which matches the requirements of the 't Hooft anomaly (42). Since $\widehat{U}_C\widehat{U}_S\widehat{U}_C^{-1} = \widehat{U}_S^{-1}$ implies that $\widehat{U}_C|\ell\rangle = |-\ell\rangle$, we find that the algebra for $\widehat{U}_C$ is unmodified from the classical one. This confirms (73).

We can now see the implications for the restoration of center symmetry by tunneling. Tunneling between $|\ell\rangle$ and $|\ell+1\rangle$ is forbidden when $N$ is even because the associated instantons carry robust fermion zero modes due to the mod 2 index theorem. But when $N$ is odd, these instantons do not have robust fermion zero modes, so tunneling from $|\ell\rangle$ to $|\ell+1\rangle$ is not forbidden. On the other hand, tunneling from $|\ell\rangle$ to $|\ell+2\rangle$ is associated with $\zeta = 0$, and so the relevant instantons do not have robust fermionic zero modes for both even and odd $N$. Therefore tunneling from $|\ell\rangle \to |\ell+2\rangle$ is not forbidden for any $N$. The minimal allowed (and disallowed) tunneling events are depicted in Fig. 3 for $N = 5, 6$.

Let us write the tunneling amplitudes more explicitly within the quantum-mechanical effective field theory. If one writes the eigenvalues of the Polyakov loop as $e^{i\theta_a}, a = 1, \dots N$, one can show that the fundamental domain of vectors $\theta_a$ obtained by considering the action of the Weyl group $S_N$ on $\theta_a$ is an $(N-1)$-simplex.[17] The $N$ corners of the fundamental domain are the $N$ center-breaking minima of the potential. It can be shown that the minimal-action tunneling events occur within the edges of the fundamental domain. To make things concrete, let us take an ansatz where $\theta_i = \alpha(t)$ for $i = 1, \cdots, N-1$ and $\theta_N = -(N-1)\alpha(t)$. The effective action for the quantum mechanics on $\mathbb{R} \times S_L^1$ becomes

$$\begin{aligned} S_{1D} &= \int dt \left[ \frac{1}{2g^2L} \sum_{i=1}^{N} \dot{\theta}_i^2 + \frac{1}{\pi L} \sum_{n=1}^{\infty} \frac{(-1)^n}{n^2} \left( 2(N-1)\cos N\alpha n + (N-1)^2 + 1 \right) \right] \\ &= \int dt \left[ \frac{N(N-1)}{2g^2L} \dot{\alpha}^2 + \frac{N^2(N-1)}{2\pi L} \min_{k\in\mathbb{Z}} \left( \alpha + \frac{2\pi k}{N} \right)^2 - \frac{\pi^2}{12\pi} \frac{N^2}{L} \right]. \end{aligned} \tag{83}$$

Note that the potential has a cusp at $\alpha = \pi/2$. The equation of motion for $\alpha$ is

$$\ddot{\alpha} = \frac{\omega^2}{2} \frac{d}{d\alpha} \left\{ \min_{k\in\mathbb{Z}} \left( \alpha + \frac{2\pi k}{N} \right)^2 \right\}, \tag{84}$$

where $\omega^2 = \lambda/\pi$. Suppose that the Euclidean time extent is finite and has extent $\beta$. Then the field configuration associated to the basic instanton is

$$\alpha(t) = \begin{cases} \dfrac{\pi}{N} \dfrac{\sinh(\omega t)}{\sinh(\omega\beta/2)} & (0 \le t < \beta/2), \\ \dfrac{2\pi}{N} + \dfrac{\pi}{N} \dfrac{\sinh(\omega(t-\beta))}{\sinh(\omega\beta/2)} & (\beta/2 \le t \le \beta). \end{cases} \tag{85}$$

---

[17]If one takes a diagonal (Polyakov) gauge for the holonomy, the Weyl group $S_N$ is a remnant discrete gauge transformation. It acts on the fields by

$$\theta_i \mapsto \theta_{\sigma(i)}, \ \psi_{ij} \mapsto \psi_{\sigma(i)\sigma(j)}. \tag{82}$$

where $\sigma \in S_N$. Of course, physical states have to be invariant under this remnant gauge transformation.

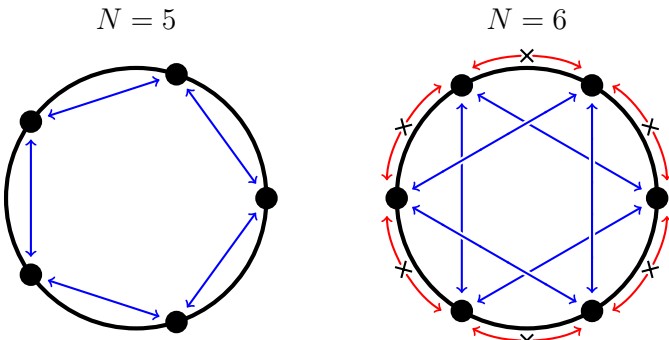

Figure 3: Examples of tunneling events between center-breaking minima on $\mathbb{R} \times S^1$ with anti-periodic boundary conditions. The blue lines depict the minimal action tunneling events without fermion zero modes. The red lines (marked with crosses) represent tunneling events with robust fermion zero modes. When $N = 5$, the $5 - 1 = 4$ zero modes found in the Abelian ansatz are all lifted in generic field configurations because $\zeta = 4 \bmod 2 = 0$. As a result, the $\mathbb{Z}_5$ center symmetry is completely unbroken. When $N = 6$, instantons associated with tunneling between neighboring minima carry $\zeta = 6 - 1 \bmod 2 = 1$ robust fermion zero modes, and are forbidden. However, instantons associated with tunneling between next-to-nearest neighbors carry $\zeta = 2(6 - 1) \bmod 2 = 0$ robust zero modes, and are not forbidden. As a result, the $\mathbb{Z}_3$ subgroup of center symmetry remains unbroken.

In the large $\beta$ limit, the action of the instanton is

$$S_I = (N-1)\sqrt{\frac{\pi^3}{L^2 \lambda}}. \tag{86}$$

We pause to again emphasize the necessity to count only robust fermion zero modes, while clarifying a potential point of confusion. As noted above, the minimal-action tunneling configuration can be described within an Abelian ansatz. Working within such an ansatz, it is known that there are $N - 1$ pairs of positive and negative chirality fermionic zero modes [31, 34]. But this does not mean that all of these fermionic zero modes are robust enough to block tunneling processes. In the full path integral one has to sum over *all* configurations with the appropriate boundary conditions, and in particular it is not enough to take into account only the specific configuration in Eq. (85). The key point is that one must count fermionic zero modes in generic field configurations with appropriate boundary conditions, because that is what matters for understanding whether a given tunneling process between bosonic vacua is allowed or not. This can be done using the mod 2 index theorem reviewed in Section 4.1. The fact that this index theorem works in mod 2 shows that most of the would-be fermionic zero modes are lifted by non-Abelian interactions.

### 7.1.1 Hilbert space structure

It is interesting to work out the Hilbert space structure of adjoint QCD on $\mathbb{R} \times S^1_L$ with anti-periodic boundary conditions. Suppose that we write the Hamiltonian as

$$\widehat{H}_{\text{eff}} = \widehat{H}_0 + \Delta\widehat{H}, \tag{87}$$

where $\widehat{H}_0$ corresponds to the Hamiltonian associated to the leading-order Lagrangian (83), while $\Delta\widehat{H}$ comes from higher-order corrections. What are the constraints on $\langle \ell_1 | \Delta\widehat{H} | \ell_2 \rangle$, for $\ell_1 \neq \ell_2$? These off-diagonal terms induces a splitting of the energies of the degenerate eigenstates of $\widehat{H}_0$ by an amount $\sim \Delta E$.

The instanton calculation described above already shows that $\Delta E$ is very small in the semi-classical regime, $\Delta E \sim e^{-(N-1)\pi^{3/2}/(L\lambda^{1/2})}$. So we can approximate

$$\exp(-\beta \widehat{H}_{\text{eff}}) \simeq 1 - \beta \Delta \widehat{H}, \tag{88}$$

if $(g^2 L)^{-1} \ll \beta \ll \Delta E^{-1}$. In that range of $\beta$,

$$\langle \ell_1 | \Delta \widehat{H} | \ell_2 \rangle \simeq -\frac{1}{N\beta} \text{tr} \left[ \exp(-\beta \Delta \widehat{H}) \widehat{U}_S^{(\ell_1-\ell_2)} \right]. \tag{89}$$

The mod 2 index theorem, or equivalently the symmetry algebra, implies that if $\zeta = (N-1)(\ell_1 - \ell_2) = 1 \mod 2$, then

$$\langle \ell_1 | \Delta \widehat{H} | \ell_2 \rangle = 0. \tag{90}$$

In general, the evaluation of the non-vanishing matrix elements of $\Delta H$ beyond leading order requires some non-trivial calculations. For example, in general one would need to work out the details of how the $2(N-1)(\ell_1-\ell_2)$ apparent fermionic zero modes seen in an Abelian ansatz are lifted modulo 4 by e.g. four-fermion couplings in $\Delta \widehat{H}$. To illustrate our point, we will instead just write down a simple model $N \times N$ Hamiltonian satisfying the symmetry constraints:

$$\langle \ell_1 | \Delta \widehat{H} | \ell_2 \rangle = \begin{cases} -\Delta E(\delta_{\ell_1, \ell_2+1} + \delta_{\ell_1+1, \ell_2}), & \text{odd } N, \\ -\Delta E(\delta_{\ell_1, \ell_2+2} + \delta_{\ell_1+2, \ell_2}), & \text{even } N, \end{cases} \tag{91}$$

with some $\Delta E > 0$. Essentially, this makes $\Delta H$ a tridiagonal Toeplitz matrix, and such matrices are easy to diagonalize.

**Odd $N$**

For odd $N$, the energy eigenstates, $|E, n\rangle$, with $n = 0, 1, \ldots, N-1$, are given by

$$|E, n\rangle = \sum_{\ell=0}^{N-1} \omega^{n\ell} |\ell\rangle, \tag{92}$$

with the eigenenergy

$$E_n = -2\Delta E \cos\left(\frac{2\pi n}{N}\right). \tag{93}$$

The ground state $|E, 0\rangle$ is unique, while the other energy eigenstates are the excited states. Given that the vacuum is unique, we find that the chiral condensate vanishes for odd $N$ with anti-periodic boundary conditions

$$\frac{\langle E, 0 | (\psi_+^a \psi_-^a) | E, 0 \rangle}{\langle E, 0 | E, 0 \rangle} = 0. \tag{94}$$

The Polyakov loop acts on these quasi-ground states as

$$\frac{1}{N} \text{tr}[P] |E, n\rangle = \sum_\ell \omega^{n\ell} (\omega^\ell |\ell\rangle) = |E, n+1\rangle. \tag{95}$$

We therefore find, for $\tau \to \infty$, that

$$\frac{\langle E, 0 | \frac{1}{N^k} \text{tr}[P^\dagger]^k \exp(-\tau \Delta(\widehat{H} - E_0)) \frac{1}{N^k} \text{tr}[P]^k | E, 0 \rangle}{\langle E, 0 | E, 0 \rangle} = \exp\left(-\tau(E_k - E_0) + O(1)\right). \tag{96}$$

This allows us to relate $E_k - E_0$ to a $k$-string tension $\sigma_k$ as

$$\sigma_k = \frac{1}{L}(E_k - E_0) = \frac{2\Delta E}{L} \left(1 - \cos\frac{2\pi k}{N}\right). \tag{97}$$

The lesson we draw from this is that the string tensions do not vanish for any $k = 1, \ldots N-1$. The detailed $k$ dependence in this expression comes from our choice of model Hamiltonian and is not expected to be accurate.

**Even $N$**

For even $N$, the energy eigenstates can be labeled as $|E, n, \pm\rangle$, with $n = 0, 1, \ldots, N/2 - 1$. In terms of the original basis $|\ell_a\rangle$ they are given by

$$|E, n, \pm\rangle = \sum_{\ell=0}^{N/2-1} \omega^{2n\ell} \left(|2\ell\rangle \pm |2\ell + 1\rangle\right). \tag{98}$$

The eigenenergies depend on $n$ but not on the label $\pm$:

$$E_n = -2\Delta E \cos\left(\frac{2\pi n}{N/2}\right). \tag{99}$$

We therefore find the doubly-degenerate ground states $|E, 0, \pm\rangle$ when $N$ is even. They are related by a discrete chiral transformation:

$$\widehat{U}_\chi |E, 0, \pm\rangle = |E, 0, \mp\rangle. \tag{100}$$

The chiral condensate is given by

$$\frac{\langle E, 0, \pm|(\psi_+^a \psi_-^a)|E, 0, \pm\rangle}{\langle E, 0, \pm|E, 0, \pm\rangle} = \pm\langle 1|(\psi_+^a \psi_-^a)|0\rangle \sim e^{-(N-1)\frac{\pi^{3/2}}{L\lambda^{1/2}}}. \tag{101}$$

We can find a formula for the string tensions by the same process we followed for odd $N$, so that

$$
\begin{aligned}
\frac{1}{N^k}\mathrm{tr}[P]^k|E, n, \pm\rangle &= \sum_{\ell=0}^{N/2-1} \omega^{2n\ell}\left(\omega^{k(2\ell)}|2\ell\rangle \pm \omega^{k(2\ell+1)}|2\ell+1\rangle\right) \\
&= e^{\frac{2\pi i k}{2N}}\left(\cos\frac{2\pi k}{2N}|E, n+k, \pm\rangle - i\sin\frac{2\pi k}{2N}|E, n+k, \mp\rangle\right). \tag{102}
\end{aligned}
$$

Therefore, the $k$-string tension is given by

$$\sigma_k = \frac{1}{L}(E_k - E_0) = \frac{2\Delta E}{L}\left(1 - \cos\frac{4\pi k}{N}\right). \tag{103}$$

The main point of this expression is to illustrate that the string tensions for $k = N/2$ disappears, but the other string tensions remain finite. This is exactly what is demonstrated in Fig. 2. The detailed $k$ dependence in the expression is again not expected to be accurate.

Overall, these results are consistent with the our claim that for even $N$, chiral symmetry is spontaneously broken and center symmetry breaks as $\mathbb{Z}_N \to \mathbb{Z}_{N/2}$, as the mixed 't Hooft anomaly minimally requires.

## 7.2 Periodic boundary conditions

With periodic boundary conditions for the fermions, the GPY effective potential becomes

$$V_{\mathrm{eff,P}}(P) = -\frac{1}{2LV}\ln\det(-D_{\mathrm{adj}}^2) = \frac{1}{\pi L^2}\sum_{n=1}^{\infty}\frac{1}{n^2}(|\mathrm{tr}(P^n)|^2 - 1). \tag{104}$$

In this presentation of the GPY potential, we have included an $L$-dependent but holonomy-independent contribution (the $-1$ term) coming from the fact that we are working with an $SU(N)$ gauge theory. This is done so that when it is evaluated on the minimizing value of the holonomy, its value corresponds to the value of $\log \mathcal{Z}_{\mathrm{PBC}}$, where $\mathcal{Z}_{\mathrm{PBC}}$ is the partition function of the theory on $\mathbb{R} \times S^1$ with periodic boundary conditions.

The minimum of the GPY potential with periodic boundary conditions is unique up to the action of the Weyl group $S_N$. It is characterized by having $\text{tr}[P^n] = 0$ for $n = 1, \ldots, N-1$. It is very tempting to interpret the fact that $\langle \text{tr}[P^n] \rangle = 0$ as implying an unbroken $\mathbb{Z}_N$ center symmetry. This is indeed true for odd $N$. We will see that the story is a little bit more subtle for even $N$ due to the mixed anomaly of center and chiral symmetry.

In Polyakov gauge, the phases of the holonomy, which are defined mod $2\pi$, are

$$\theta_a = \frac{2\pi}{N} a, \tag{105}$$

up to its permutations. These permutations can be thought of as discrete Weyl gauge transformation, and we can fix the gauge completely by requiring $\theta_a$ to have the form of Eq. (105). But then center symmetry transformations generated by the operator $\widehat{U}_{\mathsf{S}}$ shift $\theta_a$ by $\frac{2\pi}{N}$, which is not consistent with our gauge fixing condition. To avoid this problem, we can redefine center transformations by including appropriate elements of the Weyl group to make $\theta_a = 2\pi a/N$ invariant under a new version of $\widehat{U}_{\mathsf{S}}$. With this done, the 0-form center symmetry acts nontrivially only on the fermion fields of the effective theory:

$$\widehat{U}_{\mathsf{S}} : \psi_{ij} \mapsto \psi_{i+1,j+1}. \tag{106}$$

We will discuss the actions of the operators of the other symmetries, $\widehat{U}_{\mathsf{F}}, \widehat{U}_{\chi}, \widehat{U}_C$ shortly.

In contrast to the situation with anti-periodic boundary conditions, in this center-symmetric vacuum there is an adjoint Higgs mechanism at work. It renders all of the non-Cartan components of the Majorana adjoint fermions massive, but the diagonal (Cartan) Kaluza-Klein zero mode components remain massless at tree level. The massive modes can be integrated out. This gives an effective Lagrangian written in terms of the Cartan Kaluza-Klein zero modes. At tree level, this effective Lagrangian is

$$\mathcal{L}_{0,\text{eff}} = -\sum_{a=1}^{N} (\psi_{+,a} \partial_\tau \psi_{+,a} + \psi_{-,a} \partial_\tau \psi_{-,a}), \tag{107}$$

where the $\pm$ labels label the chirality, there is a constraint $\psi_1 + \cdots + \psi_N = 0$, and we write as $\psi_a = \psi_{aa}$. In what follows, we will refer to the Cartan Kaluza-Klein zero modes as KK zero modes.

Canonical quantization implies that the operators associated to the Grassmann fields in the Lagrangian obey the anti-commutation relations

$$\{\widehat{\psi}_{+,a}, \widehat{\psi}_{+,b}\} = \{\widehat{\psi}_{-,a}, \widehat{\psi}_{-,b}\} = \frac{1}{2}\left(\delta_{ab} - \frac{1}{N}\right), \quad \{\widehat{\psi}_{+,a}, \widehat{\psi}_{-,b}\} = 0. \tag{108}$$

To work with this algebra, it is convenient to introduce some creation and annihilation operators

$$\widehat{C}_k = \frac{1}{\sqrt{N}} \sum_a \omega^{-ka}(\widehat{\psi}_{+a} - i\widehat{\psi}_{-a}), \quad \widehat{C}_k^\dagger = \frac{1}{\sqrt{N}} \sum_a \omega^{ka}(\widehat{\psi}_{+a} + i\widehat{\psi}_{-a}). \tag{109}$$

They satisfy the simpler anti-commutation relation

$$\{\widehat{C}_k^\dagger, \widehat{C}_\ell\} = \delta_{k\ell} - \delta_{k0}\delta_{\ell0}, \tag{110}$$

and all other anti-commutators vanish. Note that the constraint $\psi_1 + \cdots + \psi_N = 0$ maps to the statement that $\widehat{C}_0$ and $\widehat{C}_0^\dagger$ anti-commute with everything. That means that we can consistently set $\widehat{C}_0 = \widehat{C}_0^\dagger = 0$ when acting on the physical Hilbert space. Some physical insight into the $\widehat{C}_k, \widehat{C}_k^\dagger$ operators can be obtained by noticing that one can write gauge-invariant interpolating

operators for $\widehat{C}_k$ and $\widehat{C}_k^\dagger$, along the lines used in Ref. [138, 139]. These expressions take the form

$$\widehat{C}_k \sim \text{tr}[P^{-k}(\psi_+ - i\psi_-)], \tag{111}$$

$$\widehat{C}_k^\dagger \sim \text{tr}[P^{+k}(\psi_+ + i\psi_-)]. $$

The indices are cyclic of order $N$. With all this out of the way, one can write down the action of the symmetries on the creation and annihilation operators:

$$\begin{aligned}
\widehat{U}_S \widehat{C}_k \widehat{U}_S^{-1} &= \omega^{-k}\widehat{C}_k, & \widehat{U}_S \widehat{C}_k^\dagger \widehat{U}_S^{-1} &= \omega^k \widehat{C}_k^\dagger, \\
\widehat{U}_F \widehat{C}_k \widehat{U}_F^{-1} &= -\widehat{C}_k, & \widehat{U}_F \widehat{C}_k^\dagger \widehat{U}_F^{-1} &= -\widehat{C}_k^\dagger, \\
\widehat{U}_C \widehat{C}_k \widehat{U}_C^{-1} &= \widehat{C}_{-k}, & \widehat{U}_C \widehat{C}_k^\dagger \widehat{U}_C^{-1} &= \widehat{C}_{-k}^\dagger, \\
\widehat{U}_\chi \widehat{C}_k \widehat{U}_\chi^{-1} &= \widehat{C}_{-k}^\dagger, & \widehat{U}_\chi \widehat{C}_k^\dagger \widehat{U}_\chi^{-1} &= \widehat{C}_{-k}.
\end{aligned} \tag{112}$$

For the discussion that follows, it is especially helpful to consider the action of the symmetries on number operators built from $\widehat{C}_k, \widehat{C}_k^\dagger$,

$$O_i = \widehat{C}_i^\dagger \widehat{C}_i - \frac{1}{2}, \tag{113}$$

where the index $i$ takes the values $i = 1, 2, \ldots, N/2$ for even $N$ and $i = 1, 2, \ldots, (N-1)/2$ for odd $N$. The operators $O_i$ transform as

$$\widehat{U}_S : \; O_i \rightarrow +O_i, \tag{114}$$

$$\widehat{U}_C : \; O_i \rightarrow +O_{-i}, \tag{115}$$

$$\widehat{U}_\chi : \; O_i \rightarrow -O_{-i}, \tag{116}$$

where for even $N$ we interpret $O_{-N/2}$ as $O_{N/2}$.

It is also useful to define the operator

$$\widetilde{O} = \widehat{C}_1^\dagger \widehat{C}_2^\dagger \cdots \widehat{C}_{(N-1)/2}^\dagger \widehat{C}_{-1}^\dagger \widehat{C}_{-2}^\dagger \cdots \widehat{C}_{-(N-1)/2}^\dagger + \text{h.c.}, \tag{117}$$

when $N$ is odd. The operator $\widetilde{O}$ is neutral under $(\mathbb{Z}_2)_F, (\mathbb{Z}_2)_\chi$ and center symmetry, but it is odd under $(\mathbb{Z}_2)_C$ when $N = 4n + 3$. It is neutral under $(\mathbb{Z}_2)_C$ when $N = 4n + 1$.

### 7.2.1 Hilbert space structure

The QM effective field theory is built from a finite number of fermion fields, and so it has a finite-dimensional Hilbert space. We can generate the states in this Hilbert space by starting from the completely unoccupied state $|0\rangle$ defined by

$$\widehat{C}_k |0\rangle = 0 \quad \text{for all } k = 0, 1, \ldots, N-1, \tag{118}$$

and then acting on this unoccupied state with creation operators $\widehat{C}_k^\dagger$. The resulting Hilbert space has dimension $2^{N-1}$. Using Eq. (112), we find that $|0\rangle$ is an eigenstate of $\widehat{U}_S$, $\widehat{U}_F$ and $\widehat{U}_C$. Therefore, we can consistently choose to assign it charge $+1$ under all three symmetries:

$$\widehat{U}_S |0\rangle = \widehat{U}_F |0\rangle = \widehat{U}_C |0\rangle = |0\rangle. \tag{119}$$

One can also check that

$$\widehat{C}_k^\dagger \widehat{U}_\chi |0\rangle = 0 \tag{120}$$

for any $k$, and thus

$$\widehat{U}_\chi |0\rangle = \widehat{C}_1^\dagger \widehat{C}_2^\dagger \cdots \widehat{C}_{N-1}^\dagger |0\rangle. \tag{121}$$

Using this expression, one can verify that

$$\widehat{U}_{\mathsf{S}}\widehat{U}_{\chi} = (-1)^{N-1}\widehat{U}_{\chi}\widehat{U}_{\mathsf{S}}, \ \widehat{U}_{\mathsf{F}}\widehat{U}_{\chi} = (-1)^{N-1}\widehat{U}_{\chi}\widehat{U}_{\mathsf{F}}, \ \widehat{U}_{\mathsf{C}}\widehat{U}_{\chi} = (-1)^{\frac{(N-2)(N-1)}{2}}\widehat{U}_{\chi}\widehat{U}_{\mathsf{C}}, \qquad (122)$$

which is consistent with Eq. (42), the 't Hooft anomaly equation. This algebra says that the energy eigenstates must be doubly degenerate for $N = 4n$, $4n + 2$, and $4n + 3$. Singlet energy levels are only possible for $N = 4n + 1$.

**N = 2**

For $N = 2$, we have just two states

$$|0\rangle, \widehat{C}_1^{\dagger}|0\rangle. \qquad (123)$$

No Hamiltonian $H$ acting on the KK zero modes (except for a constant) is invariant under the whole symmetry group. Nothing can split these two states, so they can be interpreted as degenerate ground states.

If we let $|1\rangle \equiv \widehat{C}_1^{\dagger}|0\rangle$, then $\widehat{U}_{\chi}|0\rangle = |1\rangle$. Also, $|0\rangle$ and $|1\rangle$ are eigenstates of center symmetry with eigenvalue $+1$ and $-1$. They are also eigenstates with eigenvalues $\pm 1$ under fermion parity and charge conjugation.

To interpret these results in terms of spontaneous symmetry breaking is a little subtle. On $\mathbb{R} \times S^1$ with periodic boundary conditions, chiral symmetry and center symmetry do not commute due to the 't Hooft anomaly. So we are not allowed to ask whether *both* are spontaneously broken, because that would imply that we could measure charges under both of them simultaneously, which would be a contradiction.

If we pick one of these symmetries, then we can choose a basis in which the symmetry permutes the degenerate ground states. For example, the states $|0\rangle$ and $|1\rangle$ go into each other under the action of chiral symmetry. This makes it tempting (and indeed correct) to say that chiral symmetry is spontaneously broken. But one could have instead worked with the states $|\pm\rangle = |0\rangle \pm |1\rangle$, and then center symmetry would permute $|\pm\rangle$. Then it would be tempting (and equally correct) to say that center symmetry is spontaneously broken.

We should emphasize that the deconfined non-contractible line operator is not the usual Polyakov loop, $\text{tr}(P)$. Instead, the deconfined objects are given by $C_1$ and $C_1^{\dagger}$, which is a composite object of the Polyakov loop and the fermionic zero mode, $\text{tr}(P\psi_{\pm})$. This happens due to the fact that not only the center symmetry but also the fermion parity has a mixed anomaly with the discrete chiral symmetry. In order to construct the kink configuration that connects two chiral-broken vacua, we need an operator that is charged under both 0-form center symmetry and fermion parity.

Finally, note that the fact that the degenerate ground states have opposite fermion parity means that a $(-1)^F$ graded partition function of this system must *vanish*.

**N = 3**

For $N = 3$, there are 4 states: $|0\rangle, \widehat{C}_1^{\dagger}|0\rangle, \widehat{C}_{-1}^{\dagger}|0\rangle, \widehat{C}_1^{\dagger}\widehat{C}_{-1}^{\dagger}|0\rangle$. The Hamiltonian that acts on the KK zero mode sector is constrained by symmetries to have only one parameter $\epsilon$:

$$H = 4\epsilon \, O_1 O_{-1}. \qquad (124)$$

The value of the coefficient $\epsilon$ could be fixed by matching to the 2d theory, but we do not so explicitly. Using this Hamiltonian, the energy and eigenstates are given by

| Energy | States |
|---|---|
| $E = \epsilon$ | $|0\rangle, \widehat{C}_1^{\dagger}\widehat{C}_{-1}^{\dagger}|0\rangle$ |
| $E = -\epsilon$ | $\widehat{C}_1^{\dagger}|0\rangle, \widehat{C}_{-1}^{\dagger}|0\rangle$ |

(125)

We find that the states are two-fold degenerate. This can be interpreted as a signal of the breaking of charge conjugation symmetry. Charge conjugation does not commute with chiral symmetry, so we could do a change of basis and instead interpret the two-fold degeneracy as a signal of chiral symmetry breaking. This is the same subtlety encountered in our discussion of chiral symmetry and center symmetry at even $N$.

Here $N$ is odd, so center symmetry commutes with chiral symmetry. This makes it meaningful to ask about its realization. One can see that center symmetry is not spontaneously broken from the fact that states with different center charge have different energies. We also note that when $N = 3$ the states are not Bose-Fermi paired. So the $(-1)^F$ graded partition function does not vanish at $N = 3$.

If a $(\mathbb{Z}_2)_C$-breaking four-Fermi term of Eq. (15) were to be turned on in the action, then we would be allowed to add $\widetilde{O}$ to the Hamiltonian of our KK zero mode effective field theory. This would split the states in (125) and lead to a singlet ground state, consistently with the Fidkowski-Kitaev result that interacting systems of 8 Majorana fermions in one spatial dimension have a trivially-gapped phase of matter [15] if the interactions only preserve the $(\mathbb{Z}_2)_F$ and $(\mathbb{Z}_2)_\chi$ symmetries.

**N = 4**

For $N = 4$, there are 8 states. The symmetries imply that the Hamiltonian that acts on the Kaluza-Klein zero modes has two free parameters:

$$H = 2\epsilon_1 (O_1 + O_{-1}) O_2 + 4\epsilon_2 O_1 O_{-1}. \tag{126}$$

The values of $\epsilon_1, \epsilon_2$ are in principle determined within the underlying 2d theory, but again, we will not do the matching explicitly. The energy and eigenstates are given by

| Energy | States |
|---|---|
| $E = \epsilon_1 + \epsilon_2$ | $\lvert 0 \rangle$, $\widehat{C}_1^\dagger \widehat{C}_2^\dagger \widehat{C}_{-1}^\dagger \lvert 0 \rangle$ |
| $E = -\epsilon_1 + \epsilon_2$ | $\widehat{C}_2^\dagger \lvert 0 \rangle$, $\widehat{C}_1^\dagger \widehat{C}_{-1}^\dagger \lvert 0 \rangle$ |
| $E = -\epsilon_2$ | $\widehat{C}_1^\dagger \lvert 0 \rangle$, $\widehat{C}_{-1}^\dagger \lvert 0 \rangle$, $\widehat{C}_1^\dagger \widehat{C}_2^\dagger \lvert 0 \rangle$, $\widehat{C}_2^\dagger \widehat{C}_{-1}^\dagger \lvert 0 \rangle$ |

$$\tag{127}$$

We see that the states are at least two-fold degenerate. States with different center charges are now degenerate, so center symmetry is spontaneously broken. One can verify that the pattern of center-breaking is $\mathbb{Z}_4 \to \mathbb{Z}_2$. Chiral symmetry is also spontaneously broken, as can be seen from the degeneracies between states with different chiral charges.

We again observe that all states are Bose-Fermi paired, so the $(-1)^F$-graded partition function of $SU(N = 4)$ adjoint QCD vanishes.

**N = 5**

For $N = 5$, we have 16 states in the KK zero mode sector. The Hamiltonian acting on these states has four parameters:

$$H = \epsilon_1(O_1 + O_{-1})(O_2 + O_{-2}) + \epsilon_2(O_1 - O_{-1})(O_2 - O_{-2}) + \epsilon_3 O_1 O_{-1} O_2 O_{-2} + \epsilon_4 \widetilde{O}. \tag{128}$$

Since $5 = 4 \times 1 + 1$, the operator $\widetilde{O}$ can appear in the Hamiltonian without breaking any symmetries. Its appearance leads to the emergence of a singlet ground state. To illustrate this point, consider setting $\epsilon_1 = \epsilon_2 = \epsilon_3 = 0$, and keep $\epsilon_4 \neq 0$. Then we find the following

spectrum:

| Energy | States |
|---|---|
| $E = \epsilon_4$ | $|0\rangle + \widehat{C}_1^\dagger \widehat{C}_2^\dagger \widehat{C}_{-2}^\dagger \widehat{C}_{-1}^\dagger |0\rangle$ |
| $E = -\epsilon_4$ | $|0\rangle - \widehat{C}_1^\dagger \widehat{C}_2^\dagger \widehat{C}_{-2}^\dagger \widehat{C}_{-1}^\dagger |0\rangle$ |
| $E = 0$ | Other 14 states |

(129)

This confirms that including the operator $\widetilde{O}$ in the effective Hamiltonian — which is allowed by symmetries and hence will generically be forced by the dynamics — can leads to a singlet ground state for $N = 5$. Therefore, $SU(5)$ 2d adjoint QCD can be in a trivial gapped phase, as predicted by general constraints on the pattern of breaking of 1-form global symmetries along with constraints from 't Hooft anomalies.

Bosonic and fermionic states are not paired when $N = 5$.

# 8 Bose-Fermi pairing and the large $N$ limit

In this section, we discuss the pattern of Bose-Fermi degeneracies in 2d adjoint QCD. The fact that there are any such degeneracies to discuss is surprising and interesting because 2d adjoint QCD is a manifestly non-supersymmetric theory. Its microscopic matter content is not consistent with the minimal non-chiral 2d supersymmetry algebra $\mathcal{N} = (1, 1)$.[18] Nevertheless, in Sec. 4 we pointed out that when $N$ is even, the mod 2 index theorem guarantees the existence of fermionic zero modes on the torus with periodic boundary conditions. From the path integral point of view, the torus partition function $\mathcal{Z}$ must vanish whenever there are fermion zero modes in the functional integral measure. If we take periodic boundary conditions on both cycles of the torus, and compute the corresponding $(-1)^F$-graded partition function $\tilde{\mathcal{Z}}$, the mod 2 index theorem implies that $\tilde{\mathcal{Z}} = 0$. From a Hilbert space perspective, the vanishing of $\tilde{\mathcal{Z}}$ at even $N$ implies exact Bose-Fermi degeneracies.

Let us discuss the details of how the pairing between bosonic and fermionic states takes place in massless 2d adjoint QCD with even $N$. The ground states are necessarily in the KK zero mode sector discussed in Sec. 7.2. We have seen that when $N$ is even, there is a bosonic ground state $|0\rangle_B$ and also a degenerate fermionic ground state $|0\rangle_F$. This fermionic ground state owes its existence to the presence of fermion zero modes. Evaluating the GPY potential in Eq. (104) on a center-symmetric holonomy gives the $L$-dependent part of the vacuum energy associated to the bosonic vacuum,

$$
\begin{aligned}
E_{\text{gs,bosonic}}(L) &= V_{\text{eff,P}}(P)_{P_{\min}} + V_{\text{two-loop}} \\
&= \frac{-1}{\pi L^2} \zeta(2) + \sum_{n=1}^{\infty} \frac{1}{(Nn)^2} N^2 + V_{\text{two-loop}} \\
&= 0 + V_{\text{two-loop}}.
\end{aligned}
$$

(130)

The one-loop contribution works out to be zero, though we do not see why the same should remain true at two loops and above.[19] In any case, to get the complete contribution to the $(-1)^F$-graded partition function we should subtract the contribution from the fermionic vacuum. For even $N$, the bosonic and fermionic vacua are paired, so the contributions of the ground state energies cancel in $\tilde{\mathcal{Z}}$. At odd $N$ there is no Bose-Fermi pairing of vacua, and hence no reason for such a cancellation to occur.

---

[18]Two-dimensional Yang-Mills theories with manifest $\mathcal{N} = (1,1)$ supersymmetry were studied in Refs. [140,141]. These models differ from 2d adjoint QCD in having adjoint scalar fields.

[19]It would be interesting to generalize the discussion in Ref. [22] to two spacetime dimensions and look at this more explicitly.

Now we turn to the non-zero modes. Consider an arbitrary bosonic creation operator $a_B^\dagger$ built out of KK non-zero modes, and also an arbitrary fermionic creation operator $a_F^\dagger$ built out of KK non-zero modes. Focusing on even $N$ and acting on the ground states $|0\rangle_B$ and $|0\rangle_F$, we find that

$$a_B^\dagger |0\rangle_B, \qquad a_B^\dagger |0\rangle_F \tag{131}$$

are degenerate finite-energy states of opposite statistics, and the same is true for

$$a_F^\dagger |0\rangle_B, \qquad a_F^\dagger |0\rangle_F. \tag{132}$$

Therefore the complete spectrum of 2d adjoint QCD is exactly Bose-Fermi paired when $N$ is even. In particular, this means that the excited states also cancel in the $(-1)^F$-graded partition function $\tilde{\mathcal{Z}}$. Now we understand the Hilbert space interpretation of the vanishing of the even $N$ torus partition function with periodic boundary conditions on both cycles. As a corollary, we also see that the $\tilde{\mathcal{Z}}$ does not vanish when $N$ is odd — and moreover $\tilde{\mathcal{Z}}$ is a non-trivial function of the geometry — because there is no Bose-Fermi pairing of vacua for odd $N$.

The above discussion shows that the two chiral-symmetry-broken vacua have opposite fermionic parity, $(-1)^F$. If we turn on a fermion mass in the theory on a circle of size $L$, the energy of the fermion zero mode gets lifted to $E \sim m\sqrt{\lambda}L$. The large $L$ and small $m$ limits do not commute, which is not surprising because chiral symmetry is spontaneously broken. But here this standard phenomenon has the curious implication that if the theory is defined on a finite $T^2$ and $m = 0$, then the partition function vanishes due to Bose-Fermi pairing. But if instead we take $L \to \infty$ before taking $m \to 0$, then the theory gets frozen into one chiral vacuum, and the fermionic 'vacuum' is invisible.

This explains why studies of 2d adjoint QCD based on numerical light-cone quantization did not observe any Bose-Fermi pairing at $m = 0$, see e.g. [5]: when working on $\mathbb{R}^2$ at $m = 0$, if one only considers states created by acting with local operators on a single vacuum, it is not possible to detect the Bose-Fermi degeneracy shown in this paper.

## 8.1 Large $N$ limit

We now turn to the large $N$ limit of 2d adjoint QCD. In the previous sections we have seen that the theory is in a confining phase on $\mathbb{R}^2$ for $N > 2$, with center symmetry either completely unbroken (odd $N$) or broken to $\mathbb{Z}_{N/2}$ (even $N$). What happens to the physics if we fix $\lambda = g^2 N$ and the coupling constants $c_1, c_2$ in Eq. (14) and take $N$ to infinity?

In the large $N$ limit, the differences between even $N$ and odd $N$ become essentially irrelevant: in either case test charges of $n$-ality $n \ll N$ are confined, and it is natural to expect the theory to have a finite fundamental string tension at $N = \infty$. A finite string tension is associated with an exponentially growing Hagedorn density of states, of the form

$$\rho(E) \sim m^\alpha e^{\beta_H E}, \qquad N = \infty, \tag{133}$$

where $\beta_H \sim \lambda^{1/2}$, and $\alpha$ is a presently unknown numerical parameter [4]. This exponential growth was numerically confirmed in Ref. [5]. Heuristically, Hagedorn scaling appears because at large $N$ the single-particle states are created by single-trace operators, and the number of single-trace operators grows exponentially with their engineering dimension [68, 142, 143]. The overlap between single-trace and multi-trace operators vanishes as $N \to \infty$, corresponding to the statement that hadronic decays are suppressed at large $N$. To see why the number of single-trace operators, and hence of single-particle states, grows exponentially, note that a typical single-trace operator can be written as

$$\text{tr}(\psi_+ \psi_- \psi_+ \psi_+ \psi_- \cdots \psi_-), \tag{134}$$

and if $K$ is the number of $\psi_\pm$ insertions then the number of such operators grows exponentially, very roughly as $2^K$. Hagedorn scaling of the density of states is generally viewed as a signature of an underlying string theory description of the system. It can only ever appear at infinite $N$, because at finite $N$ highly-excited states can decay. In fact, at finite $N$, the system's high-energy density of states must approach that of a local 2d conformal field theory

$$\rho(E) \sim e^{\sqrt{E}}, \qquad \text{finite } N. \qquad (135)$$

Hagedorn behavior has an important implication: the theory cannot remain in the confined phase for all values of the temperature $T = 1/\beta$ in the large $N$ limit. When $\beta \to \beta_H$, the confined-phase thermal partition function

$$\mathcal{Z}(\beta) = \text{tr}[e^{-\beta\hat{H}}] = \int_0^\infty dE\, \rho(E) e^{-\beta E} \qquad (136)$$

becomes singular. Note that the large $N$ limit is crucial for this argument, because Hagedorn behavior appears only at large $N$. So, at large $N$, there must be a phase transition at some $\beta_c \geq \beta_H$ to some phase without Hagedorn scaling. This phase is expected to be deconfined, characterized by a spontaneously-broken center symmetry.

This result is consistent with our analysis. As we discussed in Sec. 3, in 2d theories center symmetry cannot break at finite $N$ for any $\beta$, unless forced to do so by an anomaly. In Sec. 4, we showed that such an anomaly is only present at even $N$, and even then it only allows center symmetry to break to $\mathbb{Z}_{N/2}$. For large but finite $N$, this is very similar to no breaking at all. And indeed, in Sec. 7 we have seen that tunneling restores the naively-broken center symmetry at finite $N$.

At $N = \infty$, more careful analysis is required and the situation is different. The tunneling amplitude $\mathcal{A}$ connecting nearest-neighbor center-breaking field configurations is set by the instanton action in Eq. (86), and scales as

$$\mathcal{A} \sim e^{-N\frac{\pi^{3/2}}{\beta\lambda^{1/2}}}. \qquad (137)$$

When $N \to \infty$, the tunneling amplitude vanishes. As a result, the $\mathbb{Z}_N$ center symmetry of 2d adjoint QCD is completely broken at high temperature in the large $N$ limit.

The situation is completely different if the fermion boundary conditions are periodic. We will call the circumference of the circle $L$ in this case rather than $\beta$. The holonomy effective potential relevant to these boundary conditions, Eq. (104), indicates that center symmetry is not spontaneously broken at small $L$ for any $N$. The potential has a unique minimum, so tunneling considerations are irrelevant, and there is no deconfinement at large $N$ either. This is consistent with the absence of any phase transitions as a function of circle size with periodic boundary conditions.

To understand this result more deeply, we observe that the Euclidean path integral on $\mathbb{R} \times S_L^1$ calculates the $(-1)^F$ graded partition function

$$\tilde{\mathcal{Z}}(L) = \text{tr}[(-1)^F e^{-L\hat{H}_\mathbb{R}}] = \int_0^\infty dE\, [\rho_B(E) - \rho_F(E)] e^{-LE}, \qquad (138)$$

where $\rho_B$ and $\rho_F$ are the bosonic and fermionic densities of states, and the subscript $\mathbb{R}$ on $\hat{H}$ is a reminder that the Hamiltonian measures energies of states on $\mathbb{R}$. At the beginning of Sec. 8 we made two key observations.

One is that for even $N$ all bosonic and fermionic states — including the vacuum states — come in degenerate pairs. For finite odd $N$, there is no exact Bose-Fermi pairing. The origin of

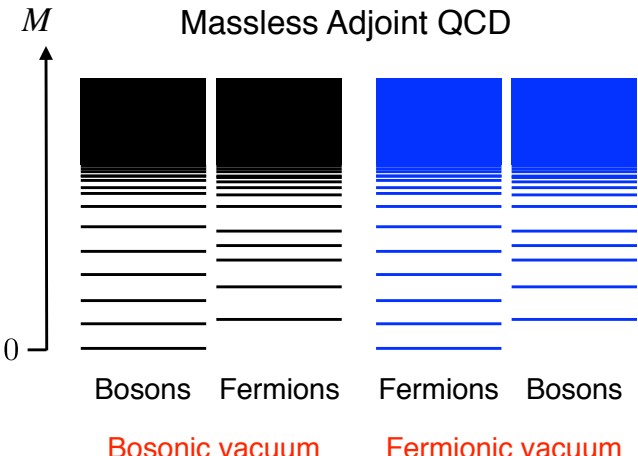

Figure 4: Sketch of the spectrum of 2d massless adjoint QCD with even $N$ on $\mathbb{R} \times S^1$. The theory has two vacua with opposite fermion parity. These vacua remain degenerate even at finite volume due to the 't Hooft anomaly involving $(\mathbb{Z}_2)_F \times (\mathbb{Z}_2)_\chi$. The Hilbert spaces built using either one of these vacua exhibit a very powerful conspiracy involving the masses of bosonic and fermionic states. They are not paired energy-level by energy-level, but their overall distribution is such that there is no Hagedorn scaling in the $(-1)^F$-graded density of states.

the exact Bose-Fermi pairing at even N is the presence of fermionic zero modes in the Euclidean path integral with periodic boundary conditions. These zero modes are a consequence of the mod 2 index theorem. This means that the torus partition function with periodic boundary conditions on both cycles *vanishes* for even $N$. The exact vanishing of the torus partition function is due to the fact that the theory contains two superselection sectors. There is no tunneling between them even in finite volume due to the anomaly.

The second key observation is that center symmetry is not spontaneously broken at small $L$. Combining this with our comments on Hagedorn behavior, as well as with the existence of two completely decoupled superselection sectors implies that all Hagedorn growth in the graded density of states calculated in *either* of the two superselection sectors sectors must cancel. This implies a remarkable Bose-Fermi conspiracy in the spectrum at large $N$. We illustrate the spectrum of massless adjoint QCD with even $N$ in Fig. 4. This phenomenon parallels the Bose-Fermi conspiracies in 4d adjoint QCD discussed in Refs. [19, 20, 22].

We expect the large $N$ limit with fixed $\lambda = g^2 N$ and fixed $c_1, c_2$ to be smooth. Then the difference between even and odd $N$ must disappear at large $N$ in the observable $\tilde{\mathcal{Z}}$. For this to be the case, the torus partition function with periodic boundary conditions on both cycles must vanish in the large $N$ limit up to $1/N$ corrections. If the two cycles of the torus have circumferences $\beta$ and $L$, the torus partition function can be thought of as

$$\tilde{\mathcal{Z}}(L, \beta) = \text{tr}[(-1)^F e^{-\beta \hat{H}_L}] = \int_0^\infty dE_L \, [\rho_B(E_L) - \rho_F(E_L)] e^{-\beta E_L}, \qquad (139)$$

where $\hat{H}_L$ is the Hamiltonian of the theory on a spatial circle of size $L$ with periodic boundary conditions. $\hat{H}_L$ has a discrete spectrum $E_L$, with the subscript serving as a reminder that the energies depend on $L$. Putting all the arguments together, we find

$$\rho_B(E_L) = \rho_F(E_L) \qquad \qquad N = \infty. \qquad (140)$$

Even though we started with a theory which is manifestly not supersymmetric at the Lagrangian level, at large $N$ we end up with a supersymmetric-looking result (140).

It is also amusing to note the following. At $N = 4n + 3$, we have seen that there is a mixed 't Hooft anomaly involving $(\mathbb{Z}_2)_F \times (\mathbb{Z}_2)_C \times (\mathbb{Z}_2)_\chi$. In particular, the partition function on $T^2$ has a fermionic zero mode if one takes periodic boundary conditions on one circle, say $x_1$, while taking anti-periodic boundary conditions supplemented by a twist by charge conjugation in the other direction $x_2$. This means that when $N = 4n + 3$,

$$\mathcal{Z}_C(L, \beta) = \text{tr}[(-1)^C e^{-\beta \hat{H}_L}] = 0. \tag{141}$$

Physically, this means that charge conjugation even and odd states contribute identically to the partition function. But at large $N$, the difference between $N = 4n, 4n+1, 4n+2$ and $4n+3$ becomes negligible. This implies that $\mathcal{Z}_C(L, \beta)$ vanishes up to $1/N$ corrections for any large $N$!

## 8.2 Bose-Fermi degeneracy without supersymmetry

In this section we explore whether a spectrum like Eq. (140) means that a theory is supersymmetric. We will see that this is not the case in general. Seeing why that is so gives extra insight into the cancellations discussed above.

Could it be that, despite the fact that 2d massless adjoint QCD with even $N$ does not enjoy any obvious supersymmetry at the Lagrangian level, nevertheless the theory has an exact hidden supersymmetry, which is spontaneously broken? Such a hypothesis would be consistent with the vanishing of the $(-1)^F$-graded partition function on $T^2$, since in a supersymmetric theory this would mean that the Witten index vanishes, so that spontaneous supersymmetry breaking is allowed. But as we have already discussed in Sec. 5, at least at the point in the parameter space of the massless theory where the parameters $c_1, c_2, c_3, d_1$ are tuned to zero, there must be a mass gap in the theory on $\mathbb{R}^2$ (at least at any finite $N$). This means that the theory cannot have a Goldstino mode. Therefore, 2d massless adjoint QCD with even $N$ cannot be a supersymmetric theory with a spontaneously broken supersymmetry.

One could also ask whether, despite the manifest absence of supersymmetry at short distance scales, there might nevertheless be an approximate (emergent) supersymmetry at long distances. The answer is no: if that were so, then there would not be *exact* Bose-Fermi pairing for *all* states in Hilbert space. And yet, precisely such exact Bose-Fermi pairings are observed for all states in the full Hilbert space of the theory.

All of this leads us to conclude that 2d massless adjoint QCD with even $N$ is a rather unusual theory: in some ways, it acts as if it was a supersymmetric quantum field theory with a spontaneously broken supersymmetry. However, we believe that it is *not* a supersymmetric theory, at least at finite $N$, because in general its spectrum does not include a massless Goldstino mode.

It would be interesting to better understand the class of non-supersymmetric theories with exact Bose-Fermi relations. For example, any system of free or interacting $2k + 1$ Majorana fermions with $(\mathbb{Z}_2)_F$ and $(\mathbb{Z}_2)_\chi$ symmetries along with a mixed $(\mathbb{Z}_2)_F \times (\mathbb{Z}_2)_\chi$ 't Hooft anomaly has this property, without the need to make the interactions supersymmetric. Adjoint QCD with even $N$ is, of course, precisely a system of this type. It would be nice to understand whether there are examples with similar properties outside of this class.

Next, we should note that the 1990s literature on 2d adjoint QCD contained some results on Bose-Fermi degeneracies, but these results were different from what we have described above. Using light-cone quantization, Kutasov and collaborators showed that 2d adjoint QCD has a hidden $\mathcal{N} = (1, 1)$ supersymmetry at $m^2 = g^2 N / \pi$ in the large $N$ limit when the classically marginal couplings $c_1, c_2, c_3, d_1, d_2$ are tuned to zero [4, 30], where $c_i$ are the four-fermion

couplings and $d_i$ are couplings of the Pauli-like terms. Kutasov showed that at large $N$, $\mathrm{tr}(-1)^F$ must vanish at the supersymmetric point $m^2 = g^2 N/\pi$, meaning that all of the states, including the vacuum states, are Bose-Fermi paired.[20]

While the punchline of Refs. [4, 30] sounds very similar to our results, the details are quite different. In our analysis, we saw that the torus partition function vanishes at the chiral point $m = 0$. It is entertaining to examine what happens when a finite quark mass is turned using our formalism. Generically, all Bose-Fermi pairings are lifted. This happens because turning on $m \neq 0$ removes the 't Hooft anomalies, all the fermion zero modes get lifted, and the large $N$ torus partition function $\tilde{\mathcal{Z}}$ becomes a non-trivial function of the torus geometry. It is instructive to see this via an explicit calculation. The one-loop GPY holonomy effective potential with massive adjoint fermions with periodic boundary conditions is

$$V(P) = \frac{m^2}{\pi} \sum_{n=1}^{\infty} \frac{K_1(nmL)}{nmL} (|\mathrm{tr}(P^n)|^2 - 1), \tag{142}$$

where $K_1$ is a modified Bessel function of the second kind. This expression is valid so long as $\lambda L^2 \ll 1$, and so long as the $c_1$ and $c_2$ couplings are either asymptotically free or tuned to zero, so that working on a small circle results in a weakly-coupled theory. Evaluating the holonomy effective potential on the minimizing value of $P$ — which is center-symmetric — gives the ground state energy as a function of $L$,

$$E_{\mathrm{gs}}(L,m) = -\frac{m^2}{\pi} \sum_{n=1}^{\infty} \frac{K_1(nmL)}{nmL} + \frac{Nm^2}{\pi} \sum_{n=1}^{\infty} \frac{K_1(nNmL)}{nmL}$$

$$\rightarrow -\frac{m^2}{\pi} \sum_{n=1}^{\infty} \frac{K_1(nmL)}{nmL}, \qquad \text{large } N. \tag{143}$$

In the lower line we took the large $N$ limit, where the second term in the top line becomes exponentially suppressed. One can think of $E_{\mathrm{gs}}(L,m)$ as the Casimir energy, and Eq. (143) shows that it depends on $L$ for any finite $m$ both at finite $N$ and at large $N$, including the point $m = \sqrt{\lambda/\pi}$. For a generic fermion mass, it costs an extra energy $\sim |m|$ to create a fermionic state. Consider a regime where $\tilde{\mathcal{Z}}$ is dominated by the Casimir energy, namely $\beta \Delta(L) \gg 1$, where $\Delta(L)$ is the mass gap. Then for generic non-zero $m$

$$\tilde{\mathcal{Z}} \sim e^{-\beta E_{\mathrm{gs}}(L,m)}, \qquad \text{finite } m. \tag{144}$$

Note that the quantity $E_{\mathrm{gs}}(L,m)$ is scheme-independent, because the $L$-dependence of $E_{\mathrm{gs}}(L)$ cannot be changed by adjusting local counter terms. So the $L$- and $\beta$-dependence of Eq. (144) is physical, and for general values of $m$ there cannot be any exact Bose-Fermi pairing. However, Refs. [4,30] observed that supersymmetry emerges at a very special value of $m^2 = \lambda/\pi$, at least in the large $N$ limit with $c_1, c_2, c_3, d_1, d_2 = 0$. Consistency with the discussion we gave above implies that there must be an interesting fermionic state. This state has zero energy at $m = 0$, but when $m > 0$ is turned on its energy moves away from zero, comes back to zero energy at the special point $m^2 = \lambda/\pi$, then moves away from zero energy again for $m^2 > \lambda/\pi$.[21] Otherwise, the Witten index could not be a constant in $L$ and $\beta$ at $m^2 = \lambda/\pi$. This suggests

---

[20] The adjoint fermion canonical anti-commutation relations at equal light-cone time $x^+$ are

$$\{\psi_{ij}(x^-), \psi_{k\ell}(y^-)\} = \frac{1}{2} \delta(x^- - y^-) \left[ \delta_{i\ell} \delta_{jk} - \frac{1}{N} \delta_{ij} \delta_{k\ell} \right],$$

where $i, j, k, \ell = 1, \ldots N$ are color indices. The $1/N$ term was set to zero in Eq. 3.5 of Ref. [4], which is an approximation that is valid at large $N$.

[21]Reference [5] verified numerically that the bosonic and fermionic ground states are degenerate at $m^2 = \lambda/\pi$ using discrete light-cone quantization at large $N$.

that the Witten index of the theory at $m^2/\lambda = 1/\pi$ must vanish, in agreement with other arguments for this conclusion given by Kutasov [4].

This means that 2d adjoint QCD is a rare example of a field theory with a partition function which vanishes at two loci in its parameter space. One of these vanishing loci — the one at $m = 0$ — is robust against the variation of four of the five dimensionless continuous parameters $c_1, c_2, c_3, d_1$ of the theory which do not violate chiral symmetry, and is present for any even $N$, as well as at large $N$. The other vanishing locus is point-like, appearing at $(m^2/\lambda, 1/N, c_1, c_2, c_3, d_1, d_2) = (1/\pi, 0, 0, 0, 0, 0, 0)$. The emergence of supersymmetry at this latter point implies, a posteriori, that setting $c_1 = c_2 = c_3 = d_1 = d_2 = 0$ is self-consistent at $m^2/\lambda = \pi$: these terms will not be generated by loop corrections due to the supersymmetry in the large $N$ limit. If $m$ deviates from the supersymmetric point, however, the five continuous parameters will be induced radiatively, so they should be included in the Lagrangian.

### 8.3   Large $N$ Bose-Fermi cancellations at finite $m$

The last thing we want to discuss is the way Bose-Fermi cancellations take place at finite $m$, which if anything is even more interesting than at $m = 0$. At finite $m$, the behavior of the theory is quite reminiscent of asymptotic/misaligned supersymmetry [144–146].[22] In particular, one can show that the miraculous spectral cancellations we saw at $m = 0$ continue to hold — in a modified way — for any finite $m$!

To see this, first note that turning on $m \neq 0$ lifts the degeneracy between bosonic and fermionic vacua. There are no longer two superselection sectors in Hilbert space, so any Bose-Fermi cancellations must take place in the *same* vacuum. Next, note that the theory with $m \neq 0$ is confining on $\mathbb{R}^2$, because it is in the same universality class as pure 2d YM theory, which confines with a string tension $\sim \lambda$, see *e.g.* [147–152]. On the other hand, one can see from Eq. (142) that putting massive adjoint QCD on a circle with periodic boundary conditions does not lead to deconfinement no matter how small we make the circle. But this means that there cannot be Hagedorn growth in the $(-1)^F$ graded partition function for massive 2d adjoint QCD, despite the fact that bosonic and fermionic states are no longer exactly degenerate. Somehow the distributions of the bosonic and fermion states are still very tightly related in just the right way to cancel all exponential growth in the graded partition function. To appreciate the strength of the cancellations, note that we are dealing with a non-trivial confining theory with an infinite number of Regge trajectories. Such theories are expected to have densities of states with large-energy expansions of the form *e.g.*

$$\rho_B(E) = e^{\beta_{H,b}^{(1)}E} b_1(E) + e^{\beta_{H,b}^{(2)}E} b_2(E) + \cdots \tag{145}$$

$$\rho_F(E) = e^{\beta_{H,f}^{(1)}E} f_1(E) + e^{\beta_{H,f}^{(2)}E} f_2(E) + \cdots, \tag{146}$$

where $\beta_{H,b}^1 > \beta_{H,b}^2 > \ldots > 0$ and $\beta_{H,f}^1 > \beta_{H,f}^2 > \ldots > 0$ are infinite sequences of Hagedorn 'temperatures', and $b_i(E), f_i(E), i \in \mathbb{N}$ are an infinite set of functions with sub-exponential growth. (In general such transseries expansions also contain terms that are oscillatory in $E$, as well as terms that are exponentially small with $E$, but they are not relevant to us here.) To avoid deconfinement for all circle sizes, it is necessary that

$$\beta_{H,b}^{(i)} = \beta_{H,b}^{(i)} \tag{147}$$
$$b_i(E) = f_i(E)$$

for all $i \in \mathbb{N}$. This is a remarkable requirement for a non-supersymmetric theory. The same requirement arises in 4d adjoint QCD [19–22] and in misaligned supersymmetry [144–146]. It

---

[22]This was already noticed in Ref. [4].

is illuminating to see this requirement appear — and be satisfied — in the tractable 2d context of 2d adjoint QCD.

# 9 Relations to previous literature

Our results do not fully agree with some previous studies of 2d adjoint QCD. Given the severity of the clash in the main physical conclusions, we feel obliged to explain the origins of the disagreements. We will mostly focus on a comparison of our results with the highly influential work of Gross, Klebanov, Matytsin, and Smilga [35]. At the end, we will also briefly comment on the numerical results of Refs. [5, 36, 46, 49].

The basic claim of Ref. [35] is that the physics of 2d adjoint QCD is largely the same as that of the Schwinger model. More specifically, Ref. [35] argued that 2d adjoint QCD does not confine test charges of any $N$-ality, and also argued that chiral symmetry is spontaneously broken for any $N$ (see also the important work [34]). We now review some of the arguments of Ref. [35] (which were widely used in much of the subsequent literature) to explain the differences from our results.

**Chiral symmetry realization:** First, let us discuss chiral symmetry breaking. The papers [34,35] proposed that discrete chiral symmetry is spontaneously broken by a fermion bilinear condensate,

$$\langle \text{tr}[\psi^T i\gamma\psi] \rangle \neq 0, \tag{148}$$

on $\mathbb{R}^2$ for *any* $N$. On the other hand, we found evidence that chiral symmetry is broken for even $N$ and $N = 4n + 3$, but saw no evidence from anomalies or systematic semiclassical calculations that chiral symmetry breaks for $N = 4n + 1$.

Refs. [34,35] ran into an apparent paradox in calculating the chiral condensate in the weak coupling semi-classical regime for $N \geq 3$ for theory compactified on $\mathbb{R} \times S^1$. The difficulty is that they found $2(N-1)$ fermion zero modes ($N-1$ zero modes of positive chirality and $N-1$ zero modes of negative chirality) associated with the tunnelings among the adjacent minima of the holonomy potential (75). References [34, 35] argued that this number is far too large to generate a fermion bilinear condensate in the semi-classical regime.

However, in this work, we have seen that most of these zero modes are not robust due to the mod 2 index theorem. All theories with even $N$ only have *one* robust pair of zero modes (of opposite chirality), and there is no difficulty in generating a chiral condensate, resolving the paradox in the semi-classical regime. For odd $N$ and anti-periodic boundary conditions, generically all zero modes are lifted and there is no chiral condensate. For example, the explicit calculations of $\zeta$ in Sec. 4.3 give an example of field configurations in which the zero modes satisfy the pattern described above. The gauge configuration described in that section (with 't Hooft flux $k = 1$) has $\zeta = 1$ for any even $N$, but $\zeta = 0$ for any odd $N$, even without doing any counting in mod 2.

This is in accordance with the presence of a persistent mixed anomaly for even $N$ on $\mathbb{R} \times S^1$, as well as its absence for odd $N$ in the same setting, as summarized in our Eq. (73). For the odd $N$ case, the anomaly persists upon compactification if and only if a $(-1)^F$ background gauge field is turned on, and also $N = 4n + 3$. Indeed, we have shown the existence of two vacua in this case in Sec. 7.2 by using weak coupling semi-classical analysis. This is also nicely consistent with the triple mixed anomaly (74). However, for $N = 4n + 1$, there is no mixed anomaly and the ground state is unique in the semi-classical regime. In fact, without an 't Hooft anomaly to force symmetry breaking, the fact that the theory has two dimensionless parameters suggests that the fate of chiral symmetry may be non-universal — that is, parameter-dependent — when $N = 4n + 1$.

The argument of [34, 35] for chiral symmetry breaking for all $N$ is based on bosonization. Applying non-Abelian bosonization [120] to $N^2 - 1$ Majorana fermions, one obtains an $O(N^2 - 1)$ level-1 Wess-Zumino-Witten model. Then one gauges an appropriate $SU(N)$ subgroup. The bosonic variable $O \in O(N^2 - 1)$ is related to the fermionic variables by $\psi_+^a \psi_-^b = \mu O^{ab}$, and $\mu$ is a renormalization scale. Setting $\mu \sim g$, Refs. [34, 35] observed that

$$\langle O^{ab} \rangle = \pm \delta^{ab}, \tag{149}$$

are degenerate minima of the classical action which are invariant under $SU(N)$ gauge transformations, $O \to \mathrm{Ad}(g) O \mathrm{Ad}(g)^\dagger$, where we view $\mathrm{Ad}(g) \in \mathrm{Ad}(SU(N)) \subset O(N^2 - 1)$. Configurations of the form of (149) are the only ones that satisfy this condition.

This argument is completely classical, while the theory is strongly interacting. To prove that chiral symmetry is spontaneously broken, one would have to prove that the configurations in Eq. (149) remain degenerate once all quantum effects are taken into account. We do not know how to do this without numerical lattice calculations. It is certainly possible that chiral symmetry may be spontaneously broken even when not required by any 't Hooft anomaly, and chiral symmetry is spontaneously broken for all $N$. It is also entirely possible that the result is $N$ dependent, as suggested by minimal 't Hooft anomaly matching.

**Screening vs. Confinement:** Let us now turn to the question of confinement, where the clash between our results and those of Ref. [35] is much more severe. To support their claim that the theory does not confine, Ref. [35] gave the following arguments:

(a) Using the Kutasov-Schwimmer universality result [153], Ref. [35] related the question of confinement in 2d adjoint QCD to confinement in 2d $SU(N)$ gauge theory with $N_f = N$ fundamental fermions. Generically theories with $N_f = N$ fundamental fermions have zero string tension, and Ref. [35] took this to imply that the same is true for 2d massless adjoint QCD.

(b) An explicit argument for the absence of confinement in 2d adjoint QCD with $N = 2$, where the physics looks very similar to that of the charge 2 Schwinger model.

(c) A relation between the Wilson loop expectation value and the chiral condensate.

(d) An argument based on considering the Schwinger-Dyson equations for Wilson loops.

Let us briefly discuss these one by one. Argument (a) is based on Ref. [153], whose idea can be summarized as follows. Consider three different theories: theory F, theory Adj, and theory F-Adj. Theory F is an $SU(N)$ gauge theory with $N_f = N$ massless fundamental Dirac fermions. Theory Adj is an $SU(N)$ gauge theory with one massless Majorana adjoint fermion — that is, 2d adjoint QCD. Theory F-Adj is a chiral gauge theory with $N_f = N$ left-moving Weyl fermions, and one Majorana-Weyl right-moving adjoint fermion. The chiral gauge theory is consistent because the possible gauge anomaly is controlled by the difference of left and right Kac-Moody levels, and here this difference is constructed to vanish. In light cone coordinates, the part of the Lagrangians containing the gauge fields and their couplings to fermions can be schematically written as

$$\mathcal{L}_{\mathrm{F}} = \frac{1}{2g} \mathrm{tr}(G^2) + A_+ J_{\mathrm{F},+} + A_- J_{\mathrm{F},-} \tag{150}$$

$$\mathcal{L}_{\mathrm{Adj}} = \frac{1}{2g} \mathrm{tr}(G^2) + A_+ J_{\mathrm{Adj},+} + A_- J_{\mathrm{Adj},-} \tag{151}$$

$$\mathcal{L}_{\mathrm{F-Adj}} = \frac{1}{2g} \mathrm{tr}(G^2) + A_+ J_{\mathrm{F},+} + A_- J_{\mathrm{Adj},-} . \tag{152}$$

Then Ref. [153] points out that if one takes e.g. $A_+ = 0$ gauge, the fundamentals decouple in $\mathcal{L}_{\text{F}-\text{Adj}}$ and the result looks the same as $\mathcal{L}_{\text{Adj}}$ in $A_+ = 0$ gauge. But if instead one takes the $A_- = 0$ gauge in $\mathcal{L}_{\text{F}-\text{Adj}}$, then the adjoint fermions decouple, and $\mathcal{L}_{\text{F}-\text{Adj}}$ looks the same as $\mathcal{L}_{\text{F}}$ in $A_- = 0$ gauge. The idea is then that for some appropriate observables, these three seemingly very different theories should have the same behavior. Generically, one expects theories with $N_f = N$ to have unsuppressed string breaking and a vanishing string tension. Putting these observations together lead Ref. [35] to conclude that the string tension should vanish also in the Adj theory, namely 2d massless adjoint QCD.

In our view, this conclusion does not follow from Ref. [153], due to several important issues.

1. The universality between the theories is argued to involve only the flavor-singlet massive mesons in Ref. [153]. String-breaking effects in the F theory, however, necessarily involve the flavor *non-singlet* sector: The heavy-light mesons which are pair created during string breaking carry charges in the fundamental representation under $SU(N_f)$. Therefore, string breaking physics, which is what is believed to give perimeter-law behavior for large Wilson loops in theories with fundamental fermions, is outside of the scope of the universality result.

2. Reference [153] argues that the flavored states of the F theory must lie in the massless sector, and they must be decoupled from the massive flavor-singlet states. Accepting this claim actually leads one to conclude that massive flavor-singlet mesons in the F theory cannot decay at large $N$. To see this, recall that the reason meson decays are possible in the Veneziano large $N$ limit (which is the large $N$ limit relevant to the F theory) is precisely the possibility of decays of flavor singlets to flavored states. If the singlets and non-singlets are decoupled, then the singlets cannot decay at large $N$. This means that the arguments of [153] actually imply that the F theory must have a massive spectrum consisting of an infinite tower of zero-width states. Using the color currents $J_{\pm}^{ij} = \sum_{a=1}^{N_f} \psi_{\pm}^{ia} \psi_{\pm}^{ja}$, we can build a list of flavor-singlet operators with increasing scaling dimension

$$\text{tr}(J_+ J_- J_- \cdots J_+), \tag{153}$$

which are expected to couple to states of increasing non-zero masses. If these massive flavor-singlet states are decoupled from massless flavored states, then they must all be stable at large $N$, and given the similarity of Eq. (153) to Eq. (134) one would expect a Hagedorn density of states in theory F.

3. In the universality argument of Ref. [153] one ignores the gauge-field zero modes. For some questions this is fine, but it is a dangerous feature for questions about e.g. Polyakov loops, whose physics is intimately related precisely to gauge field zero modes.

Argument (b) is correct for $N = 2$, which is where the explicit calculation was performed in Ref. [35], but the statement does not generalize to $N > 2$. This can be seen thanks to the recently-improved understanding of both the charge-$q$ Schwinger model in Refs. [69–71] and of 2d adjoint QCD, discussed here. The basic issue is that, in the charge-$q$ Schwinger model, there is a $\mathbb{Z}_{2q}$ chiral symmetry and a $\mathbb{Z}_q$ center symmetry. The $\mathbb{Z}_2$ subgroup of $\mathbb{Z}_{2q}$ is just fermion number. The match between the size of the internal zero-form symmetry group $\mathbb{Z}_{2q}/\mathbb{Z}_2 \simeq \mathbb{Z}_q$ and center symmetry group gives enough freedom to use chiral rotations to neutralize probe charges of any $q$-ality. In 2d adjoint QCD, however, the chiral symmetry is $\mathbb{Z}_2$ for any $N$, while the center symmetry is a $\mathbb{Z}_N$ symmetry. Thus, the complete neutralization of external probe charge via chiral rotations only works for $N = 2$. For generic even $N$, however, chiral rotations

can only neutralize external probes with $N$-ality $N/2$. For odd $N$, chiral rotations cannot be used to screen any probe charges at all.

Argument (c) is explicitly worked out for $N = 2$, and there the conclusion is correct. However, the generalization to higher $N$ is not valid, because it relies on an incorrect counting of fermion zero modes for generic $N$. To do so correctly requires the mod 2 index theorem proved in this paper.

Argument (d) in Ref. [35] is based on analyzing equations of motion for correlation functions of Wilson loops. As with any equation of motion, such Makeenko-Migdal type loop equations are derived by considering a small but otherwise completely generic variation of the appropriate quantity, which in this case is the shape of the Wilson loop [154,155].[23] This, however, is not the approach followed in Ref. [35], which instead considered a very special form for the field variations, specified in Eqs. 5.31, 5.32 in Ref. [35]. Since the variation considered in Ref. [35] is not generic, we see no reason to believe that it gives the correct equations of motion. Moreover, even if one restricts to such special variations, Ref. [35] assumes that the fermion-measure contribution is given by an anomaly equation. But there are other contributions which arise in computing the variation of the fermion measure, for example terms which involve $\text{tr}[\chi D_\mu a^\mu]$, where $\chi$ parametrizes the variation. The important roles of these terms in anomaly calculations and in 2d bosonization are discussed in e.g. Chapter 8 of Ref. [157]. These terms can be ignored in the context of anomaly calculations because in that setting they are cancelled by local counter terms. But as discussed in Chapter 8 of Ref. [157], they cannot simply be discarded in other cases, and we see no reason why they would not affect the would-be "equations of motion" obtained in a calculation along the lines of Ref. [35].

**Comments on numerical studies:** Lastly, let us comment briefly on the relation between our work and some numerical studies of 2d adjoint QCD, Refs. [5,36,46,49]. Reference [5] found numerical evidence from discretized light-cone quantization for confinement and Hagedorn behavior in massless 2d adjoint QCD, which is consistent with our results. But Ref. [5] did not see Bose-Fermi degeneracies at $m = 0$. This happened because Ref. [5] took the infinite-volume limit before the chiral limit $m \to 0$, and worked out the spectrum produced by local operators acting on a single chiral vacuum. With such a setup the Bose-Fermi pairing discussed in this paper are not visible.

References [36, 49] examined 2d adjoint QCD using non-Abelian bosonization. They also did not see any sign of Bose-Fermi pairing at $m = 0$, for the same reason as Ref. [5]. References [36, 49] also argued that there is no confinement at $m = 0$, on the basis of calculations that they interpreted as showing that the string tension vanishes at $m = 0$. These calculations assumed that parton-number changing processes are negligible, but, as the authors of Refs. [36, 49] mentioned themselves, there is no known rigorous justification for this simplifying assumption. Consequently, we do not find their results concerning the string tension persuasive.

Reference [46] also looked at the spectrum of single-trace states at large $N$ using discretized light-cone quantization. It gave numerical evidence that a few of the higher-energy states could be interpreted as combinations of weakly-interacting lower-mass states, and interpreted this as evidence for deconfinement. Reference [46] claimed that the many Regge trajectories seen in Ref. [5] should not be counted as fundamental, and there is only a single genuine Regge trajectory. But no evidence was given that multi-trace states have non-vanishing overlaps with single trace states at large $N$. Physically, such non-vanishing overlaps are required for a single-trace state to decay to a multi-trace state.

To summarize this section, it is our view that, with the benefit of hindsight, the 1990s studies of 2d large $N$ massless adjoint QCD which concluded that the theory is in a deconfined

---

[23]For a general discussion of such loop equations in gauge theories with adjoint matter, see, e.g., Ref. [156], while a discussion specifically in two spacetime dimensions was given in Ref. [140].

phase did not persuasively establish these claims.

# 10 Conclusions

Let us summarize our new results concerning the physics of 2d adjoint QCD.

- The massless version of the theory has two continuous dimensionless parameters when $N > 2$. If a mass is turned on, then its Lagrangian contains five continuous dimensionless parameters.

- At $m = 0$, the theory has a variety of 't Hooft anomalies when $N = 4n, 4n + 2, 4n + 3$. These anomalies are, of course, robust against variations of symmetry-preserving perturbations, such as e.g. the four-Fermi terms in Eq. (14).

- Minimal 't Hooft anomaly matching as well as controlled semi-classical calculations on small $\mathbb{R} \times S^1$ suggest that, in the $\mathbb{R}^2$ limit, the massless theory confines test charges of all non-trivial $N$-alities, unless $N$ is even, in which case test charges of $N$-ality $N/2$ are deconfined.

- The $\mathbb{Z}_2$ chiral symmetry of the massless theory is spontaneously broken for $N = 4n, 4n+2, 4n+3$. This conclusion is supported by minimal 't Hooft anomaly matching as well as controlled semi-classical calculations. We find no evidence that chiral symmetry is spontaneously broken when $N = 4n + 1$. If chiral symmetry breaks on $\mathbb{R}^2$ when $N = 4n+1$, this would be a non-universal consequence of strong interactions, and its fate may depend on the values of the dimensionless parameters.

To prove the existence of the 't Hooft anomalies, we used a recent mod 2 index theorem.
Our results have a number of interesting corollaries and lessons.

- One general lesson is that semi-classics does not lie when done correctly: there is a complete match between the predictions of 't Hooft anomaly matching and our semi-classical calculations. This highlights the value of the modern semi-classical approach to confining gauge theory developed in Refs. [9, 10].

- Adjoint QCD describes an interacting system of $K = N^2 - 1$ Majorana fermions in one spatial dimension. When adjoint QCD has 't Hooft anomalies, turning on a negative mass gives rise to associated systems of interacting Majorana fermions which describe SPT phases of matter. A notable case is when $N = 4n + 3$, where we have an 't Hooft anomaly and $K$ is divisible by 8. This implies that 2d massive adjoint $SU(4n + 3)$ QCD describes non-trivial SPT phases of matter built from $K = 8, 48, 120, 224, \ldots$ interacting Majorana fermions, protected by charge conjugation symmetry as defined in the gauge theory.

- Massless 2d adjoint QCD is not supersymmetric. But for even $N$, its spectrum on a circle features exact Bose-Fermi degeneracies, which can be traced to existence of degenerate bosonic and fermionic vacua. For odd $N$, there is in general no Bose-Fermi degeneracy, but it should emerge at large $N$ if we assume smoothness of the large $N$ limit.

- Our work highlights the point that exact Bose-Fermi degeneracies in a two-dimensional quantum field theory need not be synonymous with supersymmetry, if the latter term is taken to refer to invariance under one of the standard supersymmetry algebras in two spacetime dimensions.

- For generic $m \neq 0$, the fermionic vacuum gets lifted, and the exact Bose-Fermi degeneracies disappear. But their large $N$ consequences remain and become even more interesting. All Hagedorn growth in the bosonic and fermionic densities of states remains identical, and the $(-1)^F$-graded partition function continues to enjoy dramatic Bose-Fermi cancellations, which at $m \neq 0$ take place within the same super-selection sector of Hilbert space, rather than than between super-selection sectors, as is the case at $m = 0$.

- Many of the basic features of 2d massless adjoint QCD, such as confinement and chiral symmetry breaking, are also expected in 4d massless adjoint QCD. From the perspective of drawing lessons for 4d physics from studies of 2d toy models, it is encouraging that the 2d theory indeed confines and breaks chiral symmetry, at least for most values of $N$.

- Recent studies of 4d adjoint QCD at large $N$ have suggested that it has remarkable spectral properties [19–22], in addition to its attractive tractability under compactification [9]. It is interesting — and hopefully useful — that the same is true for 2d adjoint QCD, given the more constrained nature of physics in two spacetime dimensions.

We suspect that so far we have seen only a small part of the beauty of adjoint QCD in various dimensions.

## Acknowledgments

We are very grateful to A. Armoni, D. Dorigoni, S. Dubovsky, L. Fidkowski, I. Klebanov, D. Kutasov, E. Poppitz, S. Sen, M. Shifman, A. Smilga, and R. Thorngren for comments and discussions. We also thank an anonymous referee for some comments that significantly improved our exposition. Part of this work was done during the conference "Topological Solitons, Non-perturbative Gauge Dynamics and Confinement 2" at the University of Pisa, and A. C. and Y. T. are grateful for the kind hospitality of the organizers. A. C. is supported by startup funds from UMN, T. J. is supported by a UMN CSE Fellowship, Y. T. is supported by JSPS Overseas Research Fellowships, and M. Ü. is supported by the U.S. Department of Energy via the grant DE-FG02-03ER41260.

## A  Conventions for two-dimensional Majorana fermions

Here, we summarize our conventions for two-dimensional Majorana fermions in Euclidean signature. The two-dimensional Clifford algebra with positive-definite signature is

$$\{\gamma_\mu, \gamma_\nu\} = 2\delta_{\mu\nu}. \tag{154}$$

We can realize these gamma matrices as $2 \times 2$ real symmetric matrices. Concretely, we take

$$\gamma^1 = \sigma_1 = \begin{pmatrix} 0 & 1 \\ 1 & 0 \end{pmatrix}, \ \gamma^2 = \sigma_3 = \begin{pmatrix} 1 & 0 \\ 0 & -1 \end{pmatrix}. \tag{155}$$

Euclidean $SO(2)$ rotations should be lifted to the double cover Spin(2) when acting on spinors. The Spin(2) group is generated by

$$\Sigma_{12} = -\Sigma_{21} = \frac{1}{4}[\gamma^1, \gamma^2] = \frac{-i}{2}\sigma_2. \tag{156}$$

We can define chirality using

$$\gamma = i\gamma^1\gamma^2 = \sigma_2. \tag{157}$$

Since $\gamma$ commutes with $\Sigma$, the two-dimensional irreducible representation of the Clifford algebra decomposes into the positive and negative chirality states as representations of Spin(2).

With our conventions, the Dirac operator,

$$\slashed{D} = \gamma^\mu D_\mu, \tag{158}$$

can be regarded as a real and anti-symmetric operator. Suppose that $\psi$ is a real two-component spinor. Then the associated free action is

$$S = \int \psi^T (i\slashed{D} - m\gamma)\psi. \tag{159}$$

This is the Euclidean action of a free two-dimensional Majorana fermion. The partition function can be viewed as the Pfaffian,

$$Z_\psi = \int \mathcal{D}\psi \exp(-S) = \mathrm{Pf}(i\slashed{D} - m\gamma). \tag{160}$$

The system has reflection symmetries if $m = 0$. For example, reflection along $x_1$ coordinate act as

$$R_1 : \psi(x_1, x_2) \mapsto \gamma^2 \psi(-x_1, x_2). \tag{161}$$

Similarly,

$$R_2 : \psi(x_1, x_2) \mapsto \gamma^1 \psi(x_1, -x_2). \tag{162}$$

These reflections satisfy $R_i^2 = 1$ and commute with the Dirac operator $\slashed{D}$. This means that they can be used to define a pin$^+$ structure on unorientable manifolds. These reflections anticommute with the mass term, so they are symmetries only if $m = 0$.

The Majorana fermion $\psi$ can be rewritten in terms of two one-component Majorana-Weyl fermions $\psi_\pm$:

$$\psi = \frac{1}{\sqrt{2}} \left( \psi_+ \begin{pmatrix} 1 \\ i \end{pmatrix} + \psi_- \begin{pmatrix} i \\ 1 \end{pmatrix} \right). \tag{163}$$

Then the action becomes

$$S = \int \begin{pmatrix} \psi_+ & \psi_- \end{pmatrix} \begin{pmatrix} -D_1 + iD_2 & +im \\ -im & -(D_1 + iD_2) \end{pmatrix} \begin{pmatrix} \psi_+ \\ \psi_- \end{pmatrix}. \tag{164}$$

This expression makes it easy to see the emergence of the discrete chiral symmetry at $m = 0$. But this basis is not convenient for the proof of the mod 2 index theorem in Sec. 4.

Let us prove that the Pfaffian of the non-zero-mode part of the Dirac operator, $\mathrm{Pf}'(i\slashed{D} - m\gamma)$ can be consistently defined to be positive semi-definite for real $m$. This can be done using the notion of Majorana positivity [84–86]. Let us use the result of Ref. [86]: Given an anti-symmetric $4n \times 4n$ matrix $P$ of the form

$$P = \begin{pmatrix} P_1 & iP_2 \\ -iP_2^T & P_3 \end{pmatrix}, \tag{165}$$

with $2n \times 2n$ complex matrices $P_i$, $\mathrm{Pf}(P) \geq 0$ if $P_2$ is semi-positive, $P_2 = P_2^\dagger$, and $P_3 = -P_1^\dagger$. In our case, $P_1 = -D_1 + iD_2$, $P_3 = -(D_1 + iD_2) = -P_1^\dagger$, and $P_2 = m$, so the necessary conditions are satisfied if these matrices are even-dimensional. The fact that they are indeed even dimensional can be established by consulting the discussion of the non-zero Dirac spectrum in Sec. 4. So we find that $\mathrm{Pf}'(i\slashed{D} - m\gamma) \geq 0$.

More generally, recall that the action of 2d adjoint QCD includes four-Fermi interactions. For lattice calculations, it is convenient to make the fermion action quadratic in the Grassmann

fields by introducing bosonic auxiliary fields using a Hubbard-Stratanovich transformation. Then one can do the Grassmann path integral, obtaining a Pfaffian that depends on the auxiliary fields. For lattice calculations it will be important to understand the positivity properties of such Pfaffians. We suspect that depending on sign of the four-fermion terms, the auxiliary fields may induce sign problems, and the Majorana positivity discussed in Refs. [84–86] would again play an important role.

# B  Mod 2 index on a two-sphere $S^2$ with 't Hooft flux

In this Appendix, we show that the formula (42) holds on the two-sphere $S^2$. Since the sphere has a nonzero curvature, let us compute the Dirac operator of the free Majorana spinor explicitly first. Taking the usual spherical coordinates $(\theta, \varphi)$, the metric is

$$\mathrm{d}s^2 = \mathrm{d}\theta \otimes \mathrm{d}\theta + \sin^2\theta \mathrm{d}\varphi \otimes \mathrm{d}\varphi. \tag{166}$$

We introduce the vielbein $e^a = e^a_\mu \mathrm{d}x^\mu$ with $e^a_\mu e^b_\nu g^{\mu\nu} = \delta^{ab}$:

$$e^1 = \mathrm{d}\theta, \ e^2 = \sin\theta \mathrm{d}\varphi, \tag{167}$$

and define the curved-space gamma matrices as $\gamma^\mu = e^\mu_a \gamma^a$:

$$\gamma^\theta = \gamma^1 = \sigma_1, \ \gamma^\varphi = (\sin\theta)^{-1}\gamma^2 = (\sin\theta)^{-1}\sigma_3. \tag{168}$$

The Levi-Civita spin connection $\omega^{ab}$ is defined by the torsion-free condition, $\mathrm{d}e^a + \omega^a_{\ b} e^b = 0$, and $\omega^{ab} + \omega^{ba} = 0$. This gives

$$\omega^1_{\ b} \wedge e^b = 0, \ \cos\theta \mathrm{d}\theta \wedge \mathrm{d}\varphi + \omega^2_{\ a} \wedge e^a = 0. \tag{169}$$

The first condition requires $\omega^{12} \propto \mathrm{d}\varphi$, and the second condition determines the coefficient as

$$\omega^{12} = -\omega^{21} = -\cos\theta \mathrm{d}\varphi, \tag{170}$$

while the other components vanish. The covariant derivative on spinors is defined as

$$D_i \psi = \left( \partial_i + \frac{1}{2}\omega^{ab}_i \Sigma_{ab} \right)\psi, \tag{171}$$

and

$$D_\theta = \partial_\theta, \ D_\varphi = \partial_\varphi + \frac{1}{2}(-\cos\theta)(-\mathrm{i}\sigma_2). \tag{172}$$

The Dirac operator $\slashed{D} = \gamma^i D_i$ is

$$
\begin{aligned}
\slashed{D} &= \gamma^\theta D_\theta + \gamma^\varphi D_\varphi = \sigma_1 \partial_\theta + \sigma_3 \left( \frac{\partial_\varphi}{\sin\theta} + \frac{\mathrm{i}}{2}\cot\theta \sigma_2 \right) \\
&= \sigma_1 \left( \partial_\theta + \frac{1}{2}\cot\theta \right) + \sigma_3 \frac{\partial_\varphi}{\sin\theta}.
\end{aligned}
\tag{173}
$$

Let us set $x = \cos\theta \in [-1, 1]$. Then $\partial_x = (-\sin\theta)^{-1}\partial_\theta$ so that

$$\sigma_1 \slashed{D} = -\sqrt{1-x^2}\left[ \partial_x + \frac{1}{1-x^2}\left( -\frac{1}{2}x + \mathrm{i}\gamma \partial_\varphi \right) \right]. \tag{174}$$

Because $\psi$ is a spinor and has half-integer spin, $\mathrm{i}\partial_\varphi = (m + 1/2)$ with some integer $m$. This means that that there are no square-integrable zero modes with zero 't Hooft flux, i.e. $\zeta_{\text{free}} = 0$.

Now, we introduce $SU(N)$ gauge fields with 't Hooft flux. We separate $S^2$ into northern ($x \geq 0$) and southern ($x \leq 0$) half-spheres. The connection formula relating the fields in the two hemispheres is

$$\psi_{\geq 0}(0, \varphi) = \Omega^{\dagger}(\varphi)\psi_{\leq 0}(0, \varphi)\Omega(\varphi). \tag{175}$$

The cocycle condition gives

$$\Omega(\varphi + 2\pi) = \Omega(\varphi)\exp\left(\frac{2\pi i}{N}k\right), \tag{176}$$

and the label $k$ is the 't Hooft magnetic flux. For computation of mod 2 index, we can pick the simplest non-trivial example

$$\Omega(\varphi) = \exp\left(\frac{ik\varphi}{N}T\right), \tag{177}$$

where $T = \mathrm{diag}(1, \ldots, 1, -(N-1))$.

On the gauge fields, this coordinate-dependent transition function requires that

$$a_{\theta, \geq 0}(x, \varphi) = \Omega^{\dagger}a_{\theta, \leq 0}(x, \varphi)\Omega, \; a_{\varphi, \geq 0}(0, \varphi) = \Omega^{\dagger}a_{\varphi, \leq 0}(0, \varphi)\Omega + \frac{k}{N}T. \tag{178}$$

An example of such $SU(N)$ gauge field that satisfies this condition is one where $a_{\theta} = 0$ on the whole sphere, and

$$a_{\varphi}(x, \varphi) = \begin{cases} \dfrac{1-x}{2}\dfrac{k}{N}T & (x \geq 0), \\ -\dfrac{1+x}{2}\dfrac{k}{N}T & (x < 0). \end{cases} \tag{179}$$

By formally extending the above gauge field configurations $a_{\geq}$ and $a_{\leq}$ to the whole sphere, these configurations are related by the $x$-independent gauge transformations, $a_{\geq} = \Omega^{\dagger}a_{\leq}\Omega - i\Omega^{\dagger}d\Omega$. By gauge covariance, the problem of finding the normalizable zero modes is reduced to finding the square-integrable functions $f : [-1, 1] \to \mathfrak{su}(N)$ satisfying

$$\partial_x f + \frac{1}{1-x^2}\left(-\frac{1}{2}x + m + \frac{1}{2} - \frac{1-x}{2}\frac{k}{N}\mathrm{ad}(T)\right)f = 0 \tag{180}$$

for some integer $m$. The adjoint representation of $T$, $\mathrm{ad}(T)$, has eigenvalues $+N$ with degeneracy $(N-1)$, $-N$ with degeneracy $(N-1)$, and $0$ with degeneracy $(N-1)^2$. On the subspace $\mathrm{ad}(T) = 0$, the twist has no effect and there are no square-integrable zero modes.

When $\mathrm{ad}(T) = N$, then we get

$$\partial \ln f = \frac{1}{1-x^2}\left(\frac{1-k}{2}x + \frac{k-2m-1}{2}\right), \tag{181}$$

and we can solve this as

$$f^2 \propto (1-x)^m(1+x)^{k-1-m}. \tag{182}$$

$f$ is square integrable if and only if

$$0 \leq m \leq k-1. \tag{183}$$

When $k > 0$, then there are $k(N-1)$ zero modes with positive chirality.

When $\mathrm{ad}(T) = -N$, we just need to flip the sign of $k$ in the above analysis, and we find no zero modes with positive chirality for $k > 0$. As a consequence, we obtain

$$\zeta = k(N-1). \tag{184}$$

This agrees with Eq. (42) for the mod 2 index.

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
