# Peer review of "Anomalies, a mod 2 index, and dynamics of 2d adjoint QCD"

_SciPost Physics, doi:SciPost Phys. 8, 072 (2020)_

## Round 2 · Referee Report · Anonymous (Referee 1) · 2019-10-7

Report

This is a beautifully written paper, bringing some very modern tools in quantum field theory (discrete 't Hooft anomalies and generalized global symmetries) to bear on an old problem, the dynamics of adjoint QCD in 1+1 dimensions. The calculation of the discrete anomalies is very well documented and quite convincing. With some further assumptions on the dynamics of the theory (trying to rule out that the theory is topological or conformal) the authors obtain some remarkable results about the physical properties of the theory, including statements about when chiral symmetry is broken (as a function of N, the number of colors) and about an exact Bose/Fermi degeneracy in the spectrum.

I'd like the paper to be eventually published, but there are two issues I'd like to see discussed a little more clearly:

1) The authors disagree with much of the previous literature. As such their results should be evaluated critically and it is their obligation to try to do their best to identify shortcomings in the previous literature. They do a remarkable job in discussing how their analysis avoids the conclusion of [35]. [35] is a based on formal arguments that rely on certain assumptions and the authors make a good case that these assumptions don't hold. What is more troubling to me is that the new results presented here disagree with [5,36,46,49]. These references present explicit numerical calculations which, in a theory that is mostly of theoretical interest and hence does not allow experimental verification, should be the ultimate arbiter. I believe the authors should do more to explain why these numerical studies fail to find the physics they claim to be correct. This does not have to be as detailed and explicit as the nice comparison to [35], but a little more should be said.

2) The authors find a Bose/Fermi degeneracy without supersymmetry (SUSY). Their main argument for the lack of SUSY is that the UV theory is manifestly non-SUSY. But is there a chance that SUSY emerges at low energies? It would be nice to see a few more comments/details on how one can tell from the answer for the physical spectrum that SUSY is broken. Do fermions and bosons interact differently despite their degenerate masses?

---

## Round 2 · Referee Report · Anonymous (Referee 2) · 2019-10-31

Strengths

1) The paper is very clear, very well written. 2) The paper contains many interesting results that contradict old results. The authors explain where the previous works had made mistakes. Fixing incorrect results is of course an important part of scientific progress. 3) The authors attack the problem from several directions and apply a variety of new theoretical ideas to understand the fate of the theory. These include for example anomaly matching and semiclassical analysis on $\mathbb{R}\times S^1$. Deriving the same results using different methods makes the conclusions of the paper very convincing. 4) The results of the paper strengthen the relations between 2d and 4d gauge theories. In addition to confinement and chiral symmetry breaking, adjoint QCD in 2 and 4 dimensions also share the surprising feature of Bose-Fermi degeneracy without supersymmetry as pointed out by the authors. This relation can hopefully help in understanding the origin of this peculiar phenomenon.

Weaknesses

1) While the authors invest a lot in settling the disagreements with refs [34,35], this is not the case with the numerical results of refs [5,36,46,49]. The authors are aware of this problem but didn't try to explain it. 2) The conclusion of the discussion around equations (8.2),(8.3) seems correct but the explanation is confusing. If the operators $a_B^\dagger$ and $a_F^\dagger$ are built out of both zero and non-zero modes, it might be that $a_B^\dagger|0>_B$ for example is identically zero while $a_B^\dagger|0>_F$ is not. I think that a better way to rephrase the argument is to start from the zero modes sector which is explicitly Bose-Fermi degenerate as seen in section 7.2.1 (assuming the result can be generalized to any even N), and then act on these states with the operators $a_B^\dagger$ and $a_F^\dagger$, built now only out of non-zero modes. 3) The construction of the spectrum in equation (7.58) is very beautiful. However, it is not clear from the presented analysis whether the vacuum states are twofold or fourfold degenerate. These two possibilities are of course physically distinct and lead to different patterns of symmetry breaking. This depends on the values and signs of the parameters $\epsilon_{1,2}$. Since the authors know in principle how to compute these parameters, it can be interesting to see a small discussion about the possible true vacua and whether the result is universal or depends on the microscopic parameters of the 2d theory (or at least a comment if the computation is too hard).

Report

The authors revisit an old problem of 2d adjoint QCD. Using a wide set of modern tools, such as generalized symmetries and their associated anomalies, new index theorem, SPT classification and more, they arrive to conclusions which partially disagree with previous results. The authors are fully aware of these discrepancies and it is clear that they put a lot of thought and effort in order to explain the source for the contradictions. The paper is very well written, very clear and the arguments presented are convincing. On the way, the Authors also clarify several points that could have been a source for confusion. The first example for this is the discussion about the hidden assumptions in the statement that discrete 1-form symmetries cannot be broken in two dimensions. Another example is a comment about the importance of charge conjugation symmetry for whether 8 Majorana fermions are in a non-trivial SPT phase or not. To summarize, the paper is interesting and important for several reasons. It fixes incorrect statements about the vacuum of adjoint QCD, contains new and interesting results such as the analysis of anomalies and their realization in the vacuum, the Bose-Fermi degeneracy and more, it clarifies and explains in a clear way several subtle points, and sharpen the application of new (and old) theoretical ideas to strongly interacting field theories.

Requested changes

No changes are needed except for some small typos: 1) Page 2 below eq (1.1): "an Majorana"-> "a Majorana" 2) Page 12 two lines below eq (4.7): "the the"->"the" 3) Page 17, item (b): "are longer zero modes"-> "are no longer zero modes" 4) Page 26 line 9: "that someone in its parameter space"-> "that somewhere in its parameter space" 5) Page 28 line 4 from the end: "in the sense then"-> "in the sense that" 6) Page 34 before eq (7.26): "these quasi-ground state"->"these quasi-ground states" 7) Page 39 line 8: "each other the action of"-> "each other under the action of" 8) page 45 before eq (8.14): "is dominated the Casimir"-> "is dominated by the Casimir"

---

## Round 2 · Referee Report · Anonymous (Referee 3) · 2019-11-2

Strengths

The authors have studied 2 dimensional adjoint QCD by using the modern technology of global anomaly matching involving various symmetries and semiclassical analysis.

Weaknesses

Some point needs to be clarified.

Report

This is a very interesting paper which studies the dynamics of 2 dimensional adjoint QCD which is strongly coupled and cannot be solved exactly. I have one major question and a few minor questions/comments listed below.

A major question:

1) I would like to ask whether the center symmetry is really broken to a subgroup. For example, in a semiclassical analysis with periodic boundary condition, there are two states given in (7.54). It is natural to organize the states to be eigenstates of the fermion parity $(-1)^F$ because it is unbroken in the large volume limit. The two states in (7.54) are such eigenstates. Each of them breaks the chiral symmetry, but preserves the center symmetry according to the analysis above (7.54).

In fact, the mixed anomaly between the chiral symmetry and the center symmetry only requires that one of the symmetries is broken. It is not necessary (although possible) that both of them are broken.

The argument the authors give for the breaking of the center symmetry is given in Sec 5.1. I was not fully convinced by the argument there. The argument is that the domain wall for the chiral symmetry breaking has $N$-ality $N/2$, and both sides of the domain walls have the same energy. However, the criterion for the center symmetry breaking is by Wilson loops which (at least naively) do not interpolate two vacua related by the chiral symmetry breaking. If we are just in one of the chiral symmetry breaking vacua, I think it is possible that the charge with $N$-ality $N/2$ is confined in that one vacuum. (I personally suspect that the center symmetry is not broken at all.)

At least the authors should comment on how to interpret the result of the semiclassical analysis, because (after taking the eigenstates of $(-1)^F$) the center symmetry is not broken in (7.54).

Minor questions/comments:

2) The authors seem to criticize the theory without the 4-fermi interactions. However, I think there is nothing wrong to set them to be zero. Without 4-fermi interactions, the theory is super-renormalizable. In super-renormalizable theory, a marginal term is not necessary for renormalization (in the same sense that irrelevant terms are not necessary for renormalization of renormalizable theories). So it is consistent to set them to be zero. It is just a matter of taste whether to call it "fine tuning". (In other words, if it is "fine tuning" to set them to zero, it is also "fine tuning" to set all irrelevant operators to zero.)

3) I think there is a possible reason that the chiral symmetry is broken in the case $N=4n+3$. If we require a smooth large $N$ limit, they may be nonzero for all sufficiently large values of $N$ regardless of the anomaly matching. Otherwise, the chiral symmetry breaking must occur at the subleading order of large $N$ expansion. That is logically possible, but a bit surprising if it is true.

4) The authors have discussed some SPT phase interpretations of the anomaly after adding a mass term to the fermions in a few paragraphs below (6.1). A small comment is that the SPT phase with the charge conjugation symmetry, which is $Z_2$, can be seen, for example in Table 1 of https://arxiv.org/pdf/1406.7329.pdf Neglecting the parity (or time-reversal) symmetry, the relevant group is $\Omega^{spin}_2(BZ_2)=Z_2 \times Z_2$. It is indeed consistent with what the authors found in the paper by more explicit analysis (after a mass deformation).

Requested changes

I would like to request the authors to address (or at least comment on) the major point above, because it is related to the main result of the paper. The minor points are optional.

---

## Round 3 · Author Response

Warnings issued while processing user-supplied markup:

  • Inconsistency: plain/Markdown and reStructuredText syntaxes are mixed. Markdown will be used.
    Add "#coerce:reST" or "#coerce:plain" as the first line of your text to force reStructuredText or no markup.
    You may also contact the helpdesk if the formatting is incorrect and you are unable to edit your text.

We are very grateful to the referees for their positive comments and helpful questions and suggestions. We have edited our paper to take them into account, and are providing a description of the changes and comments on the questions below.

1) Two of the referees, Referee 1 and 2, asked us to comment on the discrepancy between our results and the previous results of numerical studies, especially Refs. [5,36,46,49]. Indeed, some of them claim deconfinement for massless adjoint QCD, and none of them observe Bose-Fermi degeneracy. We want to discuss the problems about confinement/deconfinement and about Bose/Fermi degeneracy separately.

About confinement/deconfinement, our results indeed agree with [5], but disagree with the interpretation of the results in [46]. Ref.[5] shows that there are infinitely many Regge-like trajectories in the large-N limit, with a density of states that exhibits Hagedorn growth. This is consistent with the hypothesis of confinement, but not with the hypothesis of deconfinement. Ref.[46] gives a reinterpretion of the same computation, and argues that some states on higher Regge trajectories can be decomposed into combinations of states from lower Regge trajectories, and uses this to claim deconfinement. However, Ref.[46] and later papers are just giving interpretation by looking at the spectrum of energies of single-trace states, but do not show that those single-trace states are indeed unstable in the large-N limit. Therefore, we do not think that this counts as persuasive evidence for deconfinement. The numerical and analytic calculations of the string tensions in [36,49] involve some uncontrolled approximations, involving e.g. neglect of parton-number changing processes. Consequently, we do not find the results persuasive.

About Bose/Fermi degeneracy, our claim is that two vacua related by broken discrete chiral symmetry have opposite fermion parity. Since all the numerical studies are using light-cone quantization, they are studying the spectrum of states created by local operators acting on one of these vacua. The fact that the other vacuum exists and has opposite fermion parity was not known prior to our work. Therefore, it is natural that Bose/Fermi degeneracy has not been observed. We have clarified this issue in the paper in the beginning of Section 8.

2 ) About the second question by Referee 1, we do not think that there is low-energy SUSY, for a number of reasons. First, if SUSY was only emergent, Bose-Fermi degeneracies could not be exact for all states - yet they are. If the theory did have exact SUSY, but it were spontaneously broken, then there would be two vacua with opposite fermion parity, as we have seen in 2d adjoint QCD. But spontaneously broken SUSY requires a massless Goldstino fermion to be present in the spectrum. There is no evidence for such a massless excitation from any of the studies of 2d adjoint QCD, including our study. Indeed, there is a point in parameter space of the massless theory where the absence of a massless fermion can be shown analytically, as discussed at the beginning of our Section 5, where we review an argument by Kutasov from the 1990s.

To take into account these answers to the questions, we have reorganized and improved the exposition of Section 8, which focuses on Bose-Fermi degeneracies and their interperation. We have also added a discussion of the relation between our work and numerical studies of 2d adjoint QCD from the 1990s at the end of Sec. 9.

====

3) About the second comment by Referee 2, we agree on their suggestion. We now separate the role of fermionic zero mode in Sec. 7.2.1, and the creation/annihilation operators are created out of non-zero modes.

4) About the third comment by Referee 3, the sign of four-fermion couplings generically depends both on the original four-fermion couplings $c_1, c_2$, and the perturbative corrections out of the gauge coupling, and we do not have a universal answer. Nevertheless, the symmetries allow us to discuss the degeneracies of the states, and this is the primary purpose of that section. So, we added a comment that we do not compute the matching of those coefficients.

====

5) Referee 3 asked whether it is really true that center symmetry is broken to subgroup, and asked us to explain this in the context of the analysis of Eq. 7.54.

We have clarified the discussion in that section and in several other places. Center symmetry is indeed broken to a subgroup due to the speciality of $1$-form symmetry in $(1+1)$ spacetime dimensions. Since our space is just a line, putting a test particle divides our space into two disconnected pieces. Therefore, we should now look for the lowest-energy states on each side while satisfying the boundary condition.

The mixed anomaly between the center and chiral symmetries suggest that the chiral generator has the charge $N/2$. Therefore, when the test particle has the charge $N/2$, we can consistently choose the ground states on both sides. We clarify this point by adding the footnote 13.

About the non-contractible loops on the torus, there is another subtlety due to the mixed anomaly between fermion-parity and chiral symmetry. When we take the periodic boundary condition, this anomaly survives under compactification, and thus the deconfined noncontractible line should become a composite of the Polyakov loop and the fermionic KK zero mode, $C, C^\dagger\sim \tr(P\psi_{\pm})$. We point out the existence of this subtlety in the footnote 13, and gives more explicit comments in Sec. 7.2.1.

6) Referee 3 argued that it is consistent to set the four-Fermi terms to zero, and said that this is not a fine-tuning.

We agree that it is consistent to set the symmetry-allowed four-Fermi to zero in a superrenormalizable theory, in the sense that these classically marginal terms are not necessary as UV counter-terms.

Still, we believe that it is customary to consider all the renormalizable interactions that are consistent with symmetry. Therefore, in the spirit of effective field theory, it is useful to consider the implications of classically marginal four-fermion interactions which are consistent with chiral symmetry.

Taking into account Referee 3's suggestion, we changed our presentation about the renormalizability and fine-tuning in order to clarify these points.

7) About the third comment of Referee 3, we also think that the smooth large-N assumption is reasonable on $\mathbb{R}^2$, and the chiral symmetry breaking would occur for a reasonable range of the parameter space. Still, we wanted to point out the logical possibility that there can be a region with non-zero measure that the N=4n+1 theory shows the unique ground state. Indeed, the $S^1$ compactified setup is an example of such a unique ground state.

8) Taking into account Referee 3's fourth comment, we changed our wordings in Sec. 6.

We again thank the referees for their detailed and very helpful comments!

Best regards, Aleksey Cherman, Theodore Jacobson, Yuya Tanizaki, Mithat Unsal

---

## Round 3 · List of Changes

-Altered equation 1.1 to include perturbatively marginal four-Fermi terms allowed by symmetries, and revised footnote 1 to reflect this change in presentation.

-Corrected the proof given in footnote 3 that we have found all internal symmetries of the theory.

-Revised the paragraphs at the end of page 6 regarding the four-Fermi terms to address comments made by Referee 3.

-Added footnote 13 to address questions raised by Referee 3 about the breaking of center symmetry to a subgroup.

-Changed the wording in the second to last paragraph of section 6, taking into account the fourth comment by Referee 3.

-Added a paragraph on page 39 clarifying which non-contractible line operator is deconfined on the cylinder with periodic boundary conditions.

-Added two new paragraphs just before section 8.1 to emphasize why numerical light-cone studies in the literature did not observe Bose-Fermi degeneracy.

-Changed the wording leading up to equation 8.2 to incorporate the suggestions of Referee 2.

-Added figure 4 to illustrate the Bose-Fermi degeneracy of the massless theory on the cylinder.

-Added a discussion at the top of page 46 about the vanishing of the charge-conjugation-twisted partition function at large N.

-Reorganized section 8 to include a subsection focusing on Bose-Fermi degeneracy without supersymmetry. We removed the misleading example of a free Majorana fermion in two spacetime dimensions, and added arguments ruling out spontaneously broken SUSY or emergent SUSY at long distances.

-Added a discussion in section 9 further commenting on the discrepancy between our results and numerical studies in the literature.

-Changed the wording in the third bullet point in section 10.

-Corrected typos generously pointed out by Referee 2.

---

## Editorial Decision

published